# ARTICLES

## OPEN

# Short- and long-range interactions in the HIV-1 5′ UTR regulate genome dimerization and packaging

Liqing Ye[1], Anne-Sophie Gribling-Burrer[1,4], Patrick Bohn[1,4], Anuja Kibe [1], Charlene Börtlein[1], Uddhav B. Ambi[1], Shazeb Ahmad[1], Marco Olguin-Nava[1], Maureen Smith[3], Neva Caliskan [1,2], Max von Kleist[3] and Redmond P. Smyth [1,2] ✉

RNA dimerization is the noncovalent association of two human immunodeficiency virus-1 (HIV-1) genomes. It is a conserved step in the HIV-1 life cycle and assumed to be a prerequisite for binding to the viral structural protein Pr55Gag during genome packaging. Here, we developed functional analysis of RNA structure-sequencing (FARS-seq) to comprehensively identify sequences and structures within the HIV-1 5′ untranslated region (UTR) that regulate this critical step. Using FARS-seq, we found nucleotides important for dimerization throughout the HIV-1 5′ UTR and identified distinct structural conformations in monomeric and dimeric RNA. In the dimeric RNA, key functional domains, such as stem-loop 1 (SL1), polyadenylation signal (polyA) and primer binding site (PBS), folded into independent structural motifs. In the monomeric RNA, SL1 was reconfigured into long- and short-range base pairings with polyA and PBS, respectively. We show that these interactions disrupt genome packaging, and additionally show that the PBS–SL1 interaction unexpectedly couples the PBS with dimerization and Pr55Gag binding. Altogether, our data provide insights into late stages of HIV-1 life cycle and a mechanistic explanation for the link between RNA dimerization and packaging.

Human immunodeficiency virus-1 (HIV-1), like all retroviruses, packages two copies of its genome into viral particles. These genomes are noncovalently associated at an RNA motif called the dimerization initiation site (DIS). This association, known as dimerization, affects multiple steps of the HIV-1 life cycle[1,2]. Dimerization is assumed to be a prerequisite for genome packaging into virions, although the mechanistic relationship between dimerization and packaging is still under debate[3–6]. It also plays a role in genome integrity and evolution by bringing two genomes in close proximity for strand-switch recombination[3,7–10]. Finally, it is linked to a structural switch that may regulate genome packaging and translation within cells[5,11–16] (Fig. 1a).

An extensive body of work maps the DIS to stem-loop 1 (SL1) of the HIV-1 5′ untranslated region (UTR)[17–19] (Fig. 1b). SL1 contains a six-nucleotide long GC-rich palindromic sequence that initiates dimerization through an inter-molecular 'kissing loop' interaction[20–22]. Although SL1 is widely considered the primary dimerization motif, numerous studies indicate that genome dimerization is also modulated by sequences outside SL1 (refs. [4,17,18,23–26]). For example, dimerization is promoted by a long-range base pairing between nucleotides overlapping the *gag* start codon (AUG) and the unique 5′ element (U5)[12,26,27] (Fig. 1b). Alternatively, it is inhibited when the region containing the AUG folds into a small hairpin, in turn freeing U5 to form a pseudoknot interaction with SL1 (refs. [12,28]) (Fig. 1b). The U5–SL1 pseudoknot interaction was originally proposed as a liable interaction between the loop region of SL1 and U5, but a recent nuclear magnetic resonance (NMR) study uncovered a more extensive base pairing between U5 and SL1 (refs. [13,14,28]). Furthermore, intrinsic transcriptional start site heterogeneity, which produces transcript variants beginning with different counts of G residues (1G, 2G or 3G), has been shown

to regulate dimerization by shifting the equilibrium between mutually exclusive structures containing either an U5–AUG or a U5–SL1 interaction[13,14,29]: 1G transcripts expose the DIS for dimerization and sequester the 5′ cap, whereas 3G variants conceal the DIS while exposing the cap to enhance translation[14]. In addition to the U5–AUG and U5–SL1 conformations, over 20 structural models of the HIV-1 genome have been proposed, suggesting that the 5′ UTR may dynamically adopt multiple conformational states[30,31]. It seems therefore likely that other structural forms of the HIV-1 genome exist to regulate genome dimerization, or other critical aspects of HIV-1 biology.

Sequences required for dimerization largely overlap with other conserved functional elements, such as those involved with genome packaging. Indeed, this genetic overlap between dimerization and packaging signals is a main reason why dimerization is considered to be a prerequisite for packaging, even though the precise molecular mechanism underlying this phenomenon is unclear. In this study, we disentangled genome dimerization from other steps of the viral life cycle and comprehensively mapped structure determinants of HIV-1 genome dimerization using a new high-throughput approach that we call functional analysis of RNA structure-sequencing (FARS-seq). Using FARS-seq, we found nucleotides throughout the HIV-1 5′ UTR influencing dimerization and identified distinct structural conformations in monomeric and dimeric RNA. The dimeric RNA folded into a 'canonical' structure of the 5′ UTR that displayed TAR, PolyA, primer binding site (PBS) and SL1–SL3 as stem loops, and contained a long-range U5–AUG interaction. In monomeric RNA, SL1 formed interactions with polyA and PBS. The PBS–SL1 interaction functionally couples primer binding with dimerization and the polyA–SL1 long-range interaction disrupts the major packaging motifs for Pr55Gag (refs. [32–34]).

[1]Helmholtz Institute for RNA-based Infection Research, Helmholtz Centre for Infection Research, Würzburg, Germany. [2]Faculty of Medicine, University of Würzburg, Würzburg, Germany. [3]P5 Systems Medicine of Infectious Disease, Robert Koch-Institute, Berlin, Germany. [4]These authors contributed equally: Anne-Sophie Gribling-Burrer, Patrick Bohn. ✉e-mail: redmond.smyth@helmholtz-hiri.de

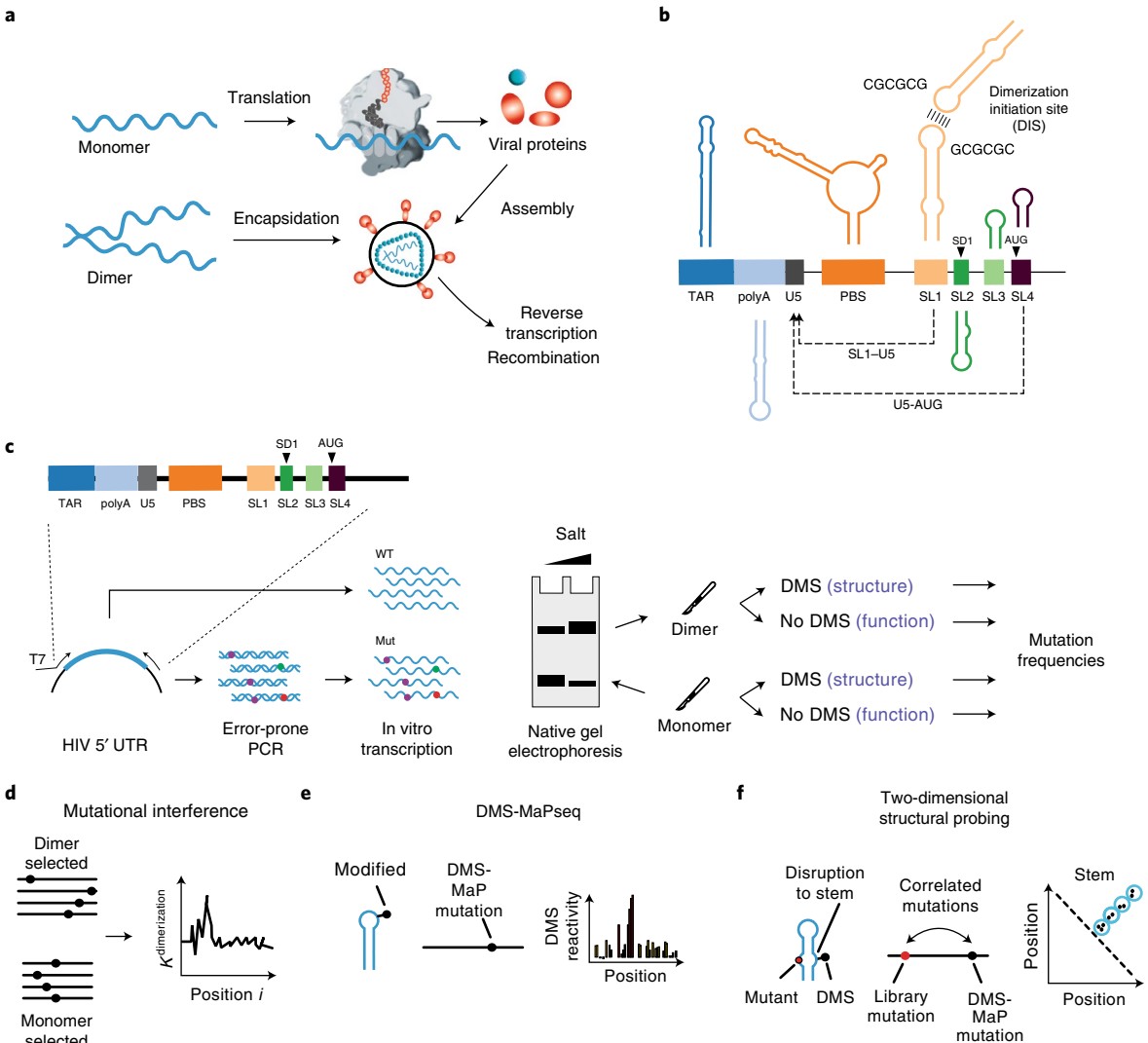

**Fig. 1 | Analysis of HIV-1 dimerization. a**, Dimerization is a key step in the HIV-1 life cycle. Monomeric RNA is thought to be preferentially translated, in contrast to dimeric RNA, which is a prerequisite for packaging into virions. Dimeric RNA helps maintain genome integrity through recombination. **b**, The HIV-1 5′ UTR is composed of distinct structural domains linked to different functions in the HIV-1 life cycle. TAR stands for transcription. PolyA stands for polyadenylation that is inactive in the 5′ UTR. U5 in unique 5 region or PBS, stands for annealing of the host tRNA for initiating reverse transcription. SL1–SL3 contain the packaging signal. SL2 contains the splice donor site. Dimerization occurs through a kissing loop interaction at a sequence in SL1. LDIs/ alternative folds involving LDIs, such as between SL1–U5 and U5–AUG may regulate dimerization. **c**, FARS-seq. Mutant RNA sequences are generated by mutagenic PCR and in vitro transcription. Mutant populations are physically separated into monomer and dimer fractions and probed with DMS or left untreated. Mutation frequencies are analyzed by next generation sequencing. **d**, Functional profiles are obtained by mutational interference. $K^{\text{dimer}}$ is a quantitative measure of dimerization based on the ratio of mutations in the dimer selected versus monomer selected population, corrected for mutations introduced during the library preparation and sequencing. **e**, Structural profiles are obtained by DMS that specifically reacts with unpaired A and C residues. DMS-MaPseq measures DMS reactivities as mutation rates in DMS treated versus untreated controls. **f**, Two-dimensional analysis identifies RNA stems through correlations between stem-disrupting mutations and mutations induced by DMS.

All in all, our data provide a mechanistic explanation for how RNA dimerization can be a prerequisite for packaging.

## Results

**Functional and structural analysis of RNA dimerization.** HIV-1 genome dimerization largely depends on the stem of SL1 and its GC-rich palindromic loop sequence. Nevertheless, evidence suggests that RNA sequences and structures outside SL1 also play a role[6,14,23,31,35–37]. We therefore devised a strategy to exhaustively survey the 5′ UTR for nucleotides influencing dimerization while at the same time generating information about RNA structure. We call this approach the FARS-seq (Fig. 1c). Fundamentally, FARS-seq uses mutational interference to generate complete, unbiased, quantitative profiles of RNA function at single nucleotide resolution[38,39] (Fig. 1d). These functional profiles are generated by physical separation of mutant RNA populations according to functionality followed by next generation sequencing and the analysis of mutation frequencies in the 'functional' and 'nonfunctional' populations. Simultaneously, structural profiles are obtained by treating the fractions with dimethyl sulfate (DMS), which is a chemical widely used for probing RNA structure[40,41]. DMS reacts with unpaired adenosine and cytosine bases to form adducts that can be read out as mutations on next generation sequencing machines[42–44]. Normally, DMS only provides information on whether a nucleotide is base paired, and not the identity of the base paring partner.

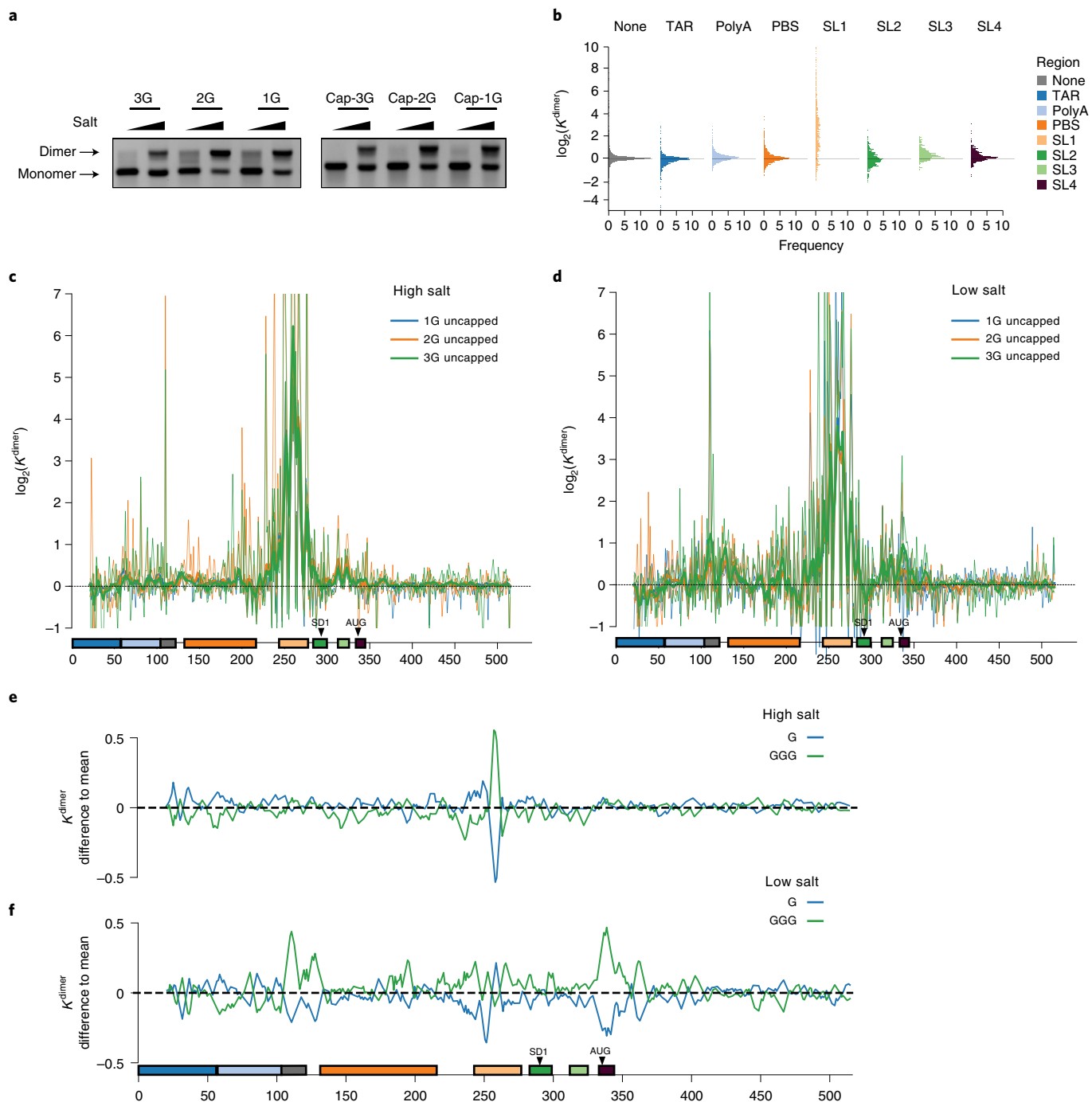

**Fig. 2 | Functional profiling of sequences involved in dimerization by mutational interference. a**, 1G, 2G and 3G capped and uncapped transcript variants migrate as distinct monomer and dimer bands on native agarose gels in both low and high salt buffers. Experiments were performed four times and representative data shown. **b–f**, $K^{dimer}$ is a relative measure of the effects of a mutation on dimerization, calculated as the ratio of mutation frequencies in the monomer versus the dimerized RNA, and corrected for errors introduced during library preparation and sequencing. **b**, The $\log_2(K^{dimer})$ values binned according to functional domain in the 5′ UTR: TAR, U5, PBS, SL1, SL2, SL3 and SL4. None refers to nucleotide positions that do not fall into any structural domain. **c,d**, Median $\log_2(K^{dimer})$ values for each genome position for all three uncapped transcript variants in high (**c**) and low (**d**) salt buffers. Lines represent median $\log_2(K^{dimer})$ values smoothed with a window size of 5 nt. P, probability. **e,f**, The $\log_2(K^{dimer})$ values of the 1G and 3G transcript variants compared to the mean of the 1G, 2G and 3G transcripts in high (**e**) and low (**f**) salt buffers.

However, when DMS modification is performed on mutational libraries it enables the direct detection of RNA stems (Fig. 1f)[45,46]. That is, when a mutation in the library occurs within a stem, it creates an unpaired nucleotide at the position facing the mutation. This newly unpaired residue becomes more accessible for DMS modification leading to correlated mutations in the sequencing

data. Thus, FARS-seq combines two different mutational read outs to experimentally couple RNA structural and functional information.

To physically separate mutants according to their effects on dimerization, we took advantage of the observation that RNA transcripts containing dimerization signals spontaneously associate

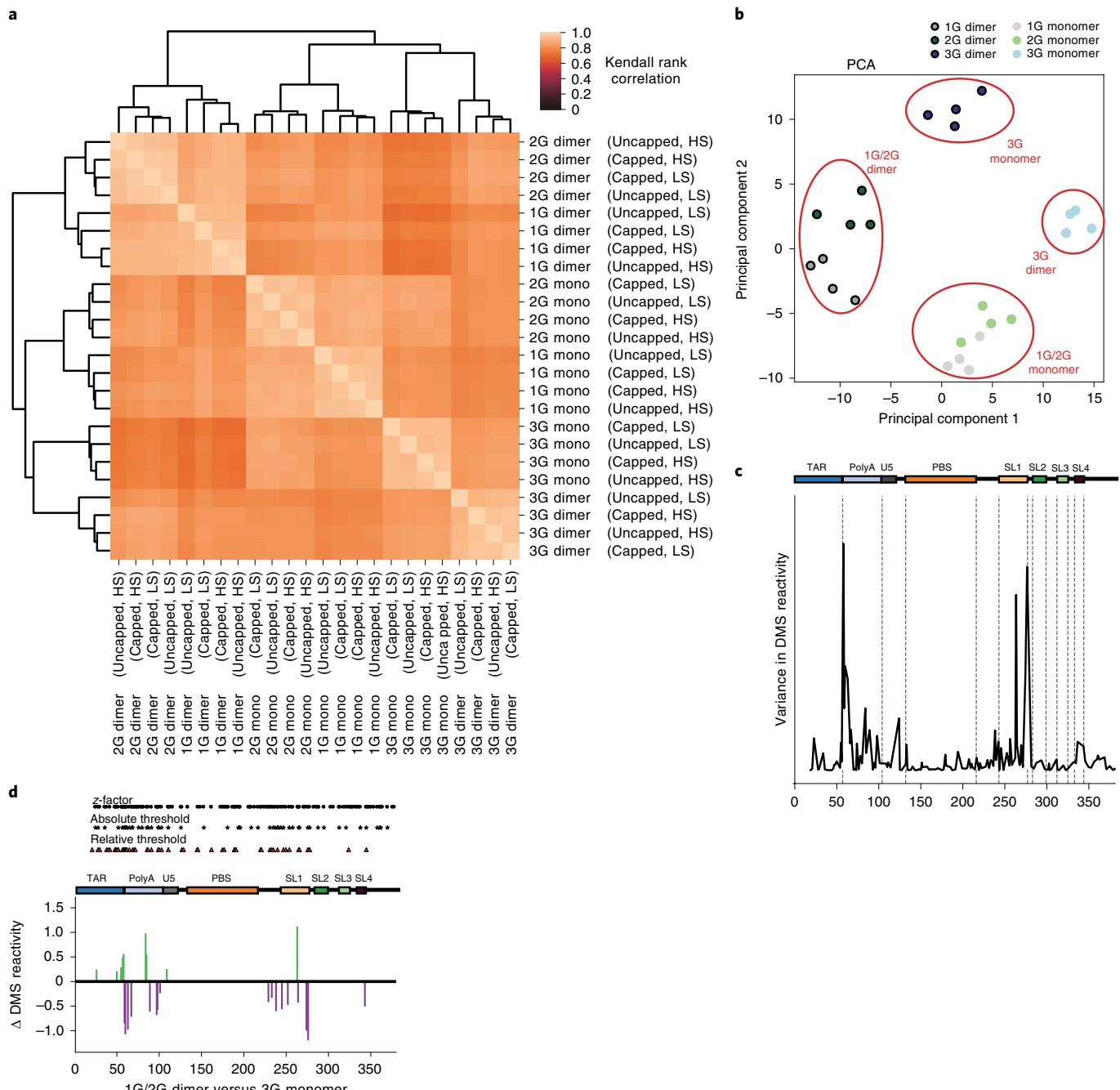

**Fig. 3 | Structural profiling identifies distinct structural conformations of the HIV-1 5′ UTR. a**, Clustering of Kendal rank correlations of DMS reactivities across all positions reveals structural relationships between monomer and dimer isolated populations from uncapped and capped transcript variants in high and low salt buffer. Relationships between sample DMS reactivities was determined by hierarchical clustering using the 'average' linking method. **b**, PCA of DMS reactivities identifies structural four structural classes of the HIV-1 5′ UTR. **c**, Variance in DMS reactivities across genome positions from all samples is enriched at the SL1 and TAR/polyA boundary. **d**, Statistical analysis of DMS reactivities in 1G/2G and 3G structural classes finds that significant differences in reactivities are mainly localized to polyA and SL1. A z-factor test identifies nucleotides where DMS reactivities change by >1.96 standard deviations of the DMS errors. An absolute difference threshold ensures that a minimum reactivity change of 0.2 is needed for the site to be considered biologically relevant. The relative threshold of 0.75-fold is used to remove false positives where DMS reactivities are high in both conditions such that a large change in reactivity is unlikely to affect RNA structure.

in vitro, producing a dimeric RNA species that can be physically separated from the monomeric species on native agarose gels (Fig. 2a). Similar gel-based assays have been instrumental in the discovery of dimerization motifs in HIV-1 (refs. [17–19]) and other viruses[47–49]. This setup also disentangles the effect of RNA structure on dimerization from other factors, such as the binding of protein or other cofactors.

To assess the effect of transcription start site heterogeneity on the dimerization properties of the HIV-1 genome we tested three transcript variants beginning with 1G, 2G or 3G (refs. [13,50]) (Fig. 2a). For each of these transcript variants, we also tested whether capping affected dimerization and assessed their dimerization properties under low salt and high salt buffers favoring monomerization

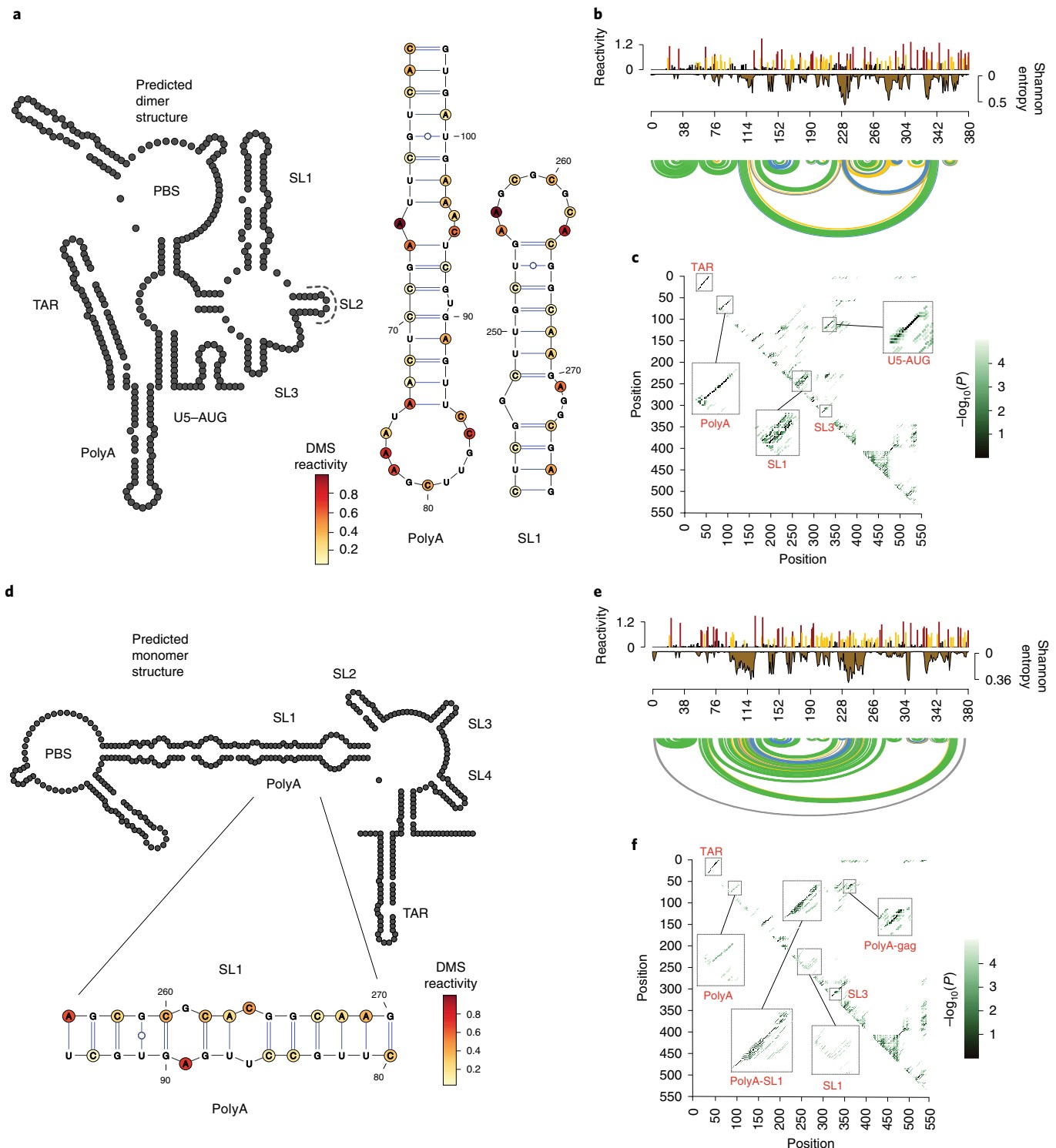

**Fig. 4 | Secondary structure model for 1G/2G dimer and 3G monomer populations. a,d,** Secondary structure model of dimer (**a**) and monomer class (**d**). Models were obtained using DMS reactivities as soft constraints for in silico folding in the Vienna RNA structure package. For the dimer structure, the U1sRNA binding site within SL2 is shown. Structures of polyA and SL1 stem loops and polyA–SL1 interaction are shown. DMS reactivities from dimer samples were mapped to A and C residues of the polyA and SL1 stem-loop structures. DMS reactivities from the monomer samples were mapped to A and C residues of the polyA–SL1 interaction. Red signifies highly reactive positions that are unpaired. Pale yellow signifies unreactive positions that are base paired. **b,e,** DMS reactivities and Shannon entropies for the 1G/2G dimer (**b**) and 3G monomer class (**e**). Arc plots show base-pairing probabilities (green 70–100%; blue 40–70%; yellow 10–40%; gray 5–10%). Gray bar in **e** signifies the polyA–SL1 interaction. **c,f,** Dot plots of RNA base-pairing probabilities for the 1G/2G dimer (**c**) and 3G monomer class (**f**), reveal alternative folding possibilities. RNA stems are shown along the diagonals.

and dimerization, respectively (Fig. 2a). After physical separation on a native gel, bands corresponding to monomeric and dimeric RNA populations were excised and either left untreated or soaked

in DMS. For the DMS sample (and its control), RNA was reverse transcribed in the presence of $Mn^{2+}$ to allow mutagenic bypass of the modified nucleotides by the reverse transcriptase[44,51]. In the

absence of DMS, mutation frequencies in the mutated and nonmutated control library were $5.4 \times 10^{-3}$ and $3.7 \times 10^{-4}$, respectively, and the mutational interference libraries with a signal to noise $D_m(i) > 2$ (Extended Data Fig. 1a, details of signal to noise in Supplementary Information). In the DMS treated samples, we saw an additional increase in mutation frequencies at the expected A and C residues indicating a successful modification of RNA (3.4- and 7.8-fold increase at C and A, respectively, Extended Data Fig. 1b).

RNA dimerization is regulated by the HIV-1 5′ UTR. We first asked which regions of the RNA were required for dimerization using mutational interference mapping (MIME) to calculate $K^{dimer}$ values for each nucleotide position. This metric is related to the ratio of mutation frequencies in the monomer versus dimer RNA. For computational analysis, however, these ratios are corrected for errors introduced during library preparation and sequencing (mechanistic derivation in the Supplementary Information). Thus, $K^{dimer}$ is a quantitative measure of the relative effect of each mutation on dimerization. Across all samples and conditions, median $\log_2(K^{dimer})$ values were heavily skewed toward positive values indicating that most mutations inhibited, rather than enhanced, dimerization indicating that the HIV-1 genome is highly optimized to dimerize as a key part of its life cycle (Fig. 2b). By segregating $K^{dimer}$ values by structural domain we found that most dimerization inhibiting mutations mapped to SL1 (Fig. 2b). Although less prominent than SL1, many other domains exhibited skewed distributions. Mutations to SL3, SL4 and polyA were biased toward inhibiting dimerization whereas mutations to TAR and SL2 preferentially enhanced dimerization. In contrast, mutations to the inter-domain regions were largely neutral with a narrow distribution centered around zero (Fig. 2b).

We next plotted median $\log_2(K^{dimer})$ values at each nucleotide position for capped and uncapped transcript variants measured under the two buffer conditions (Fig. 2c,d and Extended Data Fig. 2). All conditions exhibited a very large peak that localized to SL1, as well as a smaller double peak mapping to SL3 (Fig. 2c,d). In high salt buffer, most mutations inhibited dimerization, whereas under low salt conditions it was possible to distinguish additional dimerization enhancing or inhibiting regions (Fig. 2c,d). Notably, sequences surrounding the AUG start codon and mapping to U5 were both required for dimerization in low salt buffer, suggestive of a functionally important U5–AUG interaction (Fig. 2d). A double peak also emerged within the region 122–141 in low salt buffer (Fig. 2d). This region contains the primer activation sequence (PAS), which hints that structural changes in the PBS domain may regulate RNA dimerization[52,53]. Conversely, we found regions within TAR, polyA, PBS and SL2 that enhanced dimerization on mutation (Fig. 2d). The strongest of these regions mapped to the 3′ end of PBS and SL2. Taken together, these data reinforce the key importance of SL1 for genome dimerization, but also reveal sequences outside SL1 participate in the dimerization process.

1G and 3G RNAs have different dimerization properties. Because the HIV-1 transcription start site has been reported to alter the structure of the HIV-1 5′ UTR, we next tested which RNA sequences were important for dimerization within the 1G, 2G and 3G uncapped variants (Fig. 2e,f and Extended Data Fig. 3). We did this by plotting the absolute difference between the median $\log_2(K^{dimer})$ values of each variant to the mean values of the three transcripts. In high salt buffer, most positions were unchanged in the 1G, 2G and 3G variants (less than $\Delta 0.25 \log_2(K^{dimer})$ variant − mean) (Fig. 2e and Extended Data Fig. 3). The only exception was the nucleotides mapping to the SL1, which were functionally more important in the 3G variant, and less important in the 1G variant. On performing a similar analysis for the low salt condition, distinct functional profiles for the 1G and 3G transcript variants emerged, with divergence across regions compared to the mean of the three transcripts (Fig. 2f and Extended Data Fig. 3). The 3G variant had increased dependence on

a region spanning the U5 and PAS (nucleotides (nts) 105–117 and nts 125–131) and sequences surrounding the AUG start site (nts 335–344). Increased dependencies of smaller magnitudes were also observed in the transfer RNA PBS (nts 182–200), the anti-PAS (nts 217–223), regions flanking SL1 such as the CU rich motif (nts 228–247), a region in SL2 (nts 299–300) and a G rich region downstream of the AUG start codon (nts 360–366). We note that the regions in TAR, PBS and SL2 that enhanced dimerization on mutation in low salt conditions behaved identically in 1G, 2G and 3G variants, meaning that they affect dimerization in a way that is unrelated to transcription start site selection. We also remarked that the 1G and 2G transcripts variants behaved similar in both buffer conditions with a reduced dependency on regions external to SL1 for dimerization (Extended Data Fig. 3). Our interpretation is that the 1G and 2G transcripts readily fold into a dimer promoting conformation, whereas the 3G variant has a reduced capacity to dimerize. Capped and uncapped transcripts had near identical functional profiles (Extended Data Fig. 2). The only region that differed in capped and uncapped transcripts mapped to polyA, providing indirect evidence of a functional interaction between the 5′ cap structure and polyA (Extended Data Fig. 3).

Distinct structural signals in monomeric and dimeric RNA. So far, the analysis of the functional profiles demonstrate that sequences involved in genome dimerization map to distinct regions of the HIV-1 5′ UTR. These sequences may fold into RNA structures that are necessary for genome dimerization itself, or indirectly regulate genome dimerization by altering folding pathways. We therefore next determined RNA structural motifs present in monomers and dimers by analyzing the DMS reactivities of the FAR-seq data.

As before, we analyzed capped and uncapped 1G, 2G, 3G transcript variants in both monomer and dimer buffers. Correlations between DMS reactivities at each position among all conditions were very high (Fig. 3a; Kendall rank correlation coefficients, mean 0.84, minimum 0.70, maximum 1.0) suggesting that a large portion of the 5′ UTR was folded into a similar conformation under all conditions. Nevertheless, hierarchical clustering of the DMS reactivities revealed a clear structural distinction between monomer and dimer, as well as between the 1G/2G and 3G transcript variants (Fig. 3a). In contrast to the functional profiling, where buffer conditions had a very large effect on the functional profiles, structural information obtained under both conditions were highly correlated (correlation coefficients; low salt 0.84, high salt 0.85), as were uncapped and capped RNAs (correlation coefficients; capped 0.85, uncapped 0.84). The first branchpoint separated 1G/2G dimer structures from the 1G/2G monomer and 3G structures. Subsequent branching grouped 1G/2G monomer structures away from the 3G structures. Finally, 3G structures separated into monomer and dimer subclusters. These four structural groupings were also supported by principal component analysis (PCA) of DMS reactivities, which separated monomer from dimer, and 3G variants from 1G/2G variants (Fig. 3b). Guided by the PCA and hierarchical clustering, we pooled DMS reactivity data into four structural groups: 3G dimer, 3G monomer, 1G/2G monomer and 1G/2G dimer. Across all samples, variance in DMS reactivities localized mainly to polyA and SL1 (Fig. 3c). To further explore this, we used a statistical approach to compare DMS reactivities in the 1G/2G dimer cluster with the 3G monomer cluster as these were the most structurally divergent samples (correlation coefficient 0.740) (Fig. 3d). Between these clusters, we found statistically significant changes in reactivity that again remained localized to polyA and SL1 (Fig. 3d).

To obtain information on RNA secondary structure differences between these structural classes we used pooled DMS reactivities as soft constraints to guide in silico RNA folding[54,55] (Fig. 4 and Extended Data Fig. 4). For the 1G/2G dimer class, we obtained an RNA structure that closely resembled the 'canonical' HIV-1 5′ UTR (Fig. 4a and Extended Data Fig. 5). This structure contains the

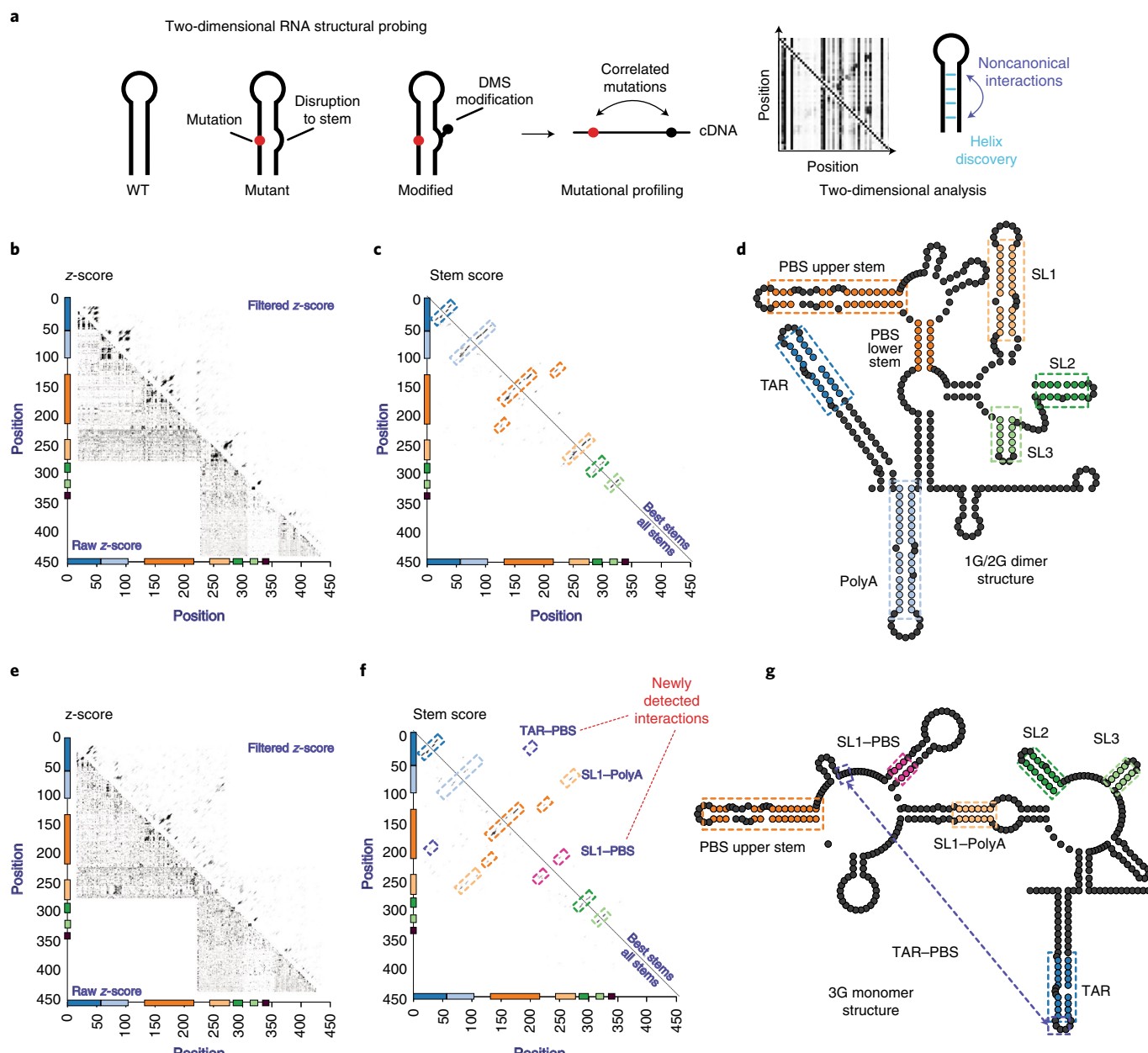

**Fig. 5 | Two-dimensional mapping of RNA structure in 1G/2G dimer and 3G monomer populations. a,** Mutations disrupting RNA stems lead to increases in DMS reactivity at positions opposite the mutation. Positions of DMS modification are read out as mutations leading to correlated mutations at pairs of nucleotides involved in RNA structure. RNA secondary structures (blue circles) are identified along the diagonals. Punctate signals (purple circles) can signify noncanonical or tertiary interactions. **b,e,** The z-score analysis of mutation frequencies from 1G/2G dimer (**b**) and 3G monomer populations (**e**). Raw z-scores (lower diagonal) reveal pairs of positions enriched with mutations. Filtered z-scores (upper diagonal) enhance stem signals by applying a convolution filter and signal threshold. Insets are zooms of the filtered z-scores for the polyA–SL1, PBS–SL1 and SL1 stems. **c,f,** Stem detection in 1G/2G dimer (**c**) and 3G monomer populations (**f**). All stems (lower diagonal) reveal all possible stems of minimum length 3 by applying a filter for Watson–Crick and Wobble base pairs to the filtered z-score. Best stem (upper diagonal) selects the best nonconflicting stems by removing conflicting stems based on filtered z-score. Colored boxes represent regions that are highlighted in enhance RNA secondary structure models. **d,g,** Enhanced RNA secondary structure models of 1G/2G dimer (**d**) and 3G monomer populations (**g**). Colored base pairings were detected in multidimensional mapping and used as hard constraints before in RNA secondary structure prediction. Dark blue is TAR. Light blue is polyA. Orange is PBS. Mustard is SL1. Dark green is SL2. Light green is SL3. Red represents the polyA–SL1 interactions, pink shows the new SL1–PBS interaction and purple the TAR–PBS interaction.

TAR, PolyA, PBS, SL1 and SL3 stem loops, as well as the AUG–U5 interaction. The basal portion of SL1 folded into an extended form containing unpaired purines that are important for genome packaging[32,56]. SL2, which can fold into alternative stem-loop structures, folded as an imperfect stem loop that exposes part of the U1snRNA binding site within the loop, and SL3 folded into its canonical short

stem-loop structure. We then assessed the robustness of this prediction by computing Shannon entropies of base-pairing probabilities at each position in the 5′ UTR (Fig. 4b and Extended Data Fig. 4). Low entropy values throughout the 5′ UTR indicated high confidence in the prediction and a well-ordered structure with only some ambiguity in the base pairing at the basal portion of SL1. This was

confirmed by dot plots of base-pairing probabilities and a bootstrapping analysis showing high confidence stem-loop structures for the TAR, PolyA, PBS, SL1 and SL3 stem loops, as well as the AUG–U5 interaction (Fig. 4c and Extended Data Fig. 4).

We next analyzed the structure of the 3G monomer sample, finding that it was dramatically reorganized (Fig. 4d). The most striking changes were seen in the polyA, AUG–U5 and SL1. PolyA and SL1 no longer folded into their canonical stem loops. Instead, these stem loops were reorganized into a long-distance interaction (LDI), with the GCGCGC palindromic loop of SL1 base pairing with the apical portion of the polyA stem. The AUG–U5 interaction was also no longer present: U5 now base paired with the 5′ stem of SL1, and the AUG containing region fold into a stem-loop structure, also referred to as SL4. Finally, we observed a new LDI between polyA and a region within the Gag coding sequence (nts 358–367). The SL1–polyA reorganization was well supported by the DMS reactivity changes (Fig. 4 and Extended Data Fig. 5). In particular, the unpaired adenosine 263 A in the SL1 loop, which was highly reactive in the dimer structure, became unreactive in the monomer due to base pairing with U87. Similarly, nucleotides C84 and C85 in polyA, which were reactive in the dimer structure, became unreactive in the monomer due to base pairing with 265G and 266G in the SL1 stem. Finally, A89 in the stem of polyA, which was unreactive in the dimer structure, became unpaired in monomer structure and reactive to DMS. Shannon entropies, base paring and bootstrapping probabilities at the predicted polyA–SL1 interaction indicated some uncertainty in the prediction, especially within U5 and the 5′ portion of SL1 (Fig. 4f and Extended Data Fig. 4). Despite the reorganization of polyA and SL1, a large proportion of the 5′ UTR folded identically in 1G/2G dimer and 3G monomeric populations, with PBS, SL2 and SL3 unchanged. TAR was present in all predictions, but in the 3G monomer the first nucleotides in the base of TAR became single stranded and potentially more available for the translation machinery.

The 3G dimer and 1G/2G monomer populations folded into the population folded into the canonical 5′ UTR structure and the alternative polyA–SL1 containing structure, respectively (Extended Data Fig. 4). However, these two structural classes showed increased Shannon entropies in polyA, U5, SL1 and the Gag coding sequence when compared to the 1/2G dimer and 3G monomer structures. Thus, 3G dimer and 1/2G monomer populations are structurally less uniform, even though we selected for pure dimer and monomer structures in the native gels. The most likely explanation is that these structures partially return to equilibrium after isolation, probably during the probing reaction at 37 °C.

Altogether, these data support a new structural rearrangement of the HIV 5′ UTR leading to extensive base pairing between SL1 and the polyA-U5 region. This monomeric rearrangement appears to be favored in the 3G populations, whereas the 1G/2G population tend toward the dimer structure.

**Refinement of monomer and dimer structures.** The incorporation of information from RNA structural probing experiments improves the accuracy of RNA structure predictions, but structural elements can still be incorrectly predicted because data from chemical probing experiments typically provide information on whether a nucleotide is base paired or not, but not its base-pairing partner[57,58]. FARS-seq enables a more powerful model-free approach to RNA structure determination by exploiting information in the mutation library to identify RNA helices directly (Fig. 5a). When mutating a nucleotide in a stem structure, the base-pairing partner, now unpaired, becomes more reactive to the chemical probe leading to correlated mutations in the sequencing data[46,58]. These two-dimensional data can directly detect RNA helices (along the diagonal) as well as noncanonical and tertiary interactions that are otherwise impossible to predict from classical one-dimensional RNA structural probing experiments.

Signals for RNA helices were visible in the raw mutational and z-score normalized data along the diagonals (Fig. 5b,e). These signals were refined by applying convolution and threshold filters to enhance stems as well as tertiary interactions (Fig. 5b,e). Finally, high confidence stems were highlighted by applying a helix filter and algorithm to select the 'best' nonconflicting stems with the highest score (Fig. 5c,f). Stem signals corresponding to SL1 were systematically present in dimer selected samples and absent in monomer selected samples (Extended Data Fig. 6 and Supplementary Information). In the 1G/2G dimer sample, both SL1 and polyA stem signals were observed. In the 3G monomer, polyA and SL1 stems were replaced with a signal matching the long-distance SL1–polyA interaction (compare Fig. 5b,c with 5e,f). In the 3G monomer, we detected an additional new interaction between the PBS loop and SL1, as well as a weaker signal between TAR and PBS, both of which were supported by a bootstrapping analysis (Fig. 4e,f and Extended Data Fig. 7). In the previous structural prediction, these regions in PBS and SL1 had high Shannon entropies and were poorly resolved (Fig. 4). In the 1G/2G dimer sample, both SL1 and polyA stem signals were observed. In the 3G monomer, polyA and SL1 stems were replaced with a signal matching the long distance SL1-polyA interaction (compare Fig. 5b,c with 5e,f). In the 3G monomer we detected an additional novel interaction between the PBS loop and SL1, as well as a weaker signal between TAR and PBS, both of which were supported by a bootstrapping analysis (Fig. 4e,f and Extended Data Fig. 7). In the previous structural prediction these regions in PBS and SL1 had high Shannon entropies and were poorly resolved (Fig. 4).

Uniquely in the 1G/2G structures, the TAR and polyA stem signals in the filtered z-scores were accompanied by punctate signals characteristic of tertiary contacts, alternative folds or noncanonical base pairings (Extended Data Fig. 6). Because these contacts were consistently present in the 1G/2G samples and missing from the 3G samples, we speculate that they help to stabilize the 5′ end of the HIV-1 transcript to inhibit the translation of 1G/2G transcripts (Extended Data Fig. 6 and Supplementary Information)[13]. Additionally, in the 1G/2G monomer, the mutually exclusive polyA stem and the polyA–SL1 interaction were both observed, strengthening the idea that 1G/2G samples are preferentially dimeric and that some interconversion occurs even when monomers are isolated (Extended Data Fig. 6).

To obtain enhanced structural models of the dimer and monomer structures we focused on the 1G/2G dimer and 3G monomer samples as these were the most structurally uniform. Here, the best

**Fig. 6 | Structure/function analysis of HIV-1 dimerization. a,c**, Single nucleotide resolution functional profiling data pooled from six low salt samples mapped the dimer (**a**) and monomer (**c**) structures expressed as $\log_2(K^{dimer})$ values. Each individual mutant shown as one of three circle in the order A,C,G,U clockwise from upper position (excluding the WT base). Validation of structural models on 3G RNA by point mutagenesis followed by native agarose gel electrophoresis in two different buffer conditions. Experiments were performed at least twice, representative data shown. Red circles show mutations inhibiting dimerization, and blue circles show mutations enhancing dimerization. $\log_2(K^{dimer})$ values above 2 are capped. **b**, Functional profiling data mapped to different structural models of SL1 containing mutually exclusive internal loop configurations. The two-internal loop (2IL), 3IL and the 3WJ are mutually exclusive models of SL1 structure based on chemical probing or biophysical measurements. Green arrows show mutations that improve dimerization by closing or reducing the size of internal loops, providing evidence that SL1 is metastable and that alternative SL1 conformations can form and dimerize. Red arrows show mutations that have complex effects on dimerization because they affect the new PBS–SL1 interaction.

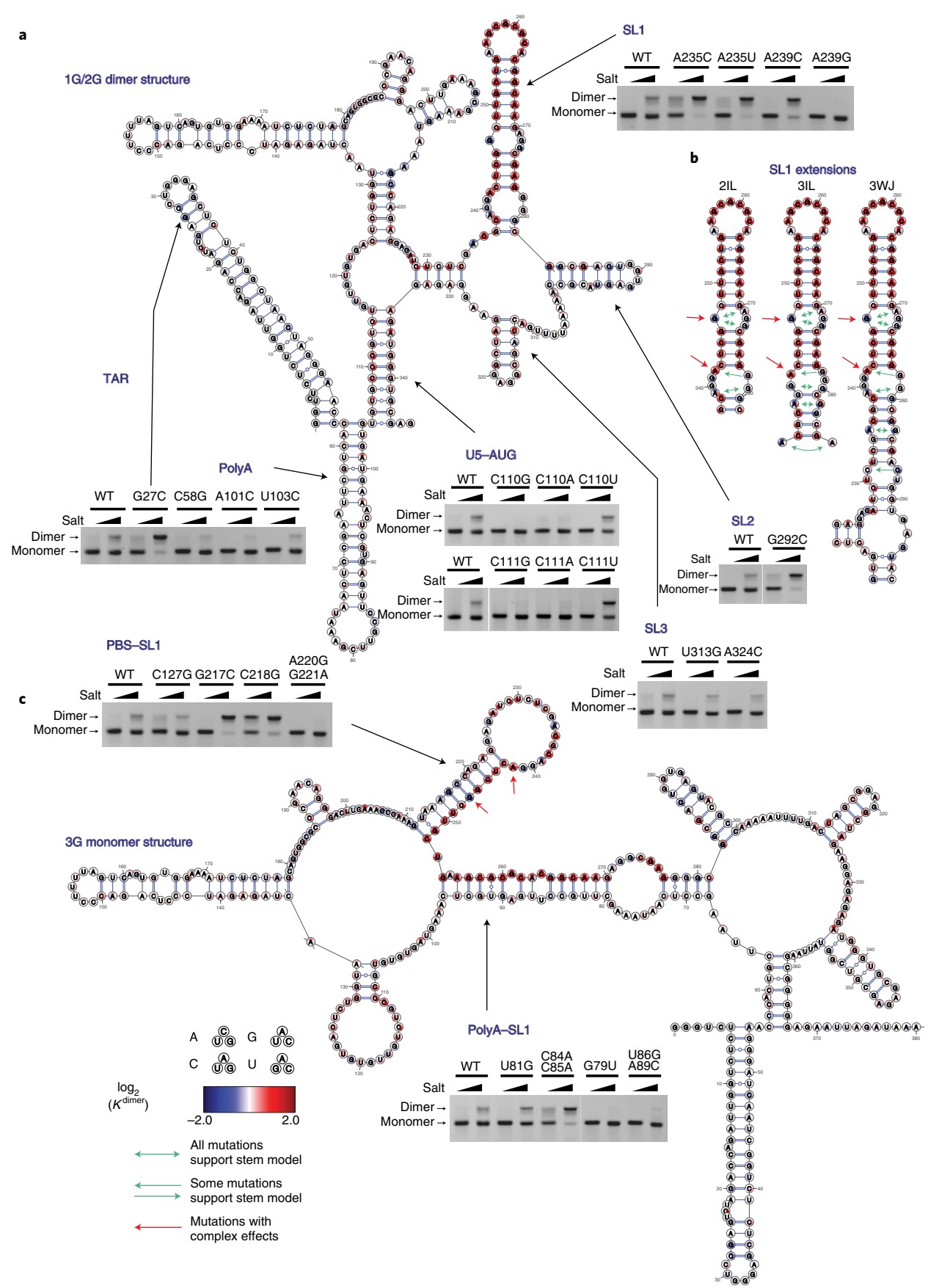

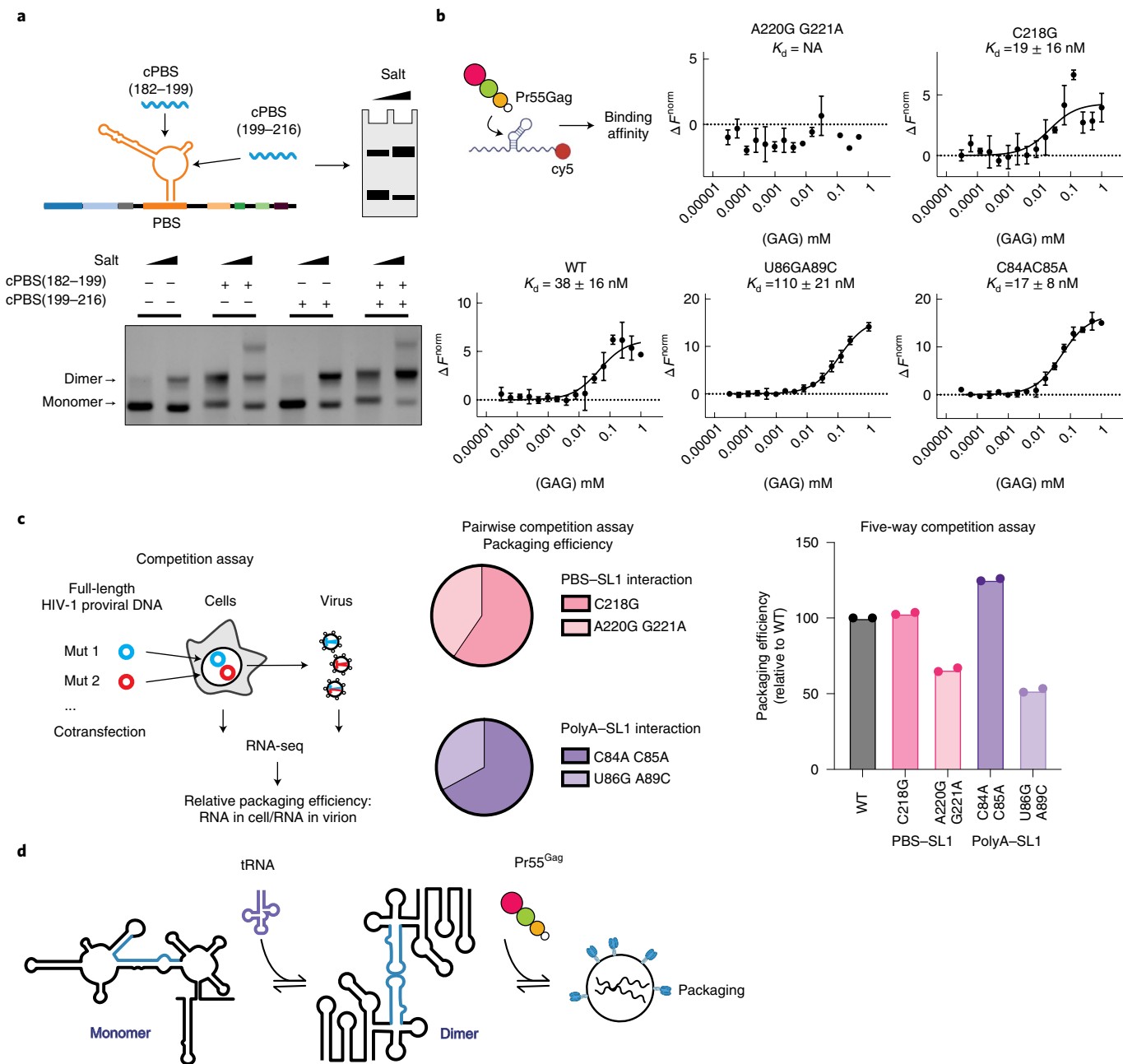

**Fig. 7 | PBS and polyA regulate HIV-1 dimerization, Pr55<sup>Gag</sup> binding and genome packaging. a**, PBS targeting oligos can trigger dimerization of a 3G RNA. cPBS(182-199) disrupts the TAR–PBS interaction leading to the formation of a higher order RNA structure. cPBS(199-216) disrupts the PAS-anti-PAS stem and enhances dimerization. The effects of both oligos are additive. Experiments were performed in duplicate, with representative data shown. **b**, Mutations targeting the polyA–SL1 and PBS–SL1 interaction affect Pr55<sup>Gag</sup> binding as measured by MST. Data from three independently experiments were analyzed. Data are represented as mean with error bars showing standard deviations. **c**, Competition assays to measure the relative effects of mutations on genome packaging into virions. Two-way competition assays show that dimer promoting mutations C218G and C84A–C85A are enhanced in genome packaging compared to monomer promoting mutations A220G-G221A and U86G-A89C. Five-way competition assays between WT HIV-1 and mutants show that dimer promoting mutants are packaged similar or better than WT, whereas monomer promoting mutants are packaged less efficiently than WT. Experiments were performed in duplicate. **d**, Model showing how the binding of host factors can regulate viral replication, in part, through remodeling RNA structure.

stems obtained by multidimensional chemical probing were used as additional hard constraints in RNA structure prediction (Fig. 5d,g). The enhanced 1G/2G dimer structure was nearly identical to that obtained without hard constraints, and contained the TAR, PolyA, PBS, SL1 and SL3 stem loops, as well as the AUG–U5 interaction as previously predicted (Fig. 5d). The enhanced 3G monomer structure contained TAR, polyA–SL1 interaction, SL2, SL3 SL4 and polyA–Gag interaction, as before, but now included a stem-loop structure

due to base paring between PBS and SL1 (Fig. 5g). A TAR–PBS pseudoknot interaction was added post hoc, as it was selected by the best stem algorithm and supported by a bootstrapping analysis, although we note that the 2*d* stem score was relatively weak. All in all, multidimensional chemical probing not only provided direct experimental evidence that 1G/2G dimer and 3G monomer fractions are structurally distinct, but also identified structural features that could not be predicted by classical RNA structural probing experiments.

SL1 stability is a key element for genome dimerization. One of the strengths of FARS-seq is the coupling of RNA structural and functional information at single nucleotide resolution. We therefore mapped the $K^{dimer}$ values onto the dimer and monomer structures. In both buffers, the median mutations with the strongest effects mapped to the apical portion of SL1, with mutations to the palindromic loop sequence revealed to be the most destabilizing for dimerization, in agreement with their crucial role in the kissing loop interaction (Fig. 6a, and Supplementary Data Table 7). The unpaired adenosine residues flanking the loop sequence were less important for dimerization than the palindromic sequences, in keeping with the observation that they can be individually mutated without disrupting dimerization[59]. Mutations to the stem of SL1 also strongly inhibited dimerization, with apical stem mutations generally having a stronger effect on dimerization compared to the basal stem mutants ($\log_2(K^{dimer})$ values 0.61–6.85 versus 0.31–3.49) (Fig. 6a). Mutations at several positions within the SL1 internal loop (G247, A271, G272, G273) strongly enhanced dimerization on mutation (Fig. 6a and Supplementary Data Table 7). Dimer enhancing mutations at these positions presumably stabilize SL1 by closing or reducing the size of the internal loop, strongly indicating that SL1 stability is a critical parameter for dimerization.

While two-dimensional structural probing identified SL1 as a short stem loop with an apical and basal stem separated by an internal loop (nucleotides 243–277), our data nevertheless reveal structural plasticity in SL1. This realization comes from mapping the functional data to different extended forms of SL1 that have been proposed in the literature: a two-internal loop model, a three-internal loop model (3IL) and three-way junction (3WJ) model (Fig. 6b). Even though these models have mutually exclusive internal loop configurations, mutations that closed or reduced the size of SL1 internal loops were invariably dimerization enhancing (Fig. 6b, green arrows). For example, A235C, A235U or G281U strongly enhanced dimerization by converting the A235-G281 internal loop into a base pair in the 3WJ model, even though these mutations would have no effect on SL1 stability on the other structural models (Fig. 6b, green arrows). Similarly, G282C and G239C would close the internal loop in the 3IL model explaining their dimerization enhancing properties (Fig. 6b, green arrows). To confirm the structural plasticity of SL1, we performed in solution DMS-MaPseq analysis of mutants A235C and A239C and showed that they reconfigured the SL1 stem, as predicted (Extended Data Fig. 8). Mutations A242C or A242U reduced the size of an SL1 internal loop in all models but nevertheless disrupted dimerization (Fig. 6b,c, red arrows). These functional effects are explained by the fact that A242C or A242U extend the PBS–SL1 interaction to stabilize the monomer structure. Thus, the core dimerization structure in SL1 comprises an apical 7-nt stem and a basal 4-nt stem separated by an internal loop that can be further stabilized by metastable stem extensions or disrupted by a base-pairing interaction with PBS.

**Inter-domain interactions regulate dimerization.** Outside SL1, we found several structural domains and inter-domain interactions that affected dimerization (Fig. 6). Our data support a role for the AUG–U5 interaction in positively regulating dimerization, as conversion of GU base pairs at U107–G342, G108–U341, G112–U337 to either AU or GC base pairs consistently enhanced dimerization, whereas mutations disrupting the interaction were inhibitory (Fig. 6a). SL3 stem mutations weakly inhibited dimerization, most likely because disruption of SL3 would induce misfolding of the RNA (Fig. 6a). Finally, mutations to SL2 were generally dimerization enhancing and these types of mutation were especially evident in the 3′ SL2 stem (Fig. 6a).

We also validated the new short- and long-range interactions between polyA–SL1 and PBS–SL1. Mutations to the base of polyA generally inhibited dimerization, indicating that destabilizing the polyA stem favors the formation of the polyA–SL1 interaction (Fig. 6a).

On the other hand, mutations to the upper portion of polyA enhanced dimerization by disrupting the polyA–SL1 base pairing (Fig. 6c). In the same vein, we found stretches of nucleotides in PBS that strongly enhanced dimerization on mutation (Fig. 6c). Functional profiles in the lower PBS stem were particularly interesting as this stem structure is universally found in contemporary models of the HIV-1 5′ UTR and contains the PAS known to be important for efficient reverse transcription[60]. We found that mutation of two nucleotides G217 and C218 in the lower PBS stem very strongly enhanced dimerization, even though mutations to this stem were generally inhibitory (Fig. 6a). This can be mechanistically explained because mutation of these nucleotides disrupted a new base pairing between PBS and SL1 that stabilizes the monomer structure.

Because these results indicated a functional interaction between primer tRNA binding and dimerization, we next assessed whether disruption of the PBS with tRNA mimic oligos affected dimerization. cPBS$_{182-199}$ annealed to the loop region disrupted the putative TAR–PBS interaction, whereas cPBS$_{199-216}$ disrupted the new PBS–SL1 stem loop (Fig. 7a). Both oligos enhanced dimerization confirming a functional interaction between PBS and dimerization. Annealing the cPBS$_{182-199}$ oligo also led to the formation of a higher, presumably tetrameric molecular species. The TAR apical loop contains a ten-nucleotide palindromic sequence that has been proposed to dimerize by a TAR–TAR kissing interaction analogous to the one used by SL1 (ref. [26]). We therefore postulate that cPBS$_{182-199}$ disrupts the TAR–PBS interaction detected by multidimensional structural probing, allowing TAR to dimerize independently of SL1.

Finally, since genome dimerization is thought to be a prerequisite for genome packaging, we selected mutations in adjacent nucleotides with divergent effects on dimerization and measured their effects on Pr55$^{Gag}$ binding by microscale thermophoresis (MST) (Fig. 7b). None of these mutations resided in the HIV-1 packaging domain (SL1–SL3). In PBS, C218G, which strongly enhanced dimerization had higher affinity ($K_d$ 19 nM) to Pr55$^{Gag}$ compared with wild-type (WT) RNA ($K_d$ 38 nM). In contrast, PBS A220G–G221A, which was unable to dimerize, did not bind Pr55$^{Gag}$ at any of the concentrations tested ($K_d$, NA). In polyA, dimerization enhancing mutation C84A–C85C bound Pr55$^{Gag}$ with higher affinity (17 nM) than WT, whereas dimerization disrupting mutation U86G–A89C bound Pr55$^{Gag}$ with lower affinity than WT (110 nM). By performing in solution DMS-MaPseq analysis in vitro, we established that mutations in polyA–SL1 and PBS–SL1 alter ensemble reactivities toward the profiles seen in the isolated monomer and dimer. (Extended Data Fig. 9). Thus, the four mutants not only alter the monomer–dimer equilibrium but produce the predicted structural changes that affect Pr55$^{Gag}$ binding. We also introduced these mutations into the full-length HIV-1 genome and assessed their effects on packaging efficiency in competition assays (Fig. 7c). In PBS, dimer promoting mutant C218G was enriched 1.5-fold in virions compared to the monomer promoting mutant A220G–G221A. In polyA, dimer promoting mutant C84A–C85A was enriched twofold in virions compared to the monomer promoting mutant U86G–A89C. In a five-way competition assay between WT HIV-1 and the mutants, dimer promoting mutants C218G and C84A–C85A were packaged equivalently or better than WT. Conversely, monomer promoting mutants U86G–A89C and A220G–G221A were deficient in packaging compared to WT. Last, we performed in solution DMS-MaPseq analysis of these four mutants directly in cells. Despite complex reactivity changes induced by cellular ligands, dimer promoting mutants folded into structures containing SL1, whereas monomer promoting mutants folded into structures where SL1 was hidden through long- and short-range interactions with polyA and PBS (Extended Data Fig. 10). Thus, we conclude that the regulatory mechanism we identified in vitro also takes place in cells.

Taken together, our results provide a clear mechanistic explanation for the link between dimerization, Pr55[Gag] binding and packaging. We also show how changes to the PBS functionally link the tRNA binding region to packaging (Fig. 7d).

## Discussion

Accumulating evidence emphasizes dimerization as a key step in HIV-1 life cycle that is regulated, at least in part, through the folding of the HIV-1 genomic RNA[5,11–16,27,28,61,62]. Here, we resolved the structure of the monomeric and dimeric RNAs using a new approach that integrates information from RNA structural probing with high-throughput functional profiling. This experimental strategy has advantages over other chemical probing methods that make ensemble measurements over all possible conformations of the RNA in solution. Such ensemble measurements, unless cautiously interpreted, can lead to false predictions when mapped to a single structure. We overcome this problem by physically isolating RNA structural conformations with respect to their function, akin to in-gel SHAPE software, which was first developed to resolve structural differences between monomeric and dimeric species of the HIV-1 5′ UTR[28]. Moreover, by performing chemical probing on mutagenic libraries we obtain model-free information on RNA helices in the same way as 'mutate and map'[58] or 'M2-seq'[46]. Finally, our approach enables a deep understanding of how RNA structures relate to RNA function by uniquely coupling structural information with a functional read out.

Altogether, our data recognize a core dimerization domain of SL1 composed of a 7 base-pair apical stem and 4 base-pair basal stem separated by an internal loop. This core dimerization domain is present in most structural models of SL1, but there is significant disagreement on whether SL1 is further extended[63–66]. In some structures, extensions to SL1 even lead to the complete disruption of SL2 (ref. [12]). Here, we found no direct evidence that SL1 is in an extended form in dimeric RNA and consistently observe signals for SL2 as a short imperfect stem containing a bulged adenosine. Nevertheless, functional profiling provides strong evidence that mutually exclusive extended forms of SL1 can be readily generated, either directly through stabilizing mutations or indirectly by destabilizing SL2. The fact that single point mutations could have such dramatic effects on dimerization provides evidence that the 5′ UTR is dynamic and metastable. In the context of viral infection, this is noteworthy because it provides a mechanism to regulate dimerization through the binding of viral or cellular factors to the genome (Fig. 7c).

The metastable nature of SL1 was revealed in monomeric RNA. In contrast to SL3, which was present in both monomer and dimer structures, SL1 was destructured in monomeric RNA. Instead of a stem loop, SL1 was reorganized into a short-range interaction with PBS and a long range interaction with polyA. These results are in agreement with the prevalent idea that RNA conformational switches regulate HIV-1 replication[33,67]. The dimer and monomer structural conformations we present here are reminiscent of the branched multiple hairpin and LDI models that were proposed as alternative structures that would regulate the dimerization, packaging, splicing and translation of the HIV-1 genome[5,15,16,27]. The branched multiple hairpin exposes the TAR, polyA, PBS, SL1, SL2 and SL3 structures, and contains the U5–AUG interaction. The LDI model includes the interaction between polyA and SL1, but also includes additional rearrangements that we did not observe, such as an extension of SL3 and a disruption of SL2. Moreover, the LDI model does not include the new PBS–SL1 interaction. Nevertheless, certain mutants designed to alter the LDI or BMH equilibrium are directly applicable to our structural model. In particular, mutations destabilizing the polyA stem inhibit dimerization and packaging[15,27], whereas mutations disrupting the polyA–SL1 interaction enhanced dimerization[16]. These data are in agreement with our results showing that polyA–SL1 regulates not only dimerization, but also genome packaging. Recent work has identified the primary Pr55[Gag] binding site for HIV-1 as SL1 (refs. [32,38,39,68]) with polyA providing an additional packaging signal in cells[39,69]. The fact that SL1 and polyA are completely disrupted in the monomer population provides a mechanistic explanation for the long-postulated link between dimerization and packaging.

Recently, the structure of the 3G capped transcript was solved by NMR revealing the disruption of the polyA stem in 3G transcripts and the formation of a long-range interaction between SL1 and U5 (ref. [14]). Thus, our results agree that 3G transcripts are preferentially monomeric, yet disagree with precise structural details, in particular the base-pairing partner of SL1. One way to reconcile these data is that the NMR structure was obtained with the Mal isolate, in contrast to the NL43 isolate used in the present study. The Mal isolate contains a 23-nucleotide duplication in the same region in PBS that we find as a regulator of dimerization. Moreover, this duplication leads to structural differences in the initiation of reverse transcription in Mal compared to the prototypic subtype B strain NL43 (refs. [52,53]). It is therefore plausible that Mal and NL43 isolates use related, yet distinct, structural rearrangements to regulate dimerization. Nonetheless, both the polyA–SL1 and PBS–SL1 interactions are conserved among 800 curated sequences in the Los Alamos HIV-1 sequence database indicating regulation of dimerization by polyA and PBS is widespread (Supplementary Information).

Finally, we identified a new interaction between PBS and SL1 that acts as negative regulator of dimerization, Pr55[Gag] binding and packaging. We demonstrated that this negative regulation can be counteracted through the binding of oligos to the PBS. Disruption of this negative regulation would mechanistically explain why tRNA annealing enhances dimerization[31,70], and also opens up the possibility that primer binding to the PBS affect other steps of the HIV-1 life cycle, such as translation, by altering the monomer–dimer equilibrium. It also reveals a general principle by which RNA structural changes induced by host factors can regulate key stages of the HIV-1 life cycle (Fig. 7d).

## Online content

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

## Methods

**Plasmid.** NL43 sequences were obtained from pDRNL43 ΔEnv plasmid, which contains full-length NL43 but without flanking cellular sequences[71] and contains a deletion in Env for biosafety.

**Protein expression and purification.** Expression, purification and characterization of NL4.3 Pr55$^{Gag}$ with an appended C-terminal His$_6$-tag was performed as described by McKinstry et al.[72].

**Mutant library preparation.** DNA templates were prepared by PCR using Taq DNA polymerase (NEB) with RNA expression plasmid pDRNL43- ΔEnv and forward primers containing T7 RNA polymerase promoter and 3G/2G/1G at the 5′ end AAAgaagacTTggggTAATACGACTCACTATAGGG TCTCTCTGGTTAGACCAG / AAAgaagacTTggggTAATACGACTCACTATA GGTCTCTCTGGTTAGACCAG / AAAgaagacTTggggTAATACGACTCAC TATAGTCTCTCTGGTTAGACCAG and reverse primer mGmATCTAAGTTC TTCTGATCCTGTCTG. PCR amplifications were performed in 1× reaction buffer, 0.2 mM dNTPs, 250 nM forward primer and reverse primer, 1 ng of plasmid as template and 1.25 U of Taq DNA polymerase (NEB) using the PCR cycling conditions: 98 °C for 30 s, followed by 32 cycles of 98 °C for 10 s, 60 °C for 30 s and 68 °C for 1 min. Products were visualized by electrophoresis on 1% agarose gels in 1× TAE buffer and column purified with NucleoSpin Gel and PCR Clean-up kit (Macherey-Nagel). The purified PCR products were used as template for error prone PCR using the Mutazyme II DNA polymerase (Agilent) and forward primer TAATACGACTCACTATA and reverse primer GTCTCGTGGGCTCGGAGATGT-GTATAAGAGACAGGATCTAAGTTCTTCTGATCCTGTCTG. The PCR reaction volume was 50 µl and consisted of 2 ng of template DNA, 1× buffer, 200 µM dNTPs, 0.25 mM for each primer and 2.5 U of Mutazyme II DNA polymerase. PCR cycling conditions were 95 °C for 2 min followed by 35 cycles of 95 °C for 30 s, 35–42 °C for 30 s and 72 °C for 1 min. Products were visualized by electrophoresis on 1% agarose gels in 1× TAE buffer. A final column purification was carried out with the NucleoSpin Gel and PCR Clean-up kit (Macherey-Nagel) kit.

**RNA preparation.** Purified WT and mutated PCR products (900 ng) were used as templates for RNA in vitro transcription with a homemade T7 RNA polymerase. Reaction contained 1× reaction buffer (40 mM Tris pH 7.5, 18 mM MgCl$_2$, 10 mM DDT, 1 mM Spermidine), 5 mM NTPs, 40 U RNasin (Molox), 900 ng of DNA template, 0.05 U of Pyrophosphatase (NEB) and 5 µl of homemade T7 RNA polymerase. The reaction was incubated at 37 °C for 3 h, followed by DNase I treatment for 30 min at 37 °C. RNA was gel purified after electrophoresis on 1% agarose gels in 1× TAE buffer using the NucleoSpin Gel and PCR Clean-up kit (Macherey-Nagel) with NTC buffer (Macherey-Nagel). Half of the purified RNA was capped with Vaccinia Capping System (NEB). Briefly, 10 µg of RNA was mixed with nuclease-free H$_2$O in a 1.5-ml microfuge tube to a final volume of 15 µl. The sample was heated at 65 °C for 5 min, then placed on ice for 5 min. Then 2 µl of 10× capping buffer, 1 µl of 10 mM GTP, 1 µl of 2 mM SAM and 1 µl of Vaccinia Capping Enzyme were added and the sample was incubated at 37 °C for 30 min. Capped RNA was column purified using the NucleoSpin Gel and PCR Clean-up kit (Macherey-Nagel) with NTC buffer (Macherey-Nagel).

**Native agarose gel electrophoresis.** RNA (600 ng) was denatured at 90 °C for 2 min followed by chilling on ice for 2 min. RNA was incubated at 37 °C for overnight (15–17 h) in high salt buffer (10 mM KH$_2$PO$_4$, pH 7.4, 1 mM MgCl$_2$, 122 mM KCl) or low salt buffer (10 mM NaCl, 10 mM Tris, pH 7.4). Samples were loaded with native loading dye (0.17% Bromophenol Blue and 40% (vol/vol) sucrose) on 1% agarose gel prepared with 1× tris-borate magnesium (TBM) buffer (89 mM Tris base, 89 mM boric acid and 0.2 mM MgCl$_2$) and fractionated at 100 V for 85 min at room temperature. In some experiments, 12 pmol of oligos cPBS(182–199) GTCCCTGTTCGGGCGCCA and/or cPBS(199–216)TTCCCTTTCGCTTTCAAG were added to the RNA before denaturing to assess the effect of disrupting the PBS on dimerization.

**Native polyacrylamide gel electrophoresis.** RNA (800 ng) was denatured at 90 °C for 2 min followed by chilling on ice for 2 min. RNA was then incubated at 37 °C for 30 min in high salt buffer (50 mM sodium cacodylate, pH 7.5, 300 mM KCl and 5 mM MgCl$_2$) or low salt buffer (50 mM sodium cacodylate, pH 7.5, 40 mM KCl and 0.1 mM MgCl$_2$). Samples were loaded with native loading dye (0.17% Bromophenol Blue and 40% (vol/vol) sucrose) on 4% acrylamide nondenaturing gel prepared with 1× TBM (89 mM Tris base, 89 mM boric acid and 0.1 mM MgCl$_2$) and fractionated at 150 V for 4 h at 4 °C, including two reference samples with SYBR gold (Invitrogen), which could be visualized under blue LED light. The dimer and monomer bands in samples were cut from the gel according to the position of reference samples by scalpel.

**In-gel DMS probing.** Each gel piece from the polyacrylamide gel was divided into two parts. Half was soaked in 1× TBM containing 170 mM DMS (dissolved in EtOH), incubated at 37 °C for 15 min, followed by quenching with 50% (final) β-mercaptoethanol. The other half was soaked in 1× TBM (89 mM Tris base, 89 mM boric acid and 0.1 mM MgCl$_2$) containing the equivalent volume of EtOH

as the DMS treated sample, and incubated at 37 °C for 15 min. Gel slices were crushed into small pieces, soaked in 1× TBM (89 mM Tris base, 89 mM boric acid and 0.1 mM MgCl$_2$) buffer at 4 °C overnight. RNA was extracted using NucleoSpin Gel and PCR Clean-up kit (Macherey-Nagel) with NTC buffer (Macherey-Nagel).

Reverse transcription of 35 ng of DMS modified RNA or 25 ng of control RNA was performed with 200 U of SuperScript II reverse transcriptase (Invitrogen), 0.1 µM reverse transcription primer GTCTCGTGGGCTCGGA GATGTGTATAAGAGACAGGATCTAAGTTCTTCTGATCCTGTCTG, 0.5 mM dNTPs, 50 mM Tris·HCl, pH 8.0, 75 mM KCl, 6 mM MnCl$_2$, 10 mM DTT in 20-µl reactions. The reverse transcription reaction was incubated at 42 °C for 3 h.

**Library preparation.** For the functional probing MIME experiments, reverse transcribed complementary DNAs were amplified with 250-nM primers forward TCGTCGGCAGCGTCAGATGTGTATAAGAGACAGGGTCTCTCTGG TTAGACC, reverse GTCTCGTGGGCTCGGAGATGTGTATAAGAGACAGG ATGGTTGTAGCTGTCCCAG, 200 µM dNTPs, 1× Q5 reaction buffer, Q5 polymerase (NEB) using the PCR cycling conditions: 98 °C for 30 s, followed by 32 cycles of 98 °C for 10 s, 55 °C for 30 s, and 72 °C for 30 s. The PCR products were visualized by electrophoresis on 1% agarose gels in 1× TAE buffer and column purified (using the NucleoSpin Gel and PCR Clean-up kit, Macherey-Nagel). Then 25 ng of purified products were used in the final sequencing library preparation with Nextera DNA Flex Library Prep (Illumina) and Nextera DNA CD Indexes (96 Indexes, 96 Samples, Illumina), according to the manufacturer's instructions. For structural profiling by DMS, we performed amplicon sequencing. PCR reaction volume was 25 µl, 200 µM dNTPs, 250 nM primer pair 1 (forward TCGTCGGCAGCGTCAGATGTGTATAAGAGACAGggtctctctggttagacc and reverse GTCTCGTGGGCTCGGAGATGTGTATAAGAGACAGGCG TACTCACCAGTCGCC) or primer pair 2 (forward TCGTCGGCAGCGT CAGATGTGTATAAGAGACAGcgaaagtaaagccagaggag and reverse GTCTCGTGGGCTCGGAGATGTGTATAAGAGACAGCTCCCTG CTTGCCCATAC), 1× GXL reaction buffer, 0.625 U of PrimeSTAR GXL DNA Polymerase (Takara Bio). Two PCR amplifications were performed using the PCR cycling conditions: 98 °C for 30 s, followed by 34 cycles of 98 °C for 10 s, 60 °C for 15 s and 68 °C for 30 s. Amplified libraries were column purified using the NucleoSpin Gel and PCR Clean-up kit (Macherey-Nagel). Paired-end PE150 sequencing was carried out on an Illumina Novaseq instrument (Novogene).

For the data analysis, sequencing data relating to MIME functional profiling experiments were first preprocessed using automated python scripts. Sequencing reads were quality trimmed and stripped of adapters with CutAdapt with the parameters '--nextseq-trim 35 – max-n 0 -A CTGTCTCTTATA -a CTGTCTCTTATA'. Second, reads were aligned to the HIV-1 5′ UTR using Novoalign with the parameters '-o SAM -o SoftClip'. Sam files were then analyzed using MIMEAnTo[73] to generate $K^{dimer}$, which is a quantitative metric relating the effect of a mutation on dimerization (derivation in Supplementary Information). Statistical methods used in MIMEAnTo are described in detail elsewhere[38,73].

Sequencing data relating to DMS structural probing were first preprocessed with ShapeMapper2 using parameters '--output-parsed-mutations--output-counted-mutations--render-mutation'. EtOH treated and DMS treated raw sequencing reads were passed to ShapeMapper2 via the modified and unmodified parameters, respectively[51]. DMS reactivities were calculated from ShapeMapper2 mutation rates using 90% Winsoring[43]. DMS reactivities were saved as XML files for processing with rf-fold module of the RNA Framework software package[54,74]. rf-fold was used to calculate Shannon entropies and base-pairing probabilities with the parameters '-ow -dp -KT -sh -g'. Initial RNA structure predictions of monomer and dimer conformations, using DMS reactivities as soft constraints, were performed with rf-fold using the RNA folding algorithms in the Vienna RNA v.2.0 package[55]. Refined RNA structure predictions using multidimensional probing results as additional hard constraints were performed using RNAfold of the Vienna RNA 2.0 package[55]. Cluster maps of DMS reactivities were generated using the clustermap function of the python Seaborn data visualization library using 'kendall' correlation method and 'average' cluster method (v.0.11.1). PCA was carried out using the PCA function of the python scikit-learn library (v.0.23.2). Variances in DMS reactivities were calculated using the var function from the python NumPy library (v.1.19.2). Pairwise comparison of DMS reactivities were carried out using a modified deltaSHAPE calculation[75]. This modified deltaSHAPE (v.1.0) analysis uses several criteria to identify statistically significant changes in reactivities. First, a z-factor test identifies nucleotides where DMS reactivities change by >1.96 standard deviations of the DMS errors. Second, a standard score threshold of 1.5 is applied, meaning that delta reactivity values are at least 1.5 standard deviations away from the mean reactivity change. To filter these statistically significant sites for biological meaning, we next applied an absolute and a relative threshold filter. The absolute difference threshold ensures that a minimum reactivity change of 0.2 is needed for the site to be considered biologically relevant. The relative threshold filter was set so that a relative change of at least 0.75-fold was needed to remove false positives where DMS reactivities are high in both conditions such that a large change in reactivity is unlikely to affect RNA structure. RNA structures were visualized using Visualization Applet for RNA (VARNA)[76].

RNA structural interference by multidimensional structural probing was carried out using the M2-seq pipeline[46]. Briefly, data were preprocessed by ShapeMapper into simple files that are string representations of mutations in each read. Simple files were converted into the rich and compact rdat format specific for RNA structure mapping experiments[77]. A two-dimensional matrix containing mutation rates at pairs of nucleotide positions was constructed. Mutation counts were subsequently normalized for total number of mutations along each row to give a true modification frequency. RNA structure signatures were further refined by calculating z-scores. A thresholding of zero was applied to remove negative values, and a convolution filter was applied to enhance cross diagonal features. RNA helices were finally identified in an unbiased manner by applying a filter for stems of Watson–Crick and G-U wobble base pairs of at least three base pairs in length. Best stems were predicted by eliminating conflicting stems by selecting the highest scoring stem. Bootstrapping analyses were performed using the rna_structure function of the Basic Inference Engine for RNA Structure (Biers) (https://ribokit.github.io/Biers/) using the default parameters (100 bootstrapping iterations).

For MST, RNA was labeled at the 3′ end using cytidine-5′-phosphate-3′-(6-aminohexyl) phosphate (Jena Biosciences) with T4 RNA ligase (NEB) overnight at 16 °C, followed by RNA Clean and Concentrator Kits (ZYMO). The 500-nM labeled and purified RNA was denatured at 90 °C for 2 min followed by chilling on ice for 2 min. RNA was folded at 37 °C for overnight (15–17 h) in high salt buffer (10 mM $KH_2PO_4$, pH 7.4, 1 mM $MgCl_2$, 122 mM KCl). For each binding experiment, RNA was diluted to 10 nM in high salt buffer (10 mM $KH_2PO_4$, pH 7.4, 1 mM $MgCl_2$, 122 mM KCl, 0.01% Triton X-100, 10 mM DTT, 0.02% BSA and yeast tRNA 0.2 mg ml⁻¹). A series of 16 tubes with Pr55[Gag] dilutions were prepared in high salt buffer, producing Pr55[Gag] ligand concentrations ranging from 30 pM to 1 μM. For measurements, each ligand dilution was mixed with one volume of labeled RNA, which led to a final concentration of 5 nM labeled RNA. The reaction was mixed by pipetting, incubated for 30 min at 37 °C, followed by 30 min on ice. Samples were then centrifuged at 10,000g for 5 min. Capillary forces were used to load the samples into Monolith NT.115 Premium Capillaries (NanoTemper Technologies). Measurements were performed using a Monolith Pico instrument (NanoTemper Technologies) at an ambient temperature of 25 °C. Instrument parameters were adjusted to 5% LED power, medium MST power and MST on-time of 1.5 s. An initial fluorescence scan was performed across the capillaries to determine the sample quality and afterward, 16 subsequent thermophoresis measurements were performed. Data from three independently pipetted measurements were analyzed for the $\Delta F_{norm}$ values and binding affinities were determined by the MO. Affinity Analysis software (v.2.3 NanoTemper Technologies). Graphs were plotted using GraphPad Prism v.8.4.3 software.

**In solution DMS-MaPseq (in vitro).** RNA (300 nM) was denatured at 90 °C for 2 min followed by chilling on ice for 2 min. Next, RNA was refolded at 37 °C for overnight (15–17 h) in high salt buffer (10 mM $KH_2PO_4$, pH 7.4, 1 mM $MgCl_2$, 122 mM KCl). DMS was added to the RNA solution to final concentration 170 mM, incubated at 37 °C for 6 min, followed by quenching with β-mercaptoethanol and purification with ethanol precipitation. The purified DMS probed RNAs followed the same reverse transcription and library preparation process as the in-gel DMS probed RNA samples.

**In solution DMS-MaPseq (in cells).** Here, 24 h before transfection, 10⁷ human embryonic kidney 293T (HEK293T) cells were plated in 10 ml of DMEM media containing 10% FBS. Next, 4 μg of plasmids expressing HIV-1 WT or HIV-1 mutants were mixed with 48 μl of polyethylenimine (1 mg ml⁻¹, PEI Max transfection grade linear polyethylenimine hydrochloride (MW 40k), Polysciences) and 500 μl of DMEM, and incubated for 10 min at room temperature before being added dropwise on the cells. Then 24 h posttransfection, the cells were probed by replacing the media with 3 ml of DMEM containing 170 mM DMS and incubated at 37 °C for 6 min. Cells were then washed with 5 ml of PBS containing 140 mM β-mercaptoethanol to quench the DMS. Next, 1 ml of TRI-reagent (Sigma-Aldrich) was directly added on the cells to extract RNA according to the manufacturer's instructions. DNA contaminants were removed by TurboDNase (Invitrogen) treatment for 30 min at 37 °C. RNA was purified using NucleoSpin Gel and PCR Clean-up kit (Macherey-Nagel) with NTC buffer (Macherey-Nagel) and eluted in 20 μl of 5 mM Tris-HCl pH 8.0. Then 7 μl of the purified DMS probed RNAs were used for the reverse transcription following the same reverse transcription and library preparation process as for the in-gel DMS probed RNA samples.

**Competition assay.** Here, 24 h before transfection, 7 × 10⁵ HEK293T cells were plated in 2 ml of DMEM media containing 10% FBS. For the cotransfection experiments, equal amounts of plasmids expressing WT or mutants (600 ng total) were mixed with 7.2 μl of polyethylenimine (1 mg ml⁻¹, Max 40k, Polysciences) and 100 μl of DMEM and incubated for 10 min at room temperature before being added dropwise on the cells. Cells and viral supernatant were collected at 24 h posttransfection. Cells were washed with 2 ml of PBS and RNA was extracted with 1,000 μl TRI-reagent (Sigma-Aldrich) according to the manufacturer's instructions. Viral supernatant was first clarified for 2 min at 17,000g, followed

by a filtration step through a 0.45-μm filter. The filtrate was then transferred into a new tube and the virus was pelleted for 2 h at 17,000g. The viral pellet was extracted with 500 μl of TRI-reagent (Sigma-Aldrich) and DNA contaminants were removed by TurboDNase (Invitrogen) treatment for 30 min at 37 °C. RNA was purified using NucleoSpin Gel and PCR Clean-up kit (Macherey-Nagel) with NTC buffer (Macherey-Nagel) and eluted in 20 μl of 5 mM Tris-HCl pH 8.0. Then 10 μl of purified RNA was heat denatured at 65 °C for 5 min together with 0.67 μM reverse primer (GATGGTTGTAGCTGTCCCAGTATTTGCC) and 1.67 mM dNTPs in 15 μl of total volume, then chilled on ice for 2 min. RNA was then reverse transcribed by adding 1× SSIV buffer, 5 mM DTT, 20U RNasin, 100 U of SSIV in 25 μl of total volume and incubating at 52 °C for 1 h. cDNAs were amplified with 250 nM primers forward TCGTCGGCAGCGTCAGATGTGTATAAGAGACAGggtctctcggttagacc, reverse GTCTCGTGGGCTCGGAGATGTGTATAAGAGACAGGCGTACTCACCAGT CGCC, 200 μM dNTPs, 1× Q5 reaction buffer and 0.02 U μl⁻¹ of Q5 polymerase (NEB) using the PCR cycling conditions: 98 °C for 1 min, followed by 22 cycles of 98 °C for 10 s, 55 °C for 20 s and 72 °C for 30 s. The PCR products were visualized by electrophoresis on 1% agarose gels in 1× TAE buffer and column purified (using the NucleoSpin Gel and PCR Clean-up kit, Macherey-Nagel). 40 ng purified products were used in the final indexing PCR using 2.5 μl of Nextera DNA CD Indexes (96 indexes, 96 samples, Illumina) in a 14 μl of reaction (200 μM dNTPs, 1× Q5 reaction buffer and 0.02 U μl⁻¹ of Q5 polymerase (NEB)). The PCR cycling conditions were 98 °C for 2 min, followed by five cycles of 98 °C for 30 s, 6,255 °C for 320 s and 72 °C for 1 min. Paired-end PE150 sequencing was carried out on an Miniseq instrument (Illumina) according to the manufacturer's instructions.

**Statistics and reproducibility.** Details of the description of statistical methods are provided in the Supplementary Information. No statistical method was used to predetermine sample size.

**Reporting Summary.** Further information on research design is available in the Nature Research Reporting Summary linked to this article.

## Data availability
The datasets generated during this study are available at the NCBI bioproject ID PRJNA771368. HIV-1 sequences were downloaded from the Los Alamos HIV-1 sequence database (https://www.hiv.lanl.gov/content/index). Additional raw and processed data files are provided as Source data and Supplementary Information with this paper.

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

## Acknowledgements
We thank J.-C. Paillart and R. Marquet for critical feedback. We also thank the Helmholtz Association (grant no. VH-NG-1347 to R.S.) and the Bundesministerium für Bildung und Forschung (grant no. COMPLS-182 to R.S. and M.v.K.). A.S.G.-B. was supported with a fellowship from the Peter und Traudl Engelhorn Stiftung. N.C. received funding from the European Research Council (ERC) grant no. 948636. The funders had no role in study design, data collection and analysis, decision to publish or preparation of the manuscript.

## Author contributions
R.P.S., A.S.G.-B. and L.Y. conceived the study. L.Y., A.S.G.-B., C.B., U.B.A., S.A. and M.O.-N. performed the experiments. R.P.S., M.v.K., P.B. and M.S. performed the analysis. A.K. and N.C. purified Pr55[Gag] and performed MST measurements. R.P.S. and L.Y. wrote the manuscript with contributions from the other authors.

## Funding

## Competing interests

The authors declare no competing interests.

## Additional information

**Extended data** are available for this paper at https://doi.org/10.1038/s41594-022-00746-2.

**Correspondence and requests for materials** should be addressed to Redmond P. Smyth.

**a**

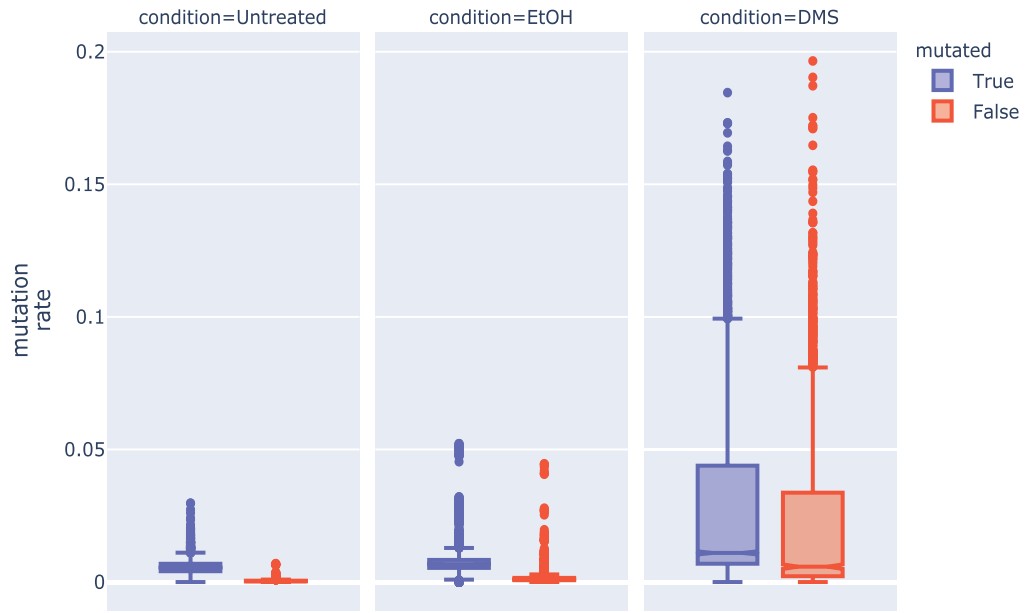

**b**

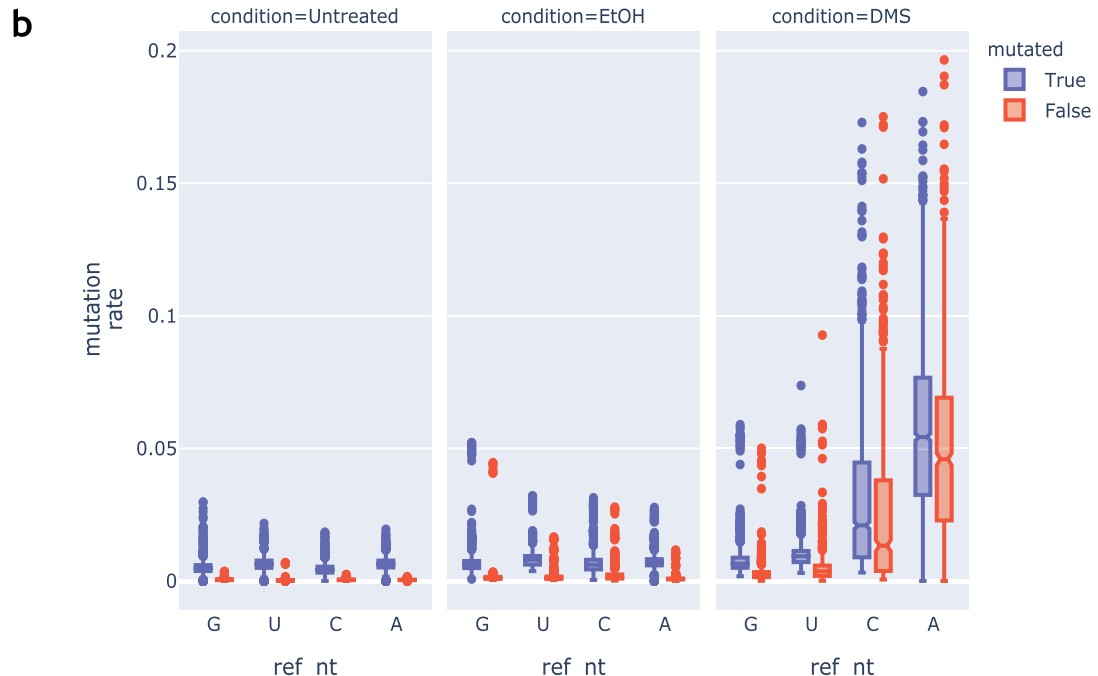

**Extended Data Fig. 1 | Global and nucleotide specific mutation rates.** Global and nucleotide specific mutation rates expressed as mutations per nucleotide. (**a**) Global mutation rates for mutated (blue) and unmutated (red) samples that were untreated (left panel), ethanol treated (middle panel) and DMS treated samples (right panel). Mutation rates are higher in mutated compared to unmutated samples. Untreated samples, and samples treated as DMS control (EtOH) have similar mutation rates. DMS treated samples show a greatly increase mutation rate in both mutated and unmutated samples compared to the controls. (**b**) Nucleotide specific mutation rates (A, C, G, U) for mutated (blue) and unmutated (red) samples that were untreated (left panel), ethanol treated (middle panel) and DMS treated samples (right panel). Mutation frequencies in the mutated samples are consistently higher at all nucleotides in the mutated compared to unmutated samples. In the DMS treated samples, mutations are greatly enriched at C and A residues, as expected by the selectivity of the DMS chemical. Box plots show quartile 1 (Q1) to quartile 3 (Q3). The second quartile (Q2) is marked by a line inside the box. Whiskers correspond to the box' edges +/− 1.5 times the interquartile range (IQR: Q3-Q1). Outliers are shown as points. Data are pooled from two independent experiments, each consisting of 32 independent samples.

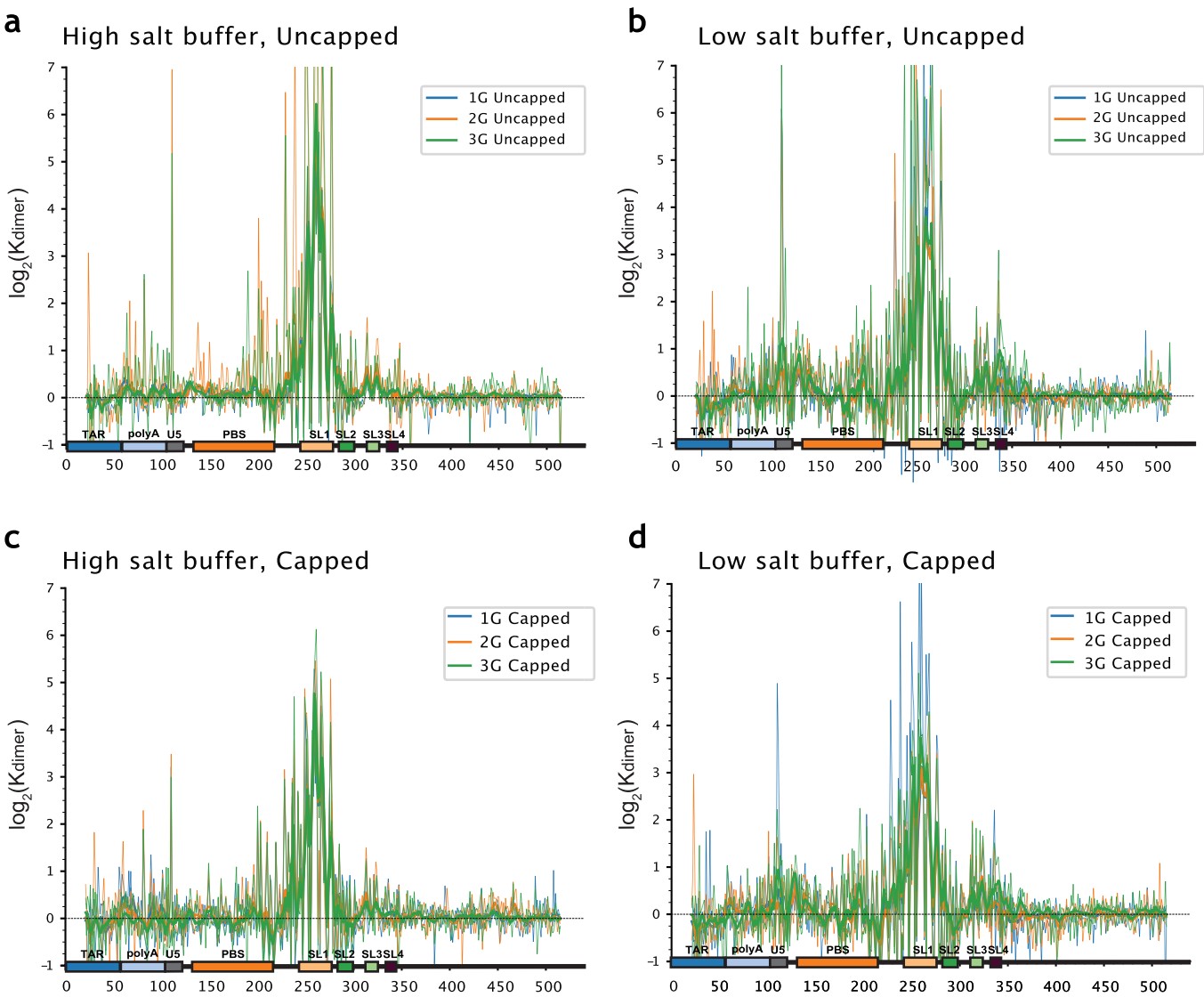

**Extended Data Fig. 2 | Functional profiling of sequences involved in dimerization.** Functional profiling of sequences involved in dimerization by analysed by mutational interference. $k^{dimer}$ is a relative measure of the effects of a mutation on dimerization. median $\log_2(k^{dimer})$ values for each genome position for all three uncapped transcript variants in high and low salt buffers. Thin lines are unsmoothed data, whereas thick lines are smoothed with a window size of 5 nt. $\log_2(k^{dimer})$ values for (**a**) high salt uncapped transcripts (**b**) low salt uncapped transcripts (**c**) high salt capped transcripts (**d**) low salt capped transcripts.

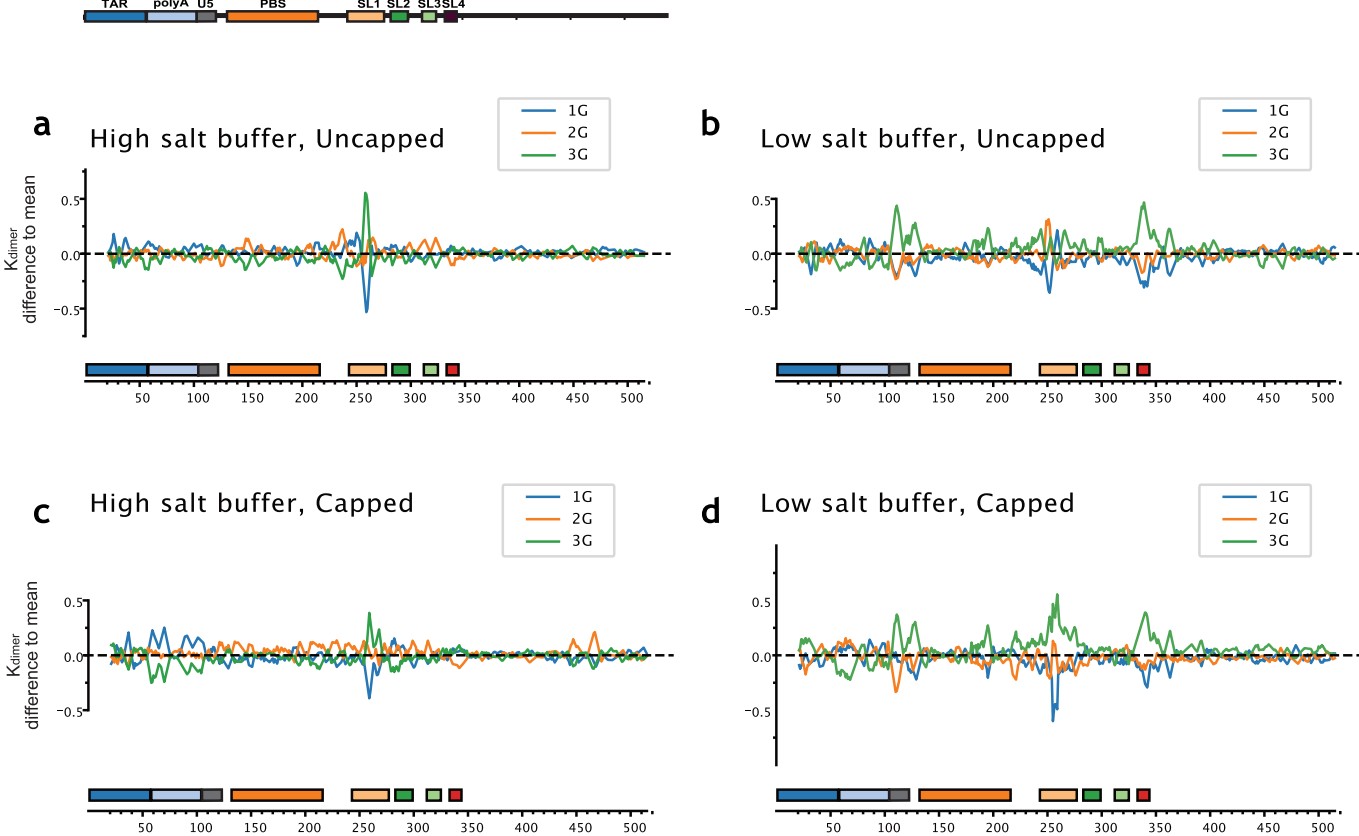

**Extended Data Fig. 3 | Relative dimerization properties of 1G, 2G, 3G transcripts.** $\log_2(k^{dimer})$ values of the 1G, 2G, and 3G transcript variants compared to the mean of the 1G, 2G and 3G transcripts for (**a**) high salt uncapped transcripts (**b**) low salt uncapped transcripts (**c**) high salt capped transcripts (**d**) low salt capped transcripts. In all conditions, regions within SL1 are more important for dimerization in 3G compared to 1G samples. In low salt conditions, the 3G variant had increased dependence on regions outside of SL1. Capped and uncapped RNAs show very similar profiles, with the exception of a region in polyA in high salt buffer, which was more important for dimerization in the 1G sample compared to 3G.

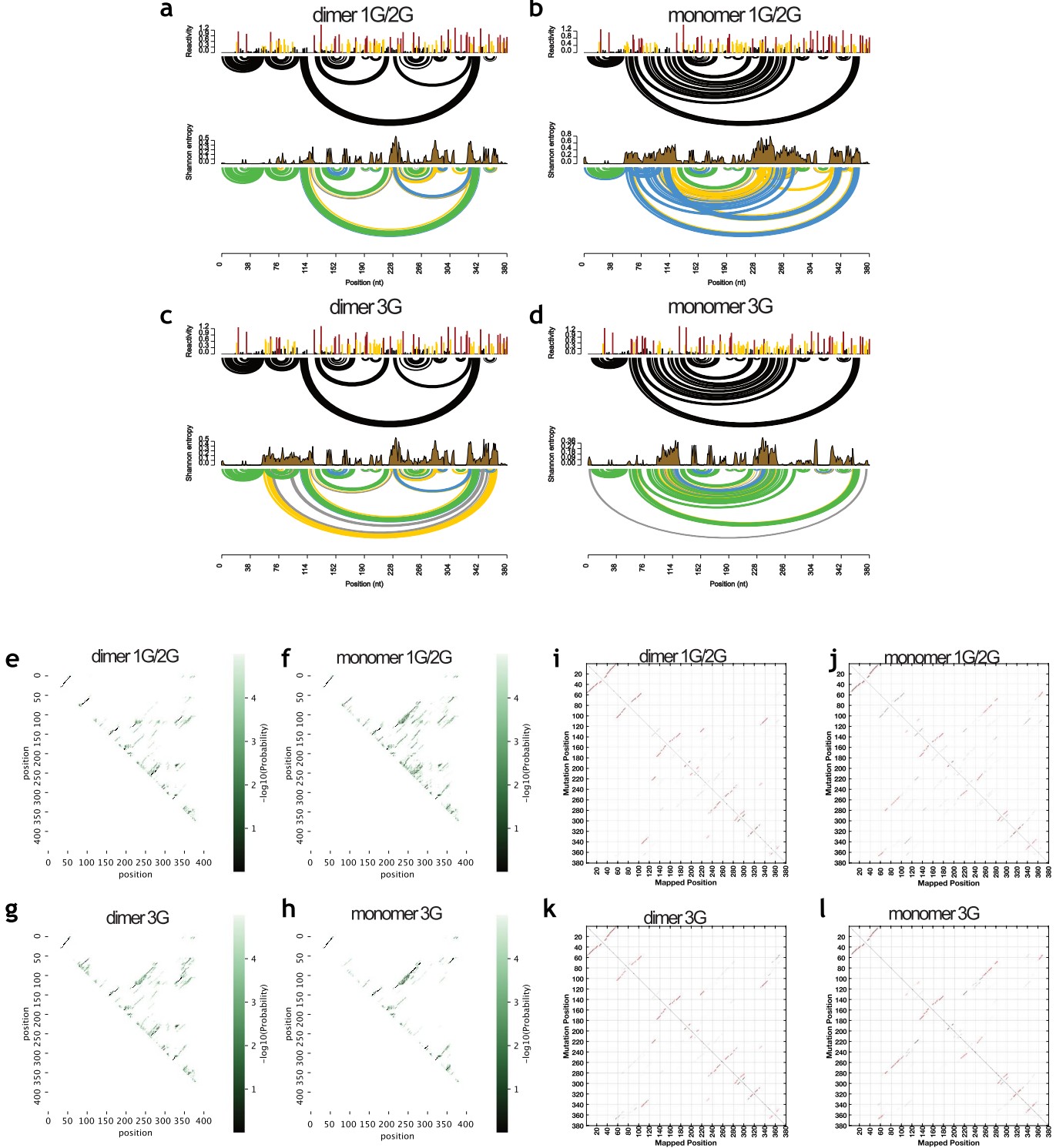

**Extended Data Fig. 4 | DMS reactivities and Shannon entropies.** DMS reactivities and Shannon entropies for the (**a**) 1G/2G dimer class (**b**) 1G/2G monomer class (**c**) 3G dimer class and (**d**) 3G monomer class. Arc plots show base pairing probabilities (green = 70-100%; blue=40-70%; yellow=10-40%; gray=5-10%). (**e-h**) Dot plots of RNA base pairing probabilities reveal alternative folding possibilities for the (**e**) 1G/2G dimer class (**f**) 1G/2G monomer class (**g**) 3G dimer class and (**h**) 3G monomer class. RNA stems are shown along the diagonals. (**i-l**) Bootstrapping analysis of the predicted dimer and monomer structure. Predicted structure is shown in red. The bootstrap support is shown in greyscale, with darker greys signifying better bootstrap support for the (**i**) 1G/2G dimer class (**j**) 1G/2G monomer class (**k**) 3G dimer class and (**l**) 3G monomer class.

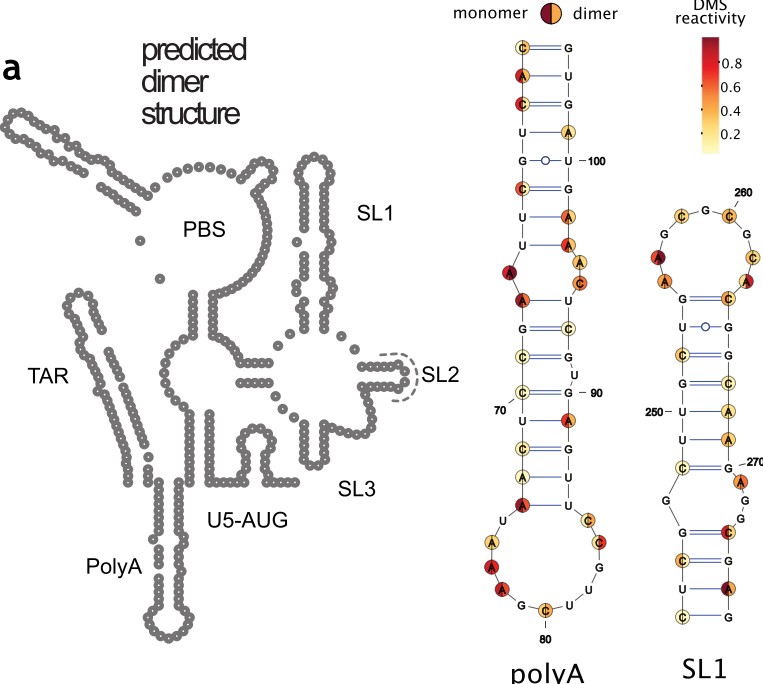

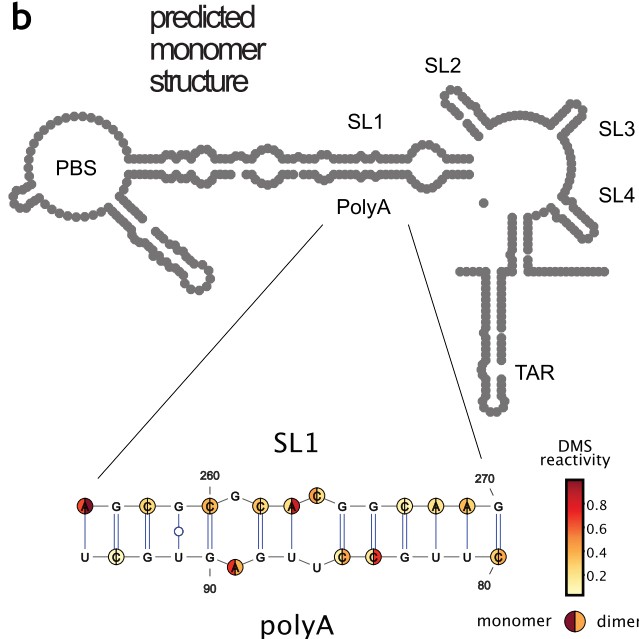

**Extended Data Fig. 5 | Secondary structure model for 1G/2G dimer and 3G monomer class.** Secondary structure model for 1G/2G dimer and 3G monomer class. (**a, b**) Secondary structure model of dimer and monomer class, respectively. Models were obtained using DMS reactivities as soft constraints for *in silico* folding in the Vienna RNA structure package. For the dimer structure, the U1sRNA binding site within SL2 is shown. For the insets, DMS reactivities from monomer and dimer samples were mapped to A and C residues. Structures of polyA and SL1 stem loops and polyA-SL1 interaction are shown. DMS reactivities for the monomer population are shown on the left hemisphere, and DMS reactivities for the dimer population on the right. Red signifies highly reactive positions that are unpaired. Pale yellow signifies unreactive positions that are base-paired.

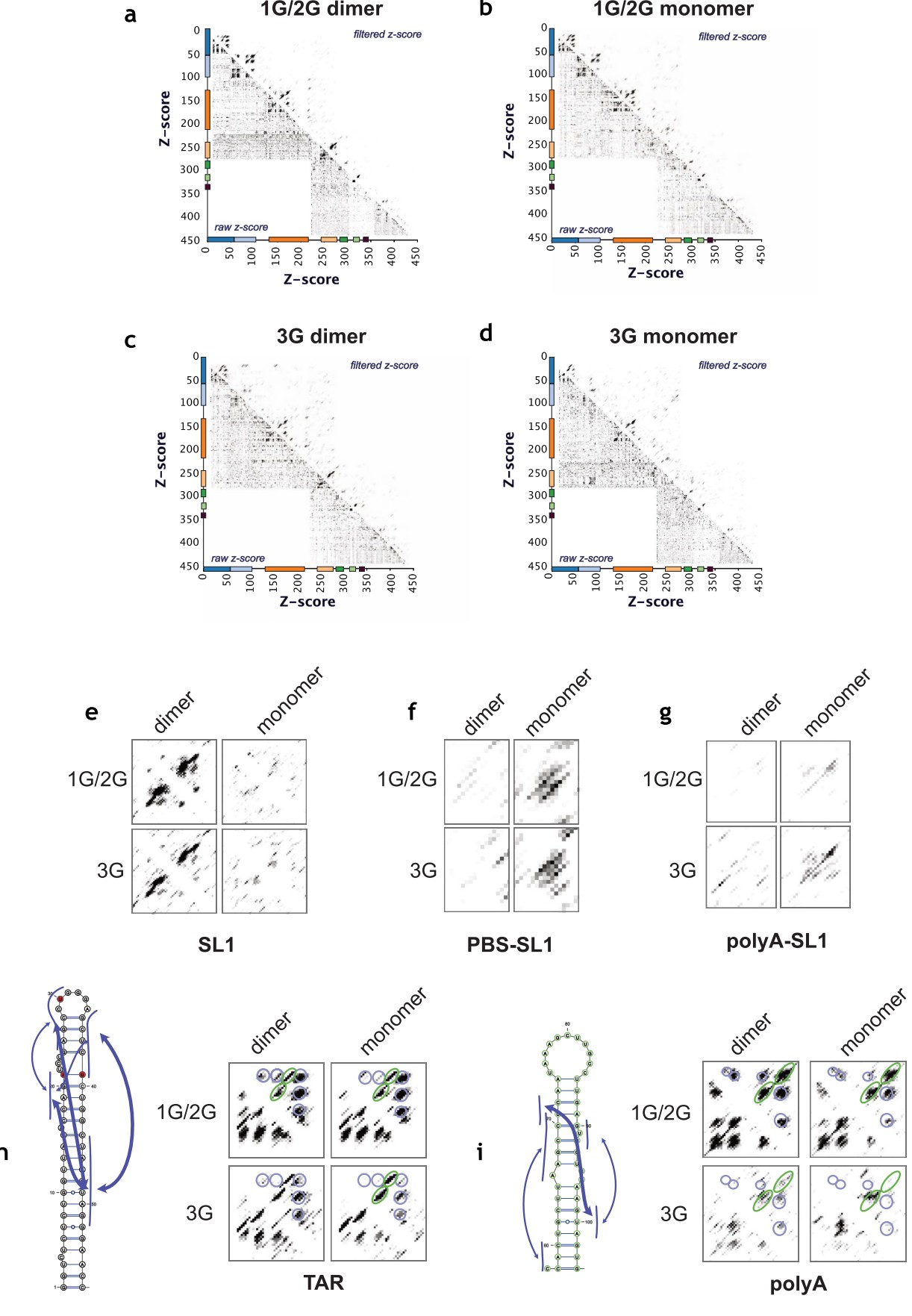

**Extended Data Fig. 6 | See next page for caption.**

**Extended Data Fig. 6 | Two dimensional plots of mutation frequencies.** Two dimensional plots of mutation frequencies for the (**a**) 1G/2G dimer class (**b**) 1G/2G monomer class (**c**) 3G dimer class and (**d**) 3G monomer class. z-scores of two-dimension structural probing data reveals RNA stems along the diagonal, as well as non-canonical or tertiary interactions. Regions in (**e**) SL1, (**f**) PBS-SL1, (**g**) polyA-SL1, (**h**) TAR and (**i**) polyA stem are highlighted. For TAR and polyA, detected stems are highlighted with green circles. Putative tertiary or non-canonical interactions are highlighted with purple circles and arrows.

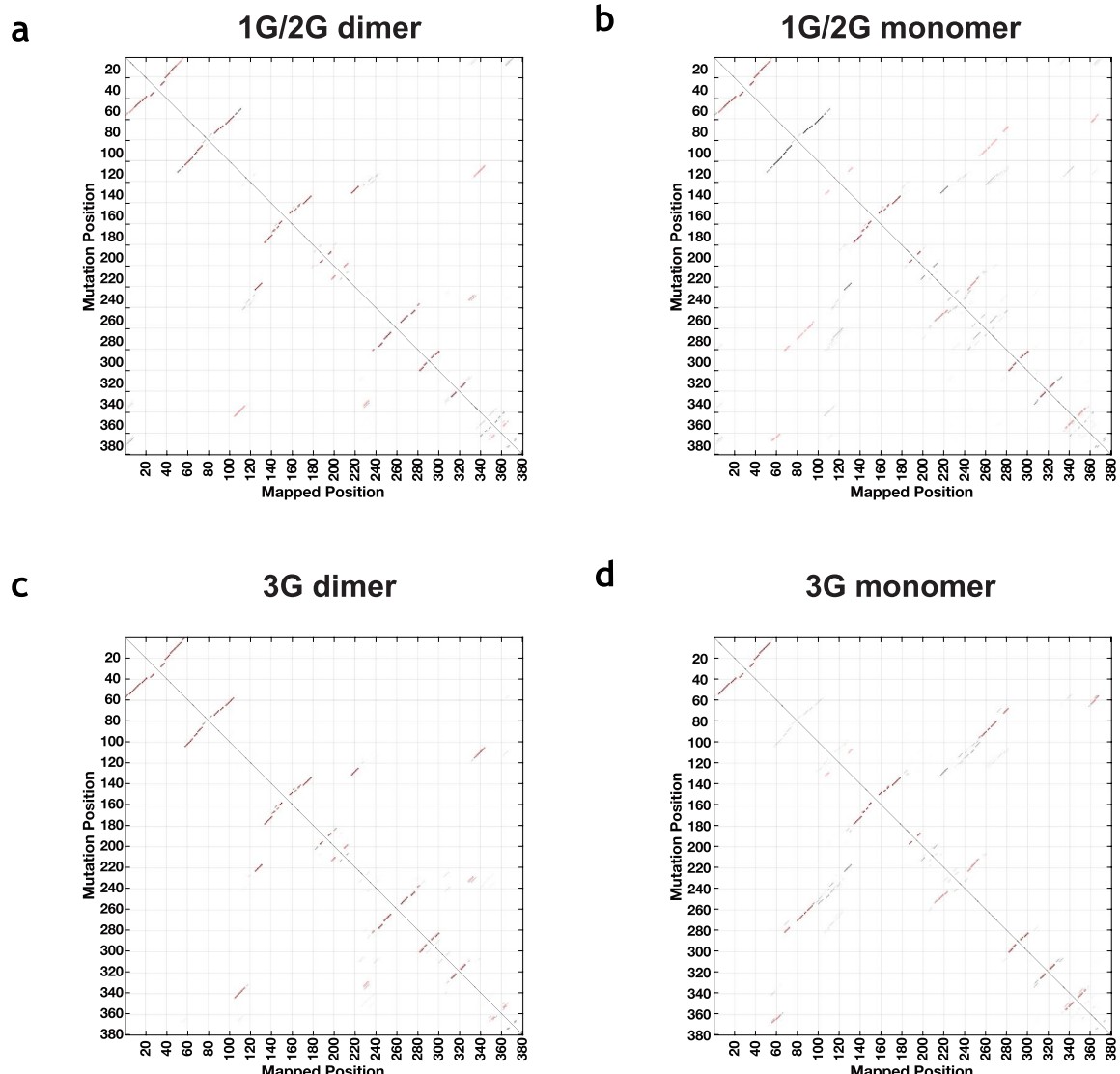

**Extended Data Fig. 7 | Bootstrapping analysis for 2-dimensional structural probing.** Bootstrapping analysis for 2-dimensional structural probing. The predicted structure for the enhanced dimer and monomer structures are shown in red. Bootstrap support is shown in greyscale, with darker greys signifying better bootstrap support for the (**a**) 1G/2G dimer class, (**b**) 1G/2G monomer class, (**c**) 3G dimer class, and (**d**) 3G monomer class.

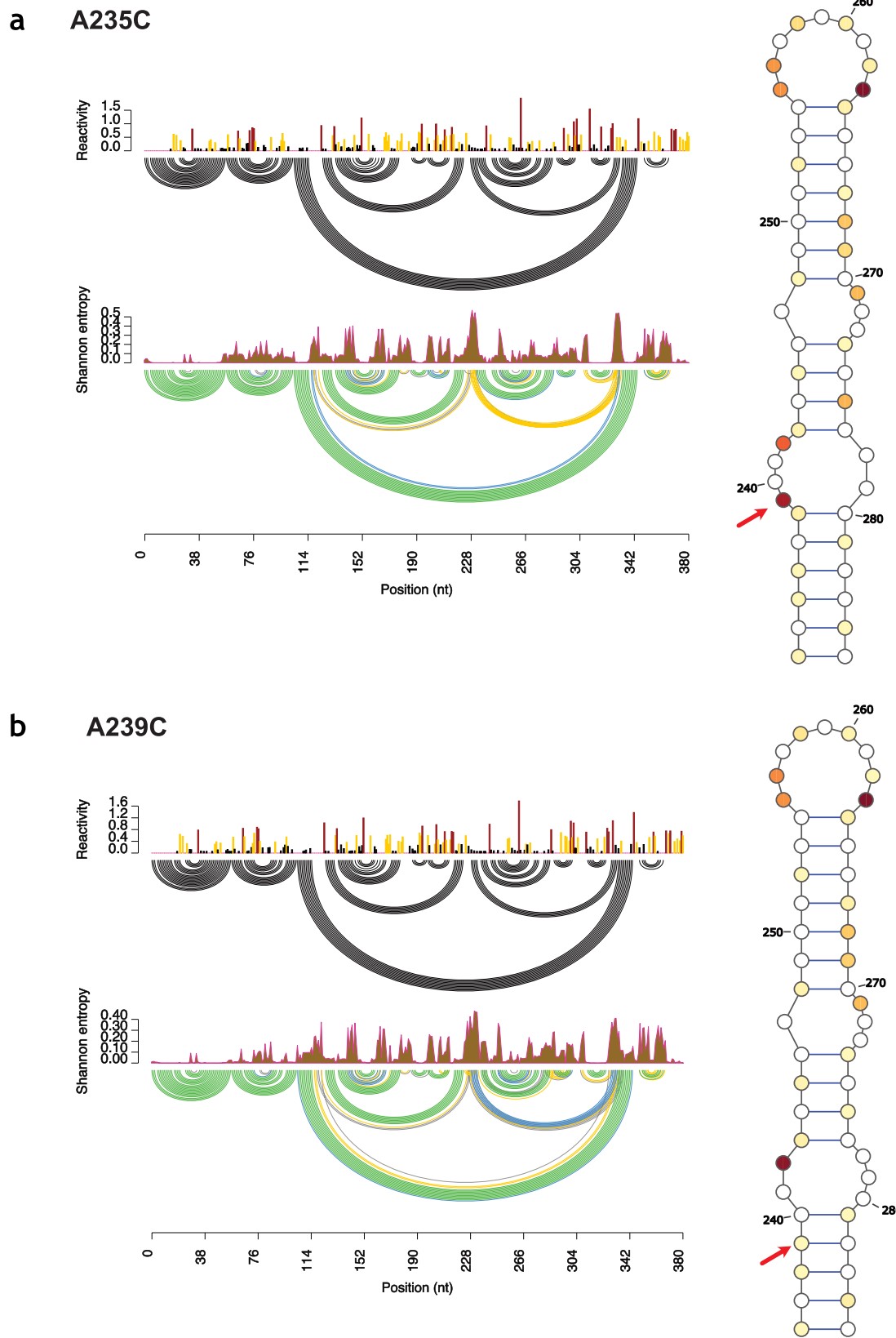

**Extended Data Fig. 8 |** See next page for caption.

**Extended Data Fig. 8 | Secondary structure model for SL1 mutants.** Secondary structure model for SL1 mutants. Models were obtained using DMS reactivities as soft constraints for in silico folding in the Vienna RNA structure package. DMS reactivities for each nucleotide position are show in the upper barchart. Shannon entropies are shown in the lower chart. Upper arc plots show consensus structure. Lower arc *plots show base pairing probabilities (green = 70-100%; blue = 40-70%; yellow = 10-40%; grey = 5-10%).* DMS reactivities from monomer and dimer samples were mapped to A and C residues on SL1. Red signifies highly reactive positions that are unpaired. Pale yellow signifies unreactive positions that are base-paired. (**a**) A235C and (**b**) A239C reconfigure the SL1 lower helix and internal loop to enhance dimerization. Red arrows highlight position 239 showing a reactivity change between the A235C and A239C mutations.

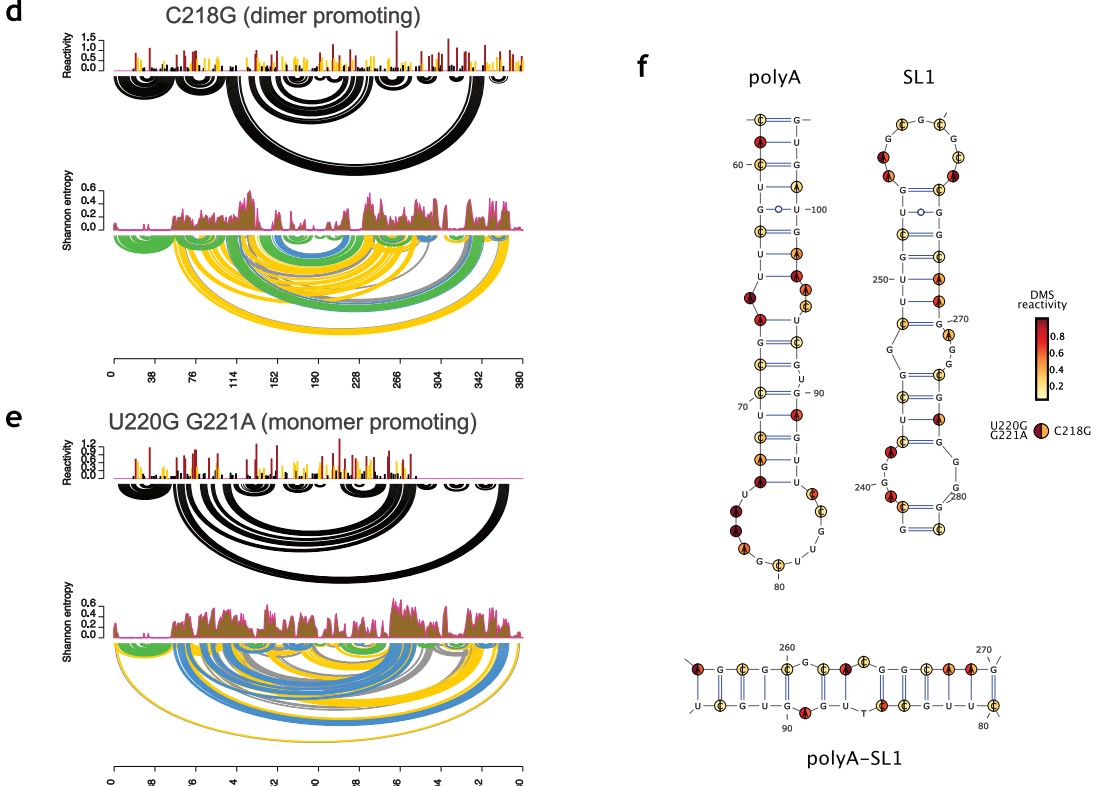

**Extended Data Fig. 9 | See next page for caption.**

**Extended Data Fig. 9 | Secondary structure predictions for dimer and monomer promoting mutants.** Secondary structure predictions for dimer and monomer promoting mutants targeting the polyA-SL1 and PBS-SL1 interactions. DMS reactivities and secondary structure models for polyA, SL1 and polyA-SL1. DMS reactivities for each nucleotide position are show in the upper barchart. Shannon entropies are shown in lower chart. Upper arc plots show consensus structure. Lower arc *plots show base pairing probabilities (green = 70-100%; blue = 40-70%; yellow = 10-40%; grey = 5-10%).* DMS reactivities from monomer and dimer samples were mapped to A and C residues. Red signifies highly reactive positions that are unpaired. Pale yellow signifies unreactive positions that are base-paired. (**a-c**) DMS reactivities and secondary structure models for polyA-SL1 mutants. (**a**) dimer promoting mutant C84A-C85A folds into the canonical 5′UTR structure (**b**) Monomer promoting mutant U86G–A89C contains the polyA-SL1 interaction. (**c**) Reactivities for both mutants U86G-A89C (left hemisphere) and C84A-C85A (right hemisphere) mapped to the structures polyA, SL1, and polyA-SL1. (**d**) Dimer promoting mutant C218G folds into the structure containing SL1 (**e**) Monomer promoting mutant U220G-G221A folds into a structure containing the PBS-SL1 interaction. (**f**) Reactivities for both mutants U220G-G221A (left hemisphere) and C218G (right hemisphere) mapped to the structures polyA, SL1 and polyA-SL1.

a 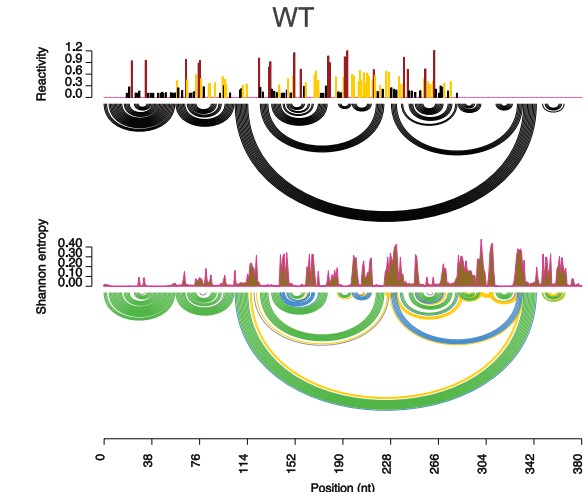

## polyA-SL1 mutants

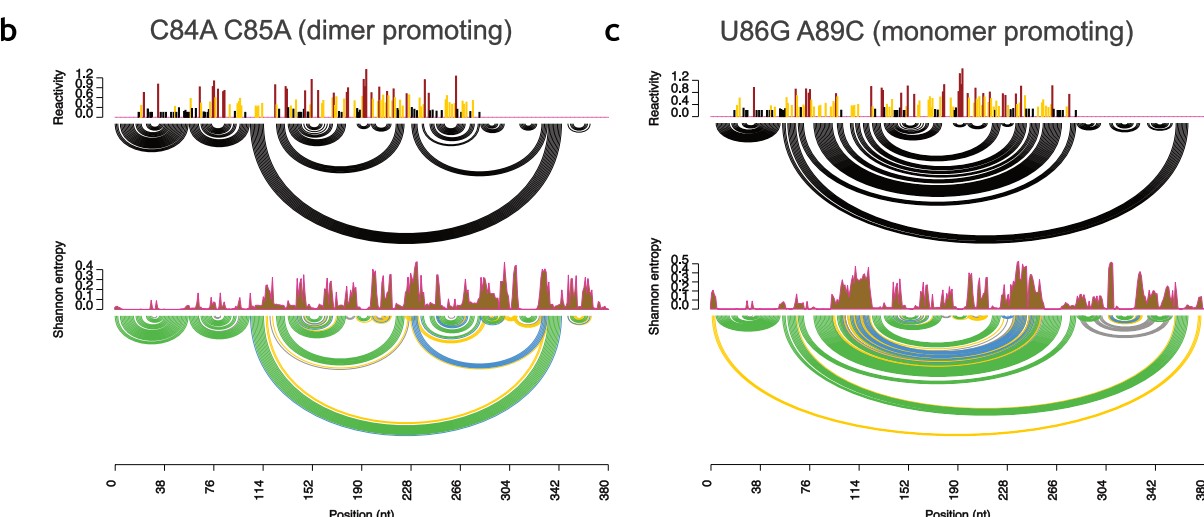

b C84A C85A (dimer promoting)

c U86G A89C (monomer promoting)

## PBS-SL1 mutants

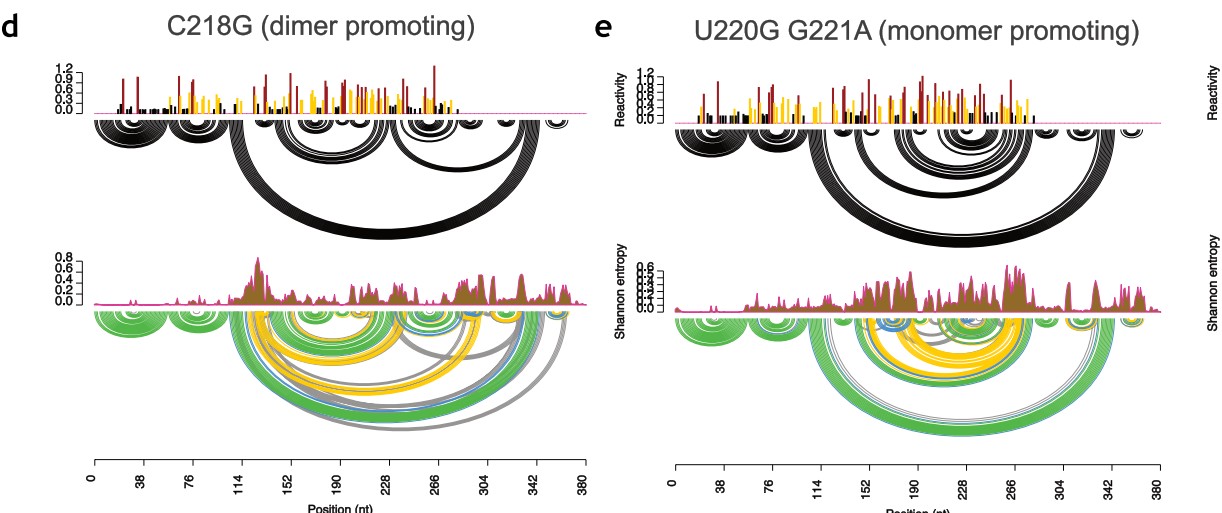

d C218G (dimer promoting)

e U220G G221A (monomer promoting)

**Extended Data Fig. 10 | See next page for caption.**

**Extended Data Fig. 10 | In cell DMS reactivities and secondary structure predictions for dimer and monomer promoting mutants.** In cell DMS reactivities and secondary structure predictions obtained for dimer and monomer promoting mutants targeting the polyA-SL1 and PBS-SL1 interactions. DMS reactivities for each nucleotide position are show in the upper barchart. Shannon entropies are shown in lower chart. Upper arc plots show consensus structure. Lower arc plots show base pairing probabilities (green = 70-100%; blue=40-70%; yellow=10-40%; grey=5-10%). (**a**) Wild-type HIV-1 (**b**) dimer promoting mutant C84A-C85A folds into the canonical 5'UTR structure (**c**) Monomer promoting mutant U86G-A89C contains the polyA-SL1 interaction. (**d**) Dimer promoting mutant C218G folds into the structure containing SL1 (**e**) Monomer promoting mutant U220G-G221A folds into a structure containing the PBS-SL1 interaction.

# Reporting Summary

## Statistics

For all statistical analyses, confirm that the following items are present in the figure legend, table legend, main text, or Methods section.

| n/a | Confirmed | |
|---|---|---|
| ☐ | ☒ | The exact sample size ($n$) for each experimental group/condition, given as a discrete number and unit of measurement |
| ☐ | ☒ | A statement on whether measurements were taken from distinct samples or whether the same sample was measured repeatedly |
| ☐ | ☒ | The statistical test(s) used AND whether they are one- or two-sided<br>*Only common tests should be described solely by name; describe more complex techniques in the Methods section.* |
| ☒ | ☐ | A description of all covariates tested |
| ☐ | ☒ | A description of any assumptions or corrections, such as tests of normality and adjustment for multiple comparisons |
| ☐ | ☒ | A full description of the statistical parameters including central tendency (e.g. means) or other basic estimates (e.g. regression coefficient) AND variation (e.g. standard deviation) or associated estimates of uncertainty (e.g. confidence intervals) |
| ☐ | ☒ | For null hypothesis testing, the test statistic (e.g. $F$, $t$, $r$) with confidence intervals, effect sizes, degrees of freedom and $P$ value noted<br>*Give P values as exact values whenever suitable.* |
| ☒ | ☐ | For Bayesian analysis, information on the choice of priors and Markov chain Monte Carlo settings |
| ☒ | ☐ | For hierarchical and complex designs, identification of the appropriate level for tests and full reporting of outcomes |
| ☒ | ☐ | Estimates of effect sizes (e.g. Cohen's $d$, Pearson's $r$), indicating how they were calculated |

*Our web collection on statistics for biologists contains articles on many of the points above.*

## Software and code

Policy information about availability of computer code

| Data collection | Data was generated on commerical Illumina sequencing (Nextseq, Novaseq and miniseq) and Nanotemper microscale thermophoresis platforms |
|---|---|
| Data analysis | Data was analysed using readily available software tools. Mutational interference data was analysed using MIMEAnTo (10.1093/bioinformatics/btw47). RNA structure data was analysed using ShapeMapper2 (10.1261/rna.061945.117), Vienna RNA 2.0 package (10.1186/1748-7188-6-26), RNA Framework (10.1007/978-1-0716-1307-8_5), and a modified deltaSHAPE analysis (10.1021/acs.biochem.5b00977). Multi-dimensional RNA structural probing data was analysed using the methods outlined in 10.1073/pnas.1619897114. Bootstrapping was performed using rna_structure function of the Basic Inference Engine for RNA structure (https://ribokit.github.io/Biers/) (MST data were processed using MO Affinity Analysis software (v 2.3; NanoTemper 565 Technologies). MST graphs were plotted using GraphPad Prism 8.4.3 software. Visualisations we made using python scikit-learn library (v 0.23.2), python NumPy library (v 1.19.2), python seaborn library (v 0.11.1) and python matplotlib library (v 3.3.2). |

For manuscripts utilizing custom algorithms or software that are central to the research but not yet described in published literature, software must be made available to editors and reviewers. We strongly encourage code deposition in a community repository (e.g. GitHub). See the Nature Portfolio guidelines for submitting code & software for further information.

## Data

Policy information about availability of data

All manuscripts must include a data availability statement. This statement should provide the following information, where applicable:

- Accession codes, unique identifiers, or web links for publicly available datasets
- A description of any restrictions on data availability
- For clinical datasets or third party data, please ensure that the statement adheres to our policy

Raw sequencing data are accessible through NCBI bioproject id PRJNA771368. HIV-1 sequences were downloaded from the Los Alamos HIV-1 sequence database (https://www.hiv.lanl.gov/content/index). In the main manuscript, conclusions are drawn from pooled data, but unpooled data are provided as extensive supplementary data. Pooled data is grouped into 4 structural classes. Unpooled data comprises 24 independently generated samples. DMS reactivities and Kdimer measurements Specifically, unpooled Kdimer and DMS reactivities are provided in table form. Also, for unpooled data, Kdimer and DMS reactivities are mapped to predicted dimer, predicted monomer, refined dimer and refined monomer strucrures. Structural predictions derived from DMS reactivities from unpooled data sets are provided as svg files. For two dimensional analysis, for each stage of the analysis (mutation rates, z-scores, convolution filtered data, detected helix and best helixes) are provided as pdf files.

# Field-specific reporting

Please select the one below that is the best fit for your research. If you are not sure, read the appropriate sections before making your selection.

☒ Life sciences          ☐ Behavioural & social sciences          ☐ Ecological, evolutionary & environmental sciences

For a reference copy of the document with all sections, see nature.com/documents/nr-reporting-summary-flat.pdf

# Life sciences study design

All studies must disclose on these points even when the disclosure is negative.

| | |
|---|---|
| Sample size | No statistical method was used to predetermine sample size. Data from 24 independent samples were pooled into 4 distinct structural classes for analysis. RNA-seq experiments to measure cellular packaging efficiencies were performed in duplicate. |
| Data exclusions | No data were excluded from the analysis |
| Replication | Conclusions were verified using gel based assays, replicated at a minimum twice |
| Randomization | Randomization was not performed. Rather, sample pooling was performed based on a PCA analysis. |
| Blinding | Blinding was not performed |

# Reporting for specific materials, systems and methods

We require information from authors about some types of materials, experimental systems and methods used in many studies. Here, indicate whether each material, system or method listed is relevant to your study. If you are not sure if a list item applies to your research, read the appropriate section before selecting a response.

## Materials & experimental systems

| n/a | Involved in the study |
|---|---|
| ☒ ☐ | Antibodies |
| ☐ ☒ | Eukaryotic cell lines |
| ☒ ☐ | Palaeontology and archaeology |
| ☒ ☐ | Animals and other organisms |
| ☒ ☐ | Human research participants |
| ☒ ☐ | Clinical data |
| ☒ ☐ | Dual use research of concern |

## Methods

| n/a | Involved in the study |
|---|---|
| ☒ ☐ | ChIP-seq |
| ☒ ☐ | Flow cytometry |
| ☒ ☐ | MRI-based neuroimaging |

## Eukaryotic cell lines

Policy information about cell lines

| | |
|---|---|
| Cell line source(s) | 293T cells, a gift from the Caliskan laboratory |
| Authentication | 293T cells were not verified |

| Mycoplasma contamination | 293T cell  lines is regularly tested for mycoplasm infection (monthly) |
| Commonly misidentified lines (See ICLAC register) | N/A |

