## [Peer Review File · Nature Structural & Molecular Biology]

Peer Review Information

Journal: Structural and Molecular Biology

Manuscript Title: Short- and long-range interactions in the HIV-1 5'UTR regulate genome dimerization and packaging

Corresponding author name(s): Professor Redmond Smyth

Editorial Notes:

Redactions – unpublished data	Parts of this Peer Review File have been redacted as indicated to maintain the confidentiality of unpublished data.
Redactions – confidential patient information	Parts of this Peer Review File have been redacted as indicated to maintain patient confidentiality.
Redactions – published data	Parts of this Peer Review File have been redacted as indicated to remove third-party material.
Redactions – reviewer opt-out	Parts of this Peer Review File have been redacted as indicated as we could not obtain permission to publish the reports of reviewer no. XX .
Reviewer comments in marked-up manuscript	In their review of the [first/second/third/...] version of this manuscript, reviewer no. XX added their comments to the manuscript file. These comments, excluding minor textual revisions, have been copied into this Peer Review File.

Reviewer Comments & Decisions:

Decision Letter, initial version:
--

27th Aug 2021

Dear Dr. Smyth,

Thank you for submitting your manuscript "Short- and long-range interactions in the HIV-1 5'UTR regulate genome dimerization and Pr55Gag binding". I apologize for the delay while we awaited the reports (copied below) from the 3 referees who evaluated the study. Unfortunately, after carefully considering their comments, we cannot offer to publish your manuscript, at least in its current form, in

Nature Structural & Molecular Biology.

You will see that while the referees acknowledge the potential interest of the findings, all 3 reviewers (each with expertise in RNA structure) query the functional relevance of the RNA structures defined in vitro, and also suggest additional assays to validate the novel FARS-seq approach that you describe—specifically, by demonstrating that that FARS-seq can predict known RNA secondary structures.

If you feel that further experimentation, analysis, and revisions would allow you to address the referees concerns in full, we would be prepared to consider an appeal of our decision, on the condition that no related work is published in the interim or has been accepted in our journal. Please feel free to contact me to discuss an appeal and potential revision. Please note that, until we have the opportunity to read the revised manuscript in its entirety, we cannot promise that it will be sent back for peer review.

I am very sorry we could not be more positive this time. I hope that you find the referees' comments useful in deciding how best to proceed.

With kind regards,

Beth

Beth Moorefield, Ph.D.
Senior Editor
Nature Structural & Molecular Biology

Reviewers' Comments:

Reviewer #1:

Remarks to the Author:

Ye et al. develop an innovative method, FARS-seq, which enables them to couple data about structure and function for HIV-1 5'UTR dimerization. The HIV-1 5'UTR is well-studied but still incompletely understood RNA which is known to play an important role in dimerization, packaging, reverse transcription and other critical HIV processes. However, its structure(s) has remained a contested question. Ye et al. not only offer new structural information but also begin to ask how structure and dimerization interact.

The paper is presented quite clearly, and brings to bear some remarkably high-information content

approaches to structure/function of a particularly complex RNA. Discovery of novel stems brings important new information to the problem. Supplemental methods and figures are thoroughly detailed. There are a few areas where we ask for clarification and a few additional studies we recommend, most involving data already collected by the authors which we believe are necessary to support their conclusions.

- Compensatory rescue mutations are the gold standard in RNA biochemistry for evaluating stems. Such analyses are missing from this study and we would request the authors conduct compensatory mutations to provide more conclusive evidence of their structures for new polyA-SL1 and PBS-SL1 stems. For example, C218G and G247C pushes the 3G equilibrium to favor the dimer, but their combination would be predicted to restore a monomer. Does that happen?
- It remains unclear how the authors decided to split their analysis of the probed RNA states. In figure 2, there are 4 distinct clusters in the PCA of DMS signal, which supports grouping 1G and 2G monomers together and 1G and 2G dimers together. However, it is unclear to this reviewer why in the rest of the main figures 1G/2G dimer and 3G monomer are the focus; what is happening with the 1G/2G monomer and 3G dimer? In fact, in supplementary figure 5, dimer 3G and monomer 1G/2G look more similar to each other than to the dimer 1G/2G or monomer 3G, respectively -- what is an explanation for this? Do the conclusions of structural difference on dimerization change if one uses the more fair comparisons of 1G/2G monomer v 1G/2G dimer and 3G monomer v 3G dimer?
- We are curious if there are previous studies or additional experiments the authors could conduct to measure actual viral function in cells? For example, are there any compensatory signals for the proposed stems in the extensive sequence alignments of HIV strains?
- The authors alluded to a conflict with 2020 Summers NMR structures due to sequence changes; we suggest comparison to other conflicting structures as well. For example, compare dimer to 2015 packaging signal structure by Keane et al (Keane et al, Science, 2015) — while Keane et al. see similar hairpins, there is a rearrangement in the central region interconnecting the PBS and SL1, SL2, SL3, involving a different proposed helix H1. How is this the model of Keane et al. reconciled with the new analysis?
- We would request an uncertainty analysis of the proposed secondary structures. For example, the bootstrapping-based approach proposed in M2-seq (Cheng et al. PNAS, 2017) would allow the authors to say if those 'central' helices (like what Keane et al. call "H1") are uncertain.

Minor comments

- Since it seems plausible that the HIV 5'UTR, even when separated by dimer identity, is heterogeneous, we suggest the authors to run their current dataset through an algorithm such as DREEM (Tomezsko et al, Nature, 2020) in order to see if alternative conformations are present.
- We would appreciate further comment on the 'Punctate signals' in Fig. S6c and d. The authors describe

them as tertiary contacts but there are alternative explanations that are not discussed. For example, could these be indicative of alternative states

(Codero, Das, PLOS comp bio, 2015) or, for the case of the dimer, further intermolecular interaction outside of the dimerization domain?

- In general, in the dimer dataset is there any indication of intermolecular interactions outside of the known dimerization site?

- Figure 6c is 3G monomer structure but shows only 2 G's

- Please add labels to mark the SD1 and AUG regions to figure 2.

- DMS reactivity coloring on secondary structures (e.g. Fig. 3) may not be friendly for red/green color blind readers.

- Is 'filtered Z-score' in upper triangles of 5b and 5e actually the M2-seq bootstrapping probability matrix?

- Fig 5 (monomer): the TAR-PBS interaction does not seem to have clear support from Z-matrix -- why was it included in the "best stems"? Why is it not in "all stems"?

Reviewer #2:

Remarks to the Author:

The manuscript entitled, "Short- and long-range interactions in the HIV-1 5'UTR regulate genome dimerization and Pr55Gag binding" re-analyzed the known RNA motif in HIV-1 5'UTR for dimerization using chemical based RNA structure probing. Overall experimental design raised several major concerns. The FARS-seq method used in this study lacks the validations of known RNA secondary structures. All the experiments were performed in vitro which did not represent the folding status in vivo. Therefore, the understanding on the function of DIS was quite weak. Overall, the novelty of this study is very weak.

Major comments:

1. The RNA motif for the dimerization is GC-rich. The method was built upon the mutation correlation between sequence mutations on the stem and DMS-induced mutations at the corresponding pairing partners. DMS is only capable of mutating A and C. Thus, the corresponding sequence mutations for the stem were only U and G. These limitations will not be possible to achieve high accuracy of RNA structure. SHAPE chemical might be a better choice for its advantages in obtaining the structural information of all four nucleotides.
2. The statistical model should be validated independently with known RNA secondary structures such as rRNAs' or tRNAs' structures.
3. DMS-MaP method relies on the RT-induced mutation rate which is quite low. For one DMS-modified nucleotide, only a small proportion of reads across this modified nucleotide contain the sequence mutation. It is not clear how the authors cope with the low RT-induced mutation efficiency in their statistical model. Also, DMS prefers A more than C. The authors should take account the modification preference into their statistical model.

4. In the methods, the authors performed the in vitro transcription on the PCR mutant libraries. Since the mixture of RNA mutants was subjected for physic separation to obtain dimers and monomers, it is highly likely that two RNAs with different mutants in their sequences could form into dimers which introduced other possibilities in interpreting the results.
5. The DMS reactivity difference between 1G/2G and 3G in figure 3d should be comprehensively assessed with all the conditions and 1G/2G/3G monomers/dimers. The variations of DMS reactivity between replications could serve as noise background.
6. In figure 4, the DMS reactivities seem not fit the structure models. The authors should provide a quantitative measurement.
7. In figure 5, the K(dimer) values ranged from -2.0 to +2.0. However, in the lines plots in figure 2c and d, the K(dimer) values ranged from -0.2 to 7. Why are the values so different in two figures?
8. In figure 5, the individual mutation should affect both dimer structure and monomer structure. The authors should provide evidence to show these mutations only affecting the dimer structure, but not the monomer structure.
9. In figure 7, the authors should design the mutations in TAR region as well.

Minor comments:

1. Accession number is "XXXXX" which the authors should provide the project or SRA numbers.
2. There is no functional analysis on the packaging. Therefore, the model in figure 7c is a bit misleading.

Reviewer #3:

Remarks to the Author:

In the manuscript Ye at al, the authors performed mutagenesis experiments on the 5' end of the HIV genome, structure probing and deep sequencing to determine features behind HIV dimerization. While the design is interesting and the study is important, most of the studies are done in vitro, using the 5' end fragment of HIV, making it difficult to know how physiologically relevant their findings are inside cells and in the full length virus genome. Additionally, many experiments in the manuscript lack critical controls, making it difficult to interpret the figures.

Figure 1

All figures shown are schematic diagram or model, is it possible to show some real data, for example, what is this the profile of mutational rate at each base for the error-prone and DMS treatment samples?

Figure 2

a) This figure shows that HIV dimerization favors high salt condition, however there is a lack of label of the size for the gel. The authors should confirm that the dimers in the gel is indeed due to dimerization initiation site (DIS) or the RNA-RNA interaction of other regions. Also, since author has done DMS sequencing, do they see changes in the mutational rate at DIS region between monomer and dimer?

b) This figure looks interesting, following the above question, is it possible to do further analysis, for example, does the mutations enriched in stem/loop region of SL1, and which positions have stronger effect for the dimerization?

Figure 2 c,d) and Figure 3d, In the introduction part, it says '1G transcripts expose the DIS for dimerization and sequester the 5' cap, whereas 2G and 3G variants conceal the DIS whilst exposing the cap to enhance translation', suggest 2G and 3G variant are similar, however, the author also claim that 1G and 2G variant are similar, it looks controversial, could the author clarify this?

Figure 3

Figure 3d looks interesting, but it is also arbitrary only show the DMS reactivity difference between 1G/2G dimer VS 3G monomer, is it possible also to show all the original reactivity or reactivity difference between monomer and dimer of 1G/2G, similar analysis between monomer and dimer of 3G?

Figure 4,

'pooled DMS reactivities as soft constraints to guide in silico RNA folding' may be not enough to predict long region RNA-RNA interaction, is it possible to validate some of the interaction by directed RNA-RNA interaction method? How accurate/well established is it to use DMS to constrain structure models accurately?

a, d) the stem region shows very high DMS reactivity, which does not make much sense. Could the authors show the entire reactivity profile of all regions, as this will help to evaluate how accurate is the structure folding method?

Besides, it is confusing to show the structure folding model for both monomer and dimer together, does the author assume that the structure folding are same between monomer and dimer? It is better show the model separately side by side.

Figure 5,

Two-dimensional RNA structure probing is a good idea to predict RNA base-pairing partners. I have a similar question as figure 1: could the authors provide the entire mutational profile showing the mutational rates at each base for the error-prone and DMS treatment samples? Also, the figure lacks the detailed cutoff settings and validation to show how robust of the analysis method.

Figure 6,

Is it possible to validate the structure model for the mutants by using DMS mutational rate or other method, this could help to understand how much the mutation affect the RNA structure and dimerization. Additionally, point mutation for the fragment may not reflect the dimerization of full length HIV1, is it possible to validate the full length HIV1 dimerization for some interesting mutants.

Figure 7,

The details are not clear, which one (1G/2G/3G or all together) were used here? Because mutations are one major reason cause dramatic RNA structure change, it is interesting to know whether the mutants in other region (for example in the mutants show in Figure 6) cause Pr55 binding.

Although we cannot publish your paper, it may be appropriate for another journal in the Nature Portfolio. If you wish to explore the journals and transfer your manuscript please use our manuscript transfer portal:

[REDACTED]

If you transfer to Nature journals or the Communications journals, you will not have to re-supply manuscript metadata and files. This link can only be used once and remains active until used.

All Nature Portfolio journals are editorially independent, and the decision on your manuscript will be taken by their editors. For more information, please see our [manuscript transfer FAQ](http://www.nature.com/authors/author_resources/transfer_manuscripts.html?WT.mc_id=EMI_NPG_1511_AUTHORTRANSF&WT.ec_id=AUTHOR) page.

Note that any decision to opt in to In Review at the original journal is not sent to the receiving journal on transfer. You can opt in to *In Review* at receiving journals that support this service by choosing to modify your manuscript on transfer. In Review is available for primary research manuscript types only.

** For Springer Nature Limited general information and news for authors, see <http://npg.nature.com/authors>.

Author Rebuttal to Initial comments

Reviewer 1

Ye et al. develop an innovative method, FARS-seq, which enables them to couple data about structure and function for HIV-1 5'UTR dimerization. The HIV-1 5'UTR is well-studied but still incompletely understood RNA which is known to play an important role in dimerization, packaging, reverse transcription and other critical HIV processes. However, its structure(s) has remained a contested question. Ye et al. not only offer new structural information but also begin to ask how structure and dimerization interact.

The paper is presented quite clearly, and brings to bear some remarkably high-information content approaches to structure/function of a particularly complex RNA. Discovery of novel stems brings important new information to the problem. Supplemental methods and figures are thoroughly detailed. There are a few areas where we ask for clarification and a few additional studies we recommend, most involving data already collected by the authors which we believe are necessary to support their conclusions.

#1 - Compensatory rescue mutations are the gold standard in RNA biochemistry for evaluating stems. Such analyses are missing from this study and we would request the authors conduct compensatory mutations to provide more conclusive evidence of their structures for new polyA-SL1 and PBS-SL1 stems. For example, C218G and G247C pushes the 3G equilibrium to favor the dimer, but their combination would be predicted to restore a monomer. Does that happen?

We acknowledge that compensatory rescue mutations are commonly used to validate RNA stems. In this study we did not attempt to do this because it is very difficult to cleanly design compensatory mutations on riboswitches, as mutations often have pleiotropic effects.

Take the suggestion by the reviewer that the combination of C218G and G247C would clearly restore the monomer. Both C218G and G247C favor the dimer by disrupting the novel PBS-SL1 interaction. But G247C additionally enhances dimerization by converting the SL1 internal loop into a bulge.

The outcome of their combination is unpredictable. Combining C218G and G247C would restore the PBS-SL1 interaction (favouring the monomer), but not restore the closure of the SL1 internal loop (which favours the dimer). As mentioned also in point #3 below, this is probably why the sequences of riboswitches are extremely highly conserved: it is very difficult to find sets of mutations that fully restore function because they have different effects on different structure adopted by the riboswitch.

In the revised manuscript, we will nevertheless test compensatory mutations for both the polyA-SL1 and PBS-SL1 stem, even though the effects might not be a clear cut as the reviewer suggests.

So far, we have tested one set of compensatory rescue mutations targeting the PBS-SL1 interaction. The C218U and G247A mutations both increase dimerization (as predicted by our data). Nevertheless, their combination does not restore the WT phenotype. This is likely because substituting the GC base pair for an AU base pair perturbs the PBS-SL1 interaction so that rescue is not possible.

#2 - It remains unclear how the authors decided to split their analysis of the probed RNA states. In figure 2, there are 4 distinct clusters in the PCA of DMS signal, which supports grouping 1G and 2G monomers together and 1G and 2G dimers together. However, it is unclear to this reviewer why in the rest of the main figures 1G/2G dimer and 3G monomer are the focus; what is happening with the 1G/2G monomer and 3G dimer?

In fact, in supplementary figure 5, dimer 3G and monomer 1G/2G look more similar to each other than to the dimer 1G/2G or monomer 3G, respectively -- what is an explanation for this? Do the conclusions of structural difference on dimerization change if one uses the more fair comparisons of 1G/2G monomer v 1G/2G dimer and 3G monomer v 3G dimer?

All four structural classes are interesting in their own right, which is why we included data for all samples for interested readers in the supplementary data. In an earlier draft of the manuscript we discussed the 1G/2G monomer and 3G dimer structures, but in the submitted version we removed this discussion to simplify the text for a general audience, and to meet guidelines for manuscript length.

In supplementary fig 5, it can be seen that “the 3G dimer folded into the canonical 5'UTR structure, whereas the 1G/2G monomer folded into the alternative structure containing the SL1-polyA interaction. Of note, both the 3G dimer and 1/2G monomer structures showed increased Shannon entropies in polyA, U5, SL1 and Gag when compared to the 1/2G dimer and 3G monomer structures. We interpret this to mean that the 3G dimer and 1/2G monomer populations are structurally less uniform, despite the fact that we selected for pure dimer and monomer structures in the native gels. This interpretation is supported by the dot plots which show a signal for SL1-polyA interaction in the 3G dimer, and a signal for SL1 in the 1/2G monomer population, even though these interactions were not predicted in the dominant fold. Altogether, our data support a novel structural rearrangement of the HIV 5'UTR leading to extensive base pairing between SL1 and the polyA-U5 region. This monomeric rearrangement appears to be favored in the 3G populations, whereas dimer structure is favoured in the 1G/2G population”

Also, in the supplementary data (2d-structure-inference), it can be seen that “in the 1G/2G monomer samples, the mutually exclusive polyA stem and the SL1-polyA interaction were both

detected, suggesting that a proportion of monomer structures converted back to the dimer after isolation (most likely during the probing reaction at 37C). As this was not evident in the 3G samples, it strengthens the idea that the 1/2G that the monomeric rearrangement appears to be favored in the 3G populations, whereas the 1G/2G population tends towards the dimer structure “.

In sum, the structures 1G/2G monomer and 3G dimer populations look ‘more similar’ to each other because the structural probing reactions are not instantaneous, and the two structures will eventually return to equilibrium even after physical isolation. This is not unexpected by us (and correctly identified by the reviewer). Therefore, the reasoning for selecting and comparing the 1G/2G dimer and 3G monomer is that these populations were the most dimeric and monomeric, respectively.

It is important to note that the conclusions of the manuscript *do not change* when analyzing 1G/2G monomer and 3G dimer samples; rather, their analysis further support one of our conclusions that 1G/2G samples are preferentially dimeric, and 3G samples are preferentially monomeric. Because the aims of our study were to define the monomer and dimer structures and show how they functionally interconvert, in our opinion, the ‘fairest’ and best samples to address these questions are the 1G/2G dimer and 3G monomer samples, as presented. We will clearly point this out in the revised manuscript.

Thanks to careful reading of the manuscript and interpretation of the data by reviewers #1, #2 and #3, we now realize that this important reasoning was not explicitly stated. In the revised manuscript, we will include a concise discussion of the 1G/2G monomer and 3G dimer classes (we proposed to do this in a supplementary text, or supplementary figure legends).

We will also include an additional supplementary figure to highlight the differences between the 4 structural classes for the 2d-structure-inference (similar to sup. fig. 5. which shows the comparison for the 1d-DMS probing)

#3 - We are curious if there are previous studies or additional experiments the authors could conduct to measure actual viral function in cells? For example, are there any compensatory signals for the proposed stems in the extensive sequence alignments of HIV strains?

To search for compensatory signals in sequence alignments is a fantastic idea in principle. In reality, the relevant regions (polyA, PBS and SL1) of the HIV-1 5'UTR are extremely well conserved (>90%), making it difficult to identify true compensatory mutations. Of course, one interpretation *is that this extreme conservation results from constraints induced by the RNA structural switch described in our work.*

Below, we show results from approximately 800 curated sequences downloaded from the Los Alamos HIV sequence database.

polyA-SL1 interaction

polyA sequence conservation, modal sequence

```

| Modal Sequence: 5' |
+-----+
| position | 80 | 81 | 82 | 83 | 84 | 85 | 86 | 87 | 88 | 89 | 90 | 91 | 92 | 93 | 94 | 95 | 96 | 97 | | |
| base | C | T | T | G | C | C | T | T | G | A | G | T | G | C | T | T | C | A |
| a | 1 | 1 | 1 | 0 | 0 | 0 | 0 | 0 | 9 | 887 | 0 | 0 | 6 | 0 | 7 | 20 | 76 | 867 |
| u | 0 | 890 | 889 | 1 | 0 | 5 | 877 | 885 | 0 | 0 | 0 | 0 | 884 | 0 | 3 | 860 | 804 | 308 | 0 |
| c | 890 | 0 | 0 | 0 | 0 | 887 | 883 | 6 | 1 | 1 | 0 | 0 | 0 | 0 | 884 | 1 | 46 | 485 | 1 |
| g | 0 | 0 | 1 | 1 | 889 | 0 | 0 | 0 | 1 | 877 | 0 | 0 | 889 | 3 | 881 | 0 | 3 | 1 | 2 | 0 |
| count | 890 | 890 | 889 | 889 | 887 | 883 | 877 | 885 | 3 | 881 | 0 | 3 | 1 | 1 | 2 | 0 |
| total | 891 | 891 | 891 | 890 | 887 | 888 | 883 | 887 | 887 | 887 | 889 | 887 | 887 | 887 | 871 | 871 | 871 | 868 |
| modal conservation | 99% | 99% | 99% | 99% | 100% | 99% | 99% | 99% | 99% | 98% | 100% | 100% | 99% | 99% | 99% | 98% | 92% | 55% | 99% |
+-----+

```

SL1 sequence conservation, modal sequence

```

| Modal Sequence: 3' |
+-----+
| position | 253 | 254 | 255 | 256 | 257 | 258 | 259 | 260 | 261 | 262 | 263 | 264 | 265 | 266 | 267 | 268 | 269 | 270 |
| base | T | G | A | A | G | C | G | C | G | C | A | C | G | G | C | A | A | G |
| a | 0 | 2 | 651 | 687 | 9 | 0 | 0 | 0 | 5 | 0 | 653 | 0 | 170 | 5 | 0 | 685 | 685 | 3 |
| u | 688 | 0 | 2 | 0 | 1 | 15 | 0 | 2 | 0 | 0 | 6 | 4 | 0 | 0 | 0 | 0 | 0 | 0 |
| c | 0 | 0 | 1 | 0 | 7 | 667 | 0 | 685 | 0 | 684 | 1 | 681 | 4 | 3 | 685 | 0 | 0 | 0 |
| g | 0 | 684 | 0 | 0 | 664 | 0 | 687 | 0 | 681 | 0 | 24 | 0 | 511 | 677 | 0 | 0 | 0 | 682 |
| count | 688 | 684 | 651 | 687 | 664 | 667 | 687 | 685 | 681 | 684 | 653 | 681 | 511 | 677 | 685 | 685 | 685 | 682 |
| total | 688 | 686 | 654 | 687 | 681 | 682 | 687 | 687 | 686 | 684 | 684 | 685 | 685 | 685 | 685 | 685 | 685 |
| modal conservation | 100% | 99% | 99% | 100% | 97% | 97% | 100% | 99% | 99% | 100% | 95% | 99% | 74% | 98% | 100% | 100% | 100% | 99% |
+-----+

```

SL1-polyA interaction conservation

```

| Conservation of Interaction |
+-----+
| position1 | 80 | 81 | 82 | 83 | 84 | 85 | 86 | 87 | 88 | 89 | 90 | 91 | 92 | 93 | 94 | 95 | 96 | 97 | |
| base1 | C | T | T | G | C | C | T | T | G | A | G | T | G | C | T | T | C | A |
| interaction | | | | | | | | | | - | | | | | | | | | |
| base2 | G | A | A | C | G | G | C | A | C | G | C | G | C | G | A | A | G | T |
| position2 | 270 | 269 | 268 | 267 | 266 | 265 | 264 | 263 | 262 | 261 | 260 | 259 | 258 | 257 | 256 | 255 | 254 | 253 |
| interacting | 679 | 682 | 681 | 681 | 674 | 515 | 0 | 672 | 671 | 0 | 686 | 684 | 678 | 662 | 679 | 590 | 617 | 687 |
| not interacting | 4 | 1 | 2 | 1 | 8 | 168 | 679 | 9 | 10 | 684 | 0 | 1 | 2 | 17 | 7 | 63 | 68 | 0 |
| conservation | 99% | 99% | 99% | 98% | 75% | 0% | 98% | 98% | 0% | 100% | 99% | 99% | 97% | 98% | 90% | 90% | 100% |
+-----+

```

These regions are highly conserved, with some polymorphism at positions 95/96 in the polyA. The modal sequence from this alignment differs slightly from the NL43 construct used in our study. Nevertheless, it is very interesting to see that the polymorphisms maintain the polyA-SL1 interaction (and even extend it).

There is one polymorphic position 265 in SL1. The G to A mutation does not conserve the interaction with its partner directly opposite, but this mutation is not disruptive as it can instead base pair with the unpaired U86 opposite.

PBS-SL1 interaction

PBS sequence conservation, modal sequence

```

-----+-----
| Modal Sequence: 5' |
-----+-----
| position | 210 | 211 | 212 | 213 | 214 | 215 | 216 | 217 | 218 | 219 | 220 | 221 | 222 | 223 | 224 |
| base | A | A | G | T | A | A | A | C | C | A | G | A | G | G |
| a | 685 | 687 | 4 | 115 | 478 | 344 | 638 | 638 | 0 | 0 | 552 | 4 | 682 | 0 | 136 |
| u | 0 | 0 | 0 | 451 | 5 | 4 | 13 | 3 | 0 | 0 | 0 | 1 | 0 | 0 | 5 |
| c | 2 | 0 | 0 | 7 | 0 | 0 | 0 | 0 | 686 | 687 | 1 | 0 | 0 | 0 | 3 |
| g | 0 | 0 | 681 | 113 | 204 | 338 | 35 | 46 | 0 | 0 | 133 | 679 | 2 | 685 | 543 |
| count | 685 | 687 | 681 | 451 | 478 | 344 | 638 | 638 | 687 | 552 | 679 | 682 | 685 | 543 |
| total | 687 | 687 | 685 | 686 | 687 | 686 | 686 | 687 | 686 | 687 | 686 | 684 | 684 | 685 | 687 |
| modal conservation | 99% | 100% | 99% | 65% | 69% | 50% | 93% | 92% | 100% | 100% | 80% | 99% | 99% | 100% | 79% |
-----+-----

```

SL1 sequence conservation, modal sequence

```

-----+-----
| Modal Sequence: 3' |
-----+-----
| position | 241 | 242 | 243 | 244 | 245 | 246 | 247 | 248 | 249 | 250 | 251 | 252 | 253 | 254 | 255 |
| base | G | A | C | T | C | G | G | C | T | T | G | C | T | G | A |
| a | 2 | 685 | 0 | 0 | 1 | 0 | 0 | 0 | 0 | 0 | 0 | 2 | 651 |
| u | 0 | 0 | 1 | 688 | 4 | 2 | 2 | 0 | 688 | 688 | 0 | 2 | 688 | 0 | 2 |
| c | 0 | 0 | 687 | 0 | 682 | 0 | 0 | 688 | 0 | 0 | 0 | 682 | 0 | 0 | 1 |
| g | 686 | 3 | 0 | 0 | 1 | 686 | 686 | 0 | 0 | 0 | 688 | 3 | 0 | 684 | 0 |
| count | 686 | 685 | 687 | 688 | 682 | 686 | 686 | 688 | 688 | 688 | 688 | 682 | 688 | 684 | 651 |
| total | 688 | 688 | 688 | 688 | 688 | 688 | 688 | 688 | 688 | 688 | 688 | 687 | 688 | 686 | 654 |
| modal conservation | 99% | 99% | 99% | 100% | 99% | 99% | 99% | 100% | 100% | 100% | 100% | 100% | 99% | 100% | 99% |
-----+-----

```

The SL1 sequence is almost universally conserved. We identified sequence variants in the PBS region, resulting in some structural variation, but all are predicted to form hairpins of similar stability. The figure below shows the predicted structure of the top 116, 71, 50 sequence variants.

In the revised manuscript, we will also include additional data showing that the structures we detect *in vitro* do exist in cells, and moreover, that they are functionally relevant.

First, we performed traditional in cell DMS-MaP-seq on mutants targeting the PBS-SL1 and polyA-SL1 interaction. We show that dimer promoting mutants fold into the dimer structure, whereas monomer promoting fold into the monomer structure.

In cell DMS-MaP-seq experimental protocol; RNA structures detected in cells

WT

84A85A
(dimer mutant)

86G89C
(monomer mutant)

Second, we performed competition experiment on mutants targeting the PBS-SL1 and polyASL1 interaction. Here, we used RNA-seq to quantify the mutants in cellular and viral RNA.

Competition experiment protocol

Calculations of packaging efficiency show that the monomer promoting mutants are defective in genome packaging compared to dimer promoting mutants

In a more stringent experiment, we competed the four mutants that either stabilised or disrupted the PBS-SL1 and polyA-SL1 interaction against wild-type HIV.

The results clearly show that dimer promoting mutants are packaged similarly or better than wild-type HIV-1. On the other hand, monomer promoting mutants are clearly defective in packaging compared to WT HIV-1.

Altogether, these data strongly support our model and prove that our *in vitro* structural data is also relevant in cells.

#4 - The authors alluded to a conflict with 2020 Summers NMR structures due to sequence changes; we suggest comparison to other conflicting structures as well. For example, compare dimer to 2015 packaging signal structure by Keane et al (Keane et al, Science, 2015) — while Keane et al. see similar hairpins, there is a rearrangement in the central region interconnecting the PBS and SL1, SL2, SL3, involving a different proposed helix H1. How is this the model of Keane et al. reconciled with the new analysis?

The structure of the 5'UTR is a hotly contested subject, and several high profile studies have put forward different structural models.

As the reviewer is aware, the Summer's lab uses *in vitro* NMR structural approaches to model the structure of the HIV-1 5'UTR (PNAS 2020, Science 2020, PNAS 2016, PNAS 2016, Science 2015, Science 2011). Similarly, the Week's lab has used an *in vitro* / *ex viro* / *in viro* structural approach (SHAPE) to do the same (Nat. Methods 2014, Nature 2009, PLoS Biol 2008).

Many of these 5'UTR structures are mutually exclusive - a fact that has so far remained unexplained. One of the satisfying results of our work is that we are able to better understand why these models are different, and why they all might be correct in their own right. That is, our data point to SL1 being in a metastable state where single point mutations can remodel the RNA structure (see Figure 6b, and discussion in manuscript). We can assume that only very slight differences in experimental conditions would be needed to shift the structure from one conformation to the next. We can obtain this insight because we derive *functional* information at the same time as probing RNA structure.

We did not want to cover the finer points of this debate in all its details because NSMB is a general audience journal. Figure 6, coupled with our extensive supplementary data, should allow the specialized reader to draw their own conclusions.

For the reviewer's own interest, we have plotted the functional data from our study (Figure 6) onto the structure of the H1 region.

H1 region; from Keane et al. Science. 2015 plotted with FARS-seq functionally profiling data from Figure 6

Although the FARS-seq data should never be interpreted on a single structure (but must take into account other possible structures), the reviewer may appreciate that there is no *strong* support for the H1 helix in our data set.

We would also like to reemphasize that the H1 helix in the three-way-junction model involves the destruction of SL2. Indeed, on I362 we stated, “In some structures, extensions to SL1 even lead to the complete disruption of SL2⁶¹. Here, we found no direct evidence that SL1 is in an extended form in dimeric RNA and consistently observe signals for SL2 as a short imperfect stem containing a bulged adenosine.”

These facts together argue that H1 is absent in our data set.

Finally, we would like to point out that, to our knowledge, direct detection of H1 by NOE has only been seen in the Keane et al. Science 2015 paper which used a short fragment of HIV-1 that was deleted for PBS and polyA - both of these domains we show are important for the structure and function of the HIV-1 5'UTR. Moreover, extensive phylogenetic surveys conducted by other groups reveal strong support for SL2 (see in our study) rather than the H1 containing three-way-junction (see Viruses 2016 Jul; 8(7): 200 [pubmed id 27455303])

#5 - We would request an uncertainty analysis of the proposed secondary structures. For example, the bootstrapping-based approach proposed in M2-seq (Cheng et al. PNAS, 2017) would allow the authors to say if those ‘central’ helices (like what Keane et al. call “H1”) are uncertain.

The M2-seq paper detects helices using the neural-network inspired M2-net algorithm, which we implemented in our work. A boot-strapping approach is not described, and the M2-net algorithm is recommended for structural analysis.

We presume the reviewer is referring to bootstrapping implemented in Biers (<https://ribokit.github.io/Biers/rnastructure/#rnastructure-M2seq>) or described in <https://pubs.acs.org/doi/full/10.1021/bi200524n>.

This is a good idea. In the revised manuscript, we will implement a bootstrapping approach to assess the certainty of the helices ‘filled in’ by the RNA structure prediction i.e. those not directly detected in the M2-net algorithm.

Minor comments

#6 - Since it seems plausible that the HIV 5'UTR, even when separated by dimer identity, is heterogeneous, we suggest the authors to run their current dataset through an algorithm such as DREEM (Tomezsko et al, Nature, 2020) in order to see if alternative conformations are present.

The reviewer is correct that even when separated by dimer identity the sample may retain some heterogeneity.

In our answer to comment #2, sample heterogeneity is already visible in the 1G/2G monomer and 3G dimer samples so algorithmic approaches, such as DREEM, are not required in this case, even though the suggestion by the reviewer is a great one.

#7 - We would appreciate further comment on the 'Punctate signals' in Fig. S6c and d. The authors describe them as tertiary contacts but there are alternative explanations that are not discussed. For example, could these be indicative of alternative states

(Codero, Das, PLOS comp bio, 2015) or, for the case of the dimer, further intermolecular interaction outside of the dimerization domain?

We thank the reviewer for making this point.

When we first observed the punctate signals in Fig S6c and d, we also thought that they represented alternative folds. We tried to identify these, but found no structures, indeed there are no possible Watson-Crick stems at these positions. Our best current explanation is that they represent tertiary contacts. We do not think they represent further intermolecular interactions outside of the dimerization domain, as these would not appear as signals in our 2d data.

However, the reviewer is completely correct that we have not ruled out alternative folds. For example, it is possible that these regions fold into structures containing non-Watson Crick base pairs.

In the revised manuscript we will carefully outline alternative explanations for these data.

#8 - In general, in the dimer dataset is there any indication of intermolecular interactions outside of the known dimerization site?

Our experiments are not designed to detect inter-molecular interactions, and we can only observe intra-molecular RNA structure in the 2-dimensional analysis. Therefore, we cannot generate *direct* information on inter-molecular interactions.

Nevertheless, on l331 of the manuscript, we state "Surprisingly, annealing the cPBS₁₈₂₋₁₉₉ oligo also led to the formation of a higher, likely tetrameric molecular species. The TAR apical loop contains a 10-nucleotide palindromic sequence that has been proposed to dimerize by a TAR-TAR kissing interaction analogous to the one used by SL1²⁶. We therefore postulate that cPBS₁₈₂₋₁₉₉ disrupts the TAR-PBS interaction detected by multi-dimensional structural probing, allowing TAR to dimerize independently of SL1"

We did not want to place too much emphasis on this exact point, because the 2-dimensional structural information supporting this interaction is indirect and “relatively weak”.

Nevertheless, our functional data does indicate that TAR plays an unexpected role in dimerization, especially considering it is widely assumed to fold into a stable stem loop structure independent of the rest of the 5'UTR. In Figure 6a, the mutation G27C provides independent validation for the role of TAR in dimerization.

#9 - Figure 6c is 3G monomer structure but shows only 2 G's

This will be corrected in the revised manuscript.

#10 - Please add labels to mark the SD1 and AUG regions to figure 2.

This will be done in the revised manuscript.

#11 - DMS reactivity coloring on secondary structures (e.g. Fig. 3) may not be friendly for red/green color blind readers.

This is an important issue that is already addressed with the current colour scheme. DMS reactivities are shown on a continuous yellow-red spectrum using the “YlOrRd” colormap in Matplotlib. Colourblind readers can still perceive the reactivity changes using this palette, as can be appreciated in the colourblind filtered figure below.

DMS reactivity legend, with deuteranopia filter

#12 - Is ‘filtered Z-score’ in upper triangles of 5b and 5e actually the M2-seq bootstrapping probability matrix?

No, the filtered Z-score represents Z-scores after I542 “thresholding and applying a convolution filter enhance cross diagonal features”.

In the revised manuscript we will elaborate on these details in the material and methods. We will also improve the figure legends.

#13 - Fig 5 (monomer): the TAR-PBS interaction does not seem to have clear support from Z-matrix -- why was it included in the "best stems"? Why is it not in "all stems"?

As mentioned in comment #7, the reviewer is correct that the TAR-PBS interaction has a weak signal in the 2-dimensional analysis.

The signal is present 'all stem', but we erroneously omitted the surrounding box - we will correct this in the revised manuscript.

Reviewer 2

The manuscript entitled, “Short- and long-range interactions in the HIV-1 5’UTR regulate genome dimerization and Pr55Gag binding” re-analyzed the known RNA motif in HIV-1 5’UTR for dimerization using chemical based RNA structure probing. Overall experimental design raised several major concerns. The FARS-seq method used in this study lacks the validations of known RNA secondary structures. All the experiments were performed *in vitro* which did not represent the folding status *in vivo*. Therefore, the understanding on the function of DIS was quite weak. Overall, the novelty of this study is very weak.

We would argue that the FARS-seq method is well validated because it identifies and explains many features of the HIV-1 genome, which is one of the most intensively studied stretches of nucleotides in biology. Our dimer structure matches the ‘canonical’ structure identified by many laboratories, providing an internal control for the FARS-seq method. Moreover, in the original manuscript our model was extensively tested by mutagenesis. Over 20 different mutants supported our model and the FARS-seq method perfectly.

We do not agree that the understanding on the function of DIS was quite weak simply because the study was conducted *in vitro*. Dimerization is difficult to study in cells because key dimerization structures are involved with other functions of viral replication. Repeatedly, critical biological insights on dimerization have been derived from *in vitro* studies, which allow the investigation of dimerization signals in the absence of co-founding factors e.g. (PNAS 2020, Science 2020, Science 2015, Science 2011).

To address these critiques, we have now performed additional validation experiments *in vitro* and in cells.

Major comments:

#1 - The RNA motif for the dimerization is GC-rich. The method was built upon the mutation correlation between sequence mutations on the stem and DMS-induced mutations at the corresponding pairing partners. DMS is only capable of mutating A and C. Thus, the corresponding sequence mutations for the stem were only U and G. These limitations will not be possible to achieve high accuracy of RNA structure. SHAPE chemical might be a better choice for its advantages in obtaining the structural information of all four nucleotides.

We state in the manuscript at l82 that “performing DMS structural probing on mutational libraries enables the direct detection of RNA stems (Fig. 1f)^{43,44}. More specifically, when mutations within the library occur with RNA stems, it creates unpaired nucleotides at the position facing the mutation.

These newly unpaired residues become more accessible for DMS modification leading to correlated mutations in the sequencing data.”

Whilst DMS only reacts with A and C residues, the 2-dimensional signal in our analysis comes from the combination of mutagenesis + DMS. Both AU and CG base pairs contain an A or a C.

Take an AU base pair: the RNA library will contain molecules with mutations to the A and U. When the U is mutated, it leads to increased reactivity at the opened A residue. With a CG base pair, when the G is mutated, it leads to increased reactivity at the opened C residue. The resulting correlations are detected as a signal in our statistical analysis.

In conclusion, our 2-dimensional probing *allows* the direct detection of all Watson-Crick base pairs in a stem (although not GU Wobble base pairs). SHAPE reagents are neither necessary, nor beneficial, as SHAPE reagents are less effective at inducing RT induced mutation rate compared to DMS.

As the reviewer has missed this important principle, in the revised manuscript we will take great care to fully describe the benefits of our approach, and expand the description of the technique (in the material and methods).

#2 - The statistical model should be validated independently with known RNA secondary structures such as rRNAs' or tRNAs' structures.

We would argue that the method is well validated as it can robustly detect known RNA structures in the HIV-1 genome - one of the most extensively characterized RNA sequences in history.

To address this critique, we have performed classical in solution RNA structural probing (e.g. in the absence of functional selection) on several key mutants *in cells*. Here, we show that dimer promoting mutants fold into the dimer structure, whereas monomer promoting fold into the monomer structure. This provides independent validation that FARS-seq provides useful and correct structural insights.

WT

In cell DMS-MaP-seq experimental protocol

84A85A
(dimer mutant)

86G89C
(monomer mutant)

#3 - DMS-MaP method relies on the RT-induced mutation rate which is quite low. For one DMSmodified nucleotide, only a small proportion of reads across this modified nucleotide contain the sequence mutation. It is not clear how the authors cope with the low RT-induced mutation efficiency in their statistical model. Also, DMS prefers A more than C. The authors should take account the modification preference into their statistical model.

The reviewer is concerned that the DMS-MaP method has a low signal to noise due to a low RT-induced mutation rate.

On the contrary, our data has exceptional signal to noise, as can be seen in Sup. Fig. 2 showing a 3.4 fold and 7.8 fold increase in mutation rates at A and C residues, respectively. This is more than sufficient to obtain high quality quantitative 1-dimensional DMS reactivities.

As described in the material and methods, the 2-dimensional structural analysis uses a Z-score statistic which measures how much the DMS signal at each nucleotide is enhanced over the mean at that position, normalized to the standard deviation at that position.

It is true that increasing the RT-induced mutation would enhance this statistical signal. This

could be done by using alternative reverse transcriptase enzymes, such as TGIRT or Marathon-RT. We did not pursue this optimization because the data quality was already good enough to make strong inferences about RNA structure in the samples. For example, Figure 6b it is possible to visually identify known HIV-1 motifs - and these signals can even be seen in the plots of the raw mutation rates before statistical treatment (see supplementary figures in 2d-structure- inference folder).

To convince the reviewer that we have good signal to noise across the RNA, in the revised manuscript we will supply additional supplementary data giving the position-wise mutation rates for each sample and compute the signal-to-noise ratio for each position

--

We do not know where the reviewer gets their assertion that DMS prefers A more than C. In our data set, it is true that we see higher mutation rates (reactivities) at A residues, but this is simply explained that As are more unpaired than Cs in our tested RNA molecule.

In the material and methods, we state “DMS reactivities were calculated from ShapeMapper2 mutation rates using 90% Winsoring⁴¹”. Thus, we use a published and standard framework tool to calculate DMS reactivities, using a widely accepted method of normalization.

We do not see the advantages of normalization using the non-standard method suggested by the reviewer. Rather, treating A and C residues differently are likely to introduce biases into the analysis.

#4 - In the methods, the authors performed the in vitro transcription on the PCR mutant libraries. Since the mixture of RNA mutants was subjected for physic separation to obtain dimers and monomers, it is highly likely that two RNAs with different mutants in their sequences could form into dimers which introduced other possibilities in interpreting the results.

The reviewer is concerned about the possibility that two RNAs, each with mutations that decrease dimerization with the ‘wildtype’, could dimerize with one another.

This is a theoretical possibility. However, the likelihood for this to happen is incredibly low. In fact, it is so low, that it would not affect the quantitative readout (estimation of change in binding affinity).

a) Statistical argument:

In the manuscript (first section of the results) we state the mutation frequencies in the RNA libraries: “*In the absence of DMS, mutation frequencies in the mutated [...] library were 5.4×10^{-3} [...]*”.

Therefore, a mutation at a dimerization-affecting site occurs with probability $p_1 \sim 5.4 \times 10^{-3} = 0.005$. The probability of a compensatory mutation is $p_2 \sim 5.4 \times 10^{-3} = 0.005$. Whereas the probability that no compensatory mutation ‘wildtype’ occurs in the RNA is $p_3 \sim 1 - 5.4 \times 10^{-3} = 0.995$.

Consequently in the pool of RNA, the signal that relates to dimerization with the ‘wildtype’ is about **199 times stronger** that the signal that relates to the dimerization of two mutant RNAs. E.g. $199 = 0.995 \cdot 0.005 / (0.005 \cdot 0.005)$.

b) Experimental argument:

On an experimental note, the fact that the strongest dimerization signal is the 6-nucleotide palindromic sequence (Fig 2c, 2c), where many trans-complementary mutations are described in the literature, provides additional evidence that the potential noise introduced by this theoretical possibility is virtually non-existent.

We will make a statement in the revised manuscript to highlight this fact about our analysis.

#5 - The DMS reactivity difference between 1G/2G and 3G in figure 3d should be comprehensively assessed with all the conditions and 1G/2G/3G monomers/dimers. The variations of DMS reactivity between replications could serve as noise background

This comment is similar to point #2 from reviewer 1. See above for a detailed discussion of the other samples.

In the revised manuscript, we will provide the additional analyses requested.

#6 - In figure 4, the DMS reactivities seem not fit the structure models. The authors should provide a quantitative measurement.

We disagree with the reviewer. DMS reactivities for the monomer and dimer fit perfectly the structural models.

As stated I210 in the manuscript, “The SL1-PolyA reorganization was well supported by the DMS reactivity changes (compare **Fig. 4a** and **Fig. 4d**). In particular, the unpaired adenosine 263A in the SL1 loop, which was highly reactive in the dimer structure, became unreactive in the monomer due to base pairing with U87. Similarly, nucleotides C84 and C85 in polyA, which were reactive in the dimer structure, became unreactive in the monomer due to base pairing with 265G and 266G in the SL1 stem. Finally, A89 in the stem of polyA, which was unreactive in the dimer structure, became unpaired in monomer structure and reactive to DMS.”

We believe that reviewer #2, like reviewer #3, has misunderstood the figure. Fig 4 shows the reactivities for the **both** monomer and dimer populations on **both** structural models, using differently coloured half circles to represent the monomer and dimer reactivity data.

We presented the data this way because we wanted to highlight positions where DMS reactivities change, and to emphasize that these reactivity changes do explain the structural models.

Since reviewer #2 and #3 found this representation confusing, we will follow the suggestion of reviewer #3 to present only the dimer data on the dimer structure, and only the monomer data on the monomer structure. We will move the hybrid figure to the supplementary data.

#7 - In figure 5, the K(dimer) values ranged from -2.0 to +2.0. However, in the lines plots in figure 2c and d, the K(dimer) values ranged from -0.2 to 7. Why are the values so different in two figures?

The K(dimer) data ranges from -0.2 to +7, as shown in Figure 2.

Figure 6 uses a blue-white-red colour scheme to represent mutations with a negative (red), neutral (white), and positive effect (blue). This representation must be symmetrical to avoid introducing perceptual bias.

The values between the two figures *are the same*, but the colour of value above 2 are capped. If we were to represent values to -7 to +7 most information would be visually lost, and the figure would not be useful to the readers.

In the revised manuscript we will expand the figure legend to clarify this point.

#8 - In figure 5, the individual mutation should affect both dimer structure and monomer structure. The authors should provide evidence to show these mutations only affecting the dimer structure, but not the monomer structure.

We do not fully understand the reviewer's comment.

Is the reviewer referring to Figure 6, and not Figure 5, where we validate our structural models using point mutations?

We have already designed and tested many mutations, including many that specifically target the monomer and dimer structures. In all cases, the mutations had the predicted effect. In a few instances, we even assessed the effect of different mutations at a single position (e.g. C110, C111, A235, A239). Again, these mutations behaved as predicted according to our structural models.

We assume the reviewer comment is similar to point 6 of reviewer 3, who requests additional structural validation of mutants.

In the revised manuscript, we will now include additional experiments support validating the structures proposed in Figure 6b. To do this, we performed traditional in solution chemical probing on two mutants that we predicted would fold the RNA into the 3IL and 3WJ structures. The results of these experiments reveal very subtle changes in the SL1 structure exactly as predicted by our hypothesis.

#9 - In figure 7, the authors should design the mutations in TAR region as well.

We must apologize as it is unclear to us which mutants the reviewer is referring to, and what experiments the reviewer is requesting.

In Figure 6, we already included one TAR mutation (see panel A).

Possibly the reviewer is referring to the Figure 7a, where we hypothesize a tetrameric species due to a TAR-TAR inter-molecular interaction. As stated in point #7 to reviewer 1, whilst potentially of biological interest, a TAR-TAR interaction is not something we claim, nor want

to emphasize in the manuscript, as it is not a phenomenon directly observable in the FARS-seq data.

Minor comments:

#10 - Accession number is "XXXXX" which the authors should provide the project or SRA numbers.

All data (current and revised) will be accessible through the SRA.

#11 There is no functional analysis on the packaging. Therefore, the model in figure 7c is a bit misleading.

We will remove the word 'packaging' from the figure

Reviewer 3

In the manuscript Ye et al, the authors performed mutagenesis experiments on the 5' end of the HIV genome, structure probing and deep sequencing to determine features behind HIV dimerization. While the design is interesting and the study is important, most of the studies are done *in vitro*, using the 5' end fragment of HIV, making it difficult to know how physiologically relevant their findings are inside cells and in the full length virus genome. Additionally, many experiments in the manuscript lack critical controls, making it difficult to interpret the figures.

The critique that most of the studies are done *in vitro* is similar to that raised by reviewer #2. We have convincingly addressed this point by performing structural validation experiments on full length HIV RNA in cells (see data in point #3 of reviewer one).

We disagree that many experiments lack critical controls, and it is not clear from the critique what controls are missing that prevent interpretation of the figures. Our revised manuscript fully addresses all of the point-by-point concerns raised below.

#1 - Figure 1

All figures shown are schematic diagram or model, is it possible to show some real data, for example, what is this the profile of mutational rate at each base for the error-prone and DMS treatment samples?

Figure 1 is intended as a schematic diagram to explain the study design. We do not think it appropriate, nor possible, to place 'real' data here.

In Sup. Fig 1 and Fig 2, the reviewer can already find the global and base specific mutation rate for the mutant libraries and DMS treated samples.

Furthermore, at l102 in the main text we state "In the absence of DMS, mutation frequencies in the mutated and non-mutated control library were 5.4×10^{-3} and 3.7×10^{-4} , respectively, and the mutational interference libraries with (sic) a signal to noise $D_m(i) > 2$ (**Sup. Fig. 1**; details in **Sup. Methods**). In the DMS treated samples, we saw an additional increase in mutation frequencies at the expected A and C residues indicating a successful modification of RNA (3.4- and 7.8-fold increase at C and A, respectively, **Sup. Fig. 2**)."

In the revised manuscript we will additionally provide position-wise mutation rates for each sample.

#2 - Figure 2

a) This figure shows that HIV dimerization favors high salt condition, however there is a lack of label of the size for the gel.

Figure 2a is a native agarose gel, and it is not possible to generate accurate size standards.

The labelling of the gel follows widely used conventions. If requested by the editor or reviewer, we are of course happy to provide uncropped images of gels.

The authors should confirm that the dimers in the gel is indeed due to dimerization initiation site (DIS) or the RNA-RNA interaction of other regions.

The dimerization properties of RNA fragments containing the HIV 5'UTR have been extensively characterized in 10's (if not 100's) of experimental studies. Under conditions used in our study, the only region known to be required for dimerization is SL1.

We refer the reviewer to El-Wahab and Smyth et al. Nature Comm (2014), where supplementary Fig 5 contains many examples of SL1 deletion and point mutants. These mutants clearly show that the primary dimerization site is SL1, and dimerization can be completely disrupted by mutants to SL1.

Also, since author has done DMS sequencing, do they see changes in the mutational rate at DIS region between monomer and dimer?

Yes we do.

We refer the reviewer to Fig 4a and 4d which shows DMS reactivity changes in SL1 supporting the structural models proposed.

At l210 in the text we state "The SL1-PolyA reorganization was well supported by the DMS reactivity changes (compare **Fig. 4a** and **Fig. 4d**). In particular, the unpaired adenosine 263A in the SL1 loop, which was highly reactive in the dimer structure, became unreactive in the monomer due to base pairing with U87. Similarly, nucleotides C84 and C85 in polyA, which were reactive in the dimer structure, became unreactive in the monomer due to base pairing with 265G and 266G in the SL1 stem. Finally, A89 in the stem of polyA, which was unreactive in the dimer structure, became unpaired in monomer structure and reactive to DMS."

In the revised manuscript, we will provide additional supplementary tables with raw mutation frequencies for each sample at each position.

b) This figure looks interesting, following the above question, is it possible to do further analysis, for example, does the mutations enriched in stem/loop region of SL1, and which positions have stronger effect for the dimerization?

These questions were extensively covered in the manuscript. We refer the reviewer to Figure 6, and the extended data tables in the supplementary data.

Figure 2 c,d) and Figure 3d, In the introduction part , it says '1G transcripts expose the DIS for dimerization and sequester the 5' cap, whereas 2G and 3G variants conceal the DIS whilst exposing the cap to enhance translation', suggest 2G and 3G variant are similar, however, the author also claim that 1G and 2G variant are similar, it looks controversial, could the author clarify this?

The 2G variant is somewhat intermediate to the 1G and 3G variants. In our PCA and hierarchical clustering, 2G variants clustered with the 1G variants, which is why we treated these samples together.

The Summer's group has concluded that "the 2G and 3G variants conceal the DIS whilst exposing the cap to enhance translation". However, native agarose gels in their 2016 PNAS paper clearly show that 1G (lane 1) and 2G (lane 3) transcripts are more 'dimeric' than 3G transcripts (lane 5). Thus, their work is in line with our own results.

Figure 4a. Kharytonchyk et al. PNAS (2016).

- 1: 1G
- 2: 1Gcap
- 3: 2G
- 4: 2Gcap
- 5: 3G
- 6: 3Gcap

In the revised manuscript, we can modify the introduction to prevent confusion to readers not versed in these details of HIV-1 research.

#3 - Figure 3

Figure 3d looks interesting, but it is also arbitrary only show the DMS reactivity difference between 1G/2G dimer VS 3G monomer, is it possible also to show all the original reactivity or reactivity difference between monomer and dimer of 1G/2G, similar analysis between monomer and dimer of 3G?

All three reviewers made the same comment. This is addressed in detail in point #2 of reviewer 1.

In the revised manuscript, we will provide the additional analyses requested.

#4 - Figure 4, 'pooled DMS reactivities as soft constraints to guide *in silico* RNA folding' may be not enough to predict long region RNA-RNA interaction ...

We agree with the reviewer that 'pooled DMS reactivities as soft constraints to guide *in silico* RNA folding' may be not enough. This is exactly why we additionally present 2-dimensional structural analysis in Figure 5.

... is it possible to validate some of the interaction by directed RNA-RNA interaction method?

We appreciate the suggestion to validate the structural models using RNA-RNA interactome methods. However, we do not believe that these recently developed methods are currently suitable. These techniques use crosslinking reagents, such as 4'-aminomethyltrioxsalen hydrochloride (AMT), which have strong sequence preferences.

To convince the reviewer of this fact, below we show that AMT is unable to crosslink the well-known DIS interaction within SL1 of HIV 5'UTR (see figure below). This is because AMT has a strong preference for A-U base pairing, while the 6-nucleotides palindromic sequence within SL1 are all GC base-pairing.

AMT is unable to crosslink an RNA molecule corresponding to SL1, shown as absence of size shift on denaturing polyacrylamide gel.

How accurate/well established is it to use DMS to constrain structure models accurately?

DMS is a very well-established method for probing RNA structure, and is one of the oldest and most widely used RNA structural probes. The reviewer can refer to our paper on the evolution of RNA structural probing methods for more details (<https://pubmed.ncbi.nlm.nih.gov/30485688/>).

As stated in the manuscript, “The incorporation of information from RNA structural probing experiments dramatically improves the accuracy of RNA structure predictions, but structural elements can still be incorrectly predicted because data from chemical probing experiments typically provide information on whether a nucleotide is base-pair or not, but not its base pairing partner^{55,56}”

As mentioned in point #4 above, we would like to reemphasize that these limitations of classical DMS structural probing are solved using 2-dimensional DMS probing (see Figure 5)

a, d) the stem region shows very high DMS reactivity, which does not make much sense. Could the authors show the entire reactivity profile of all regions, as this will help to evaluate how accurate is the structure folding method?

Besides, it is confusing to show the structure folding model for both monomer and dimer together, does the author assume that the structure folding are same between monomer and dimer? It is better show the model separately side by side.

This comment is similar to point #6 from reviewer 2 (see response therein). Reviewer 3 has likely misunderstood Fig 4, where reactivities for the **both** monomer and dimer structures were placed on **both** structural models, using each half of the circle to represent the monomer and dimer data.

Since reviewer #2 and #3 found this representation confusing, we will follow the suggestion of reviewer #3 to present only the dimer data on the dimer structure, and only the monomer data on the monomer structure. We will move the hybrid figure to the supplementary data.

The reactivity profiles of all regions requested by the reviewer are already provided in the supplementary data as (i) reactivity plots, (ii) structure plots with DMS reactivities, and (iii) as tables of DMS reactivities.

#5 - Figure 5, Two-dimensional RNA structure probing is a good idea to predict RNA base-pairing partners. I have a similar question as figure 1: could the authors provide the entire mutational profile showing the mutational rates at each base for the error-prone and DMS treatment samples? Also, the figure lacks the detailed cutoff settings and validation to show how robust of the analysis method.

The mutational rates are already provided in figure format in the supplementary data for the pooled samples (see 2d-structure interference folder, files with mut_rates.pdf extension).

We thank the reviewer for the question regarding cutoff setting. We will include this detail in the revised materials and methods.

The detected interactions were extensively validated in Figure 6 and 7, but in the revised manuscript we will provide additional structural validation of the mutants (see #6 below).

#6 - Figure 6, Is it possible to validate the structure model for the mutants by using DMS mutational rate or other method, this could help to understand how much the mutation affect the RNA structure and dimerization.

In the revised manuscript we include additional independent verification of the structures in Figure 6b (see data in point #8 of reviewer two).

We have also independently verified the structure of four key mutants in cells that stabilize the monomer and dimer structures (see data in point #2 of reviewer two).

Additionally, point mutation for the fragment may not reflect the dimerization of full length HIV1, is it possible to validate the full length HIV1 dimerization for some interesting mutants.

As mentioned above, we have now validated the structures of key mutants on full length RNA in cellular assays. These mutants were introduced into the full length genome (deleted for Env for biosafety reasons).

#7 - Figure 7, The details are not clear, which one (1G/2G/3G or all together) were used here?

The validation experiments were conducted with the 3G RNA. We will add the details in the figure legends and material and methods in our revised manuscript.

Because mutations are one major reason cause dramatic RNA structure change, it is interesting to know whether the mutants in other region (for example in the mutants show in Figure 6) cause Pr55 binding.

We have tested additional mutants for Gag binding e.g. mutants targeting the polyA-SL1 interaction.

Both these mutants behave as predicted according to our model. Specifically, the monomer promoting mutation U86GA89C is impaired in Gag binding and the dimer promoting mutant C84AC85A is enhanced in Gag binding.

Decision Letter, first revision:

18th Jan 2022

Dear Redmond,

Thank you again for submitting your manuscript "Short- and long-range interactions in the HIV-1 5'UTR regulate genome dimerization and packaging". The reports of the referees are below, and based on these comments, we are happy to accept your paper, in principle, for publication as an Article in Nature Structural & Molecular Biology, on the condition that you revise your manuscript

in response to the comments of the referees and our editorial requirements.

You will see that reviewers 1 and 3 are satisfied with the revisions, but that reviewer 2 had remaining concerns, which you have addressed by email. We would like you to include the following items included in your rebuttal letter:

-a brief discussion to explain the differences between in vitro and in-cell DMS probing data, and the limits of the in-cell data to clarify reviewer 2's queries on this point.

-cite and refer in the text to the reference demonstrating that DMS has been successfully used as an RNA structural probe that, on benchmarking tests, outperforms SHAPE mapping in determining RNA secondary structures.

-incorporate the updated Sup.Fig. 9 provided in the rebuttal, where you have highlighted the positions of the diagnostic reactivity changes (as you have suggested in your rebuttal letter).

- include an updated Fig 7d with a clearer schematic (which you also agreed to in your rebuttal letter).

The text and figures require revisions. Note that, within a few days, we will send you detailed instructions for the final revision, along with information on editorial and formatting requirements. We recommend that you do not start revising the manuscript until you receive this additional information.

To facilitate our work at this stage, we would appreciate if you could send us the main text as a Word file. Please make sure to copy the NSMB account (cc'ed above).

Data availability: this journal strongly supports public availability of data. Please place the data used in your paper into a public data repository, or alternatively, present the data as Supplementary Information. If data can only be shared on request, please explain why in your Data Availability Statement, and also in the correspondence with your editor. Please note that for some data types, deposition in a public repository is mandatory - more information on our data deposition policies and available repositories can be found below:
<https://www.nature.com/nature-research/editorial-policies/reporting-standards#availability-of-data>

TRANSPARENT PEER REVIEW

Nature Structural & Molecular Biology offers a transparent peer review option for new original research manuscripts submitted from 1st December 2019. We encourage increased transparency in peer review by publishing the reviewer comments, author rebuttal letters and editorial decision letters if the authors agree. Such peer review material is made available as a supplementary peer review file. **Please state in the cover letter 'I wish to participate in transparent peer review' if you want to opt in, or 'I do not wish to participate in transparent peer review' if you don't.** Failure to state your preference will result in delays in accepting your manuscript for publication.

Please note: we allow redactions to authors' rebuttal and reviewer comments in the interest of confidentiality. If you are concerned about the release of confidential data, please let us know specifically what information you would like to have removed. Please note that we cannot incorporate redactions for any other reasons. Reviewer names will be published in the peer review files if the reviewer signed the comments to authors, or if reviewers explicitly agree to release

their name. For more information, please refer to our [FAQ page](https://www.nature.com/documents/nr-transparent-peer-review.pdf).

ORCID

Nature Structural & Molecular Biology is committed to improving transparency in authorship. As part of our efforts in this direction, we are now requesting that all authors identified as 'corresponding author' create and link their Open Researcher and Contributor Identifier (ORCID) with their account on the Manuscript Tracking System (MTS) prior to acceptance. ORCID helps the scientific community achieve unambiguous attribution of all scholarly contributions. For more information please visit <http://www.springernature.com/orcid>

For all corresponding authors listed on the manuscript, please follow the instructions in the link below to link your ORCID to your account on our MTS before submitting the final version of the manuscript. If you do not yet have an ORCID you will be able to create one in minutes. <https://www.springernature.com/gp/researchers/orcid/orcid-for-nature-research>

IMPORTANT: All authors identified as 'corresponding author' on the manuscript must follow these instructions. Non-corresponding authors do not have to link their ORCIDs but are encouraged to do so. Please note that it will not be possible to add/modify ORCIDs at proof. Thus, if they wish to have their ORCID added to the paper they must also follow the above procedure prior to acceptance.

To support ORCID's aims, we only allow a single ORCID identifier to be attached to one account. If you have any issues attaching an ORCID identifier to your MTS account, please contact the [Platform Support Helpdesk](http://platformsupport.nature.com/).

We hope that you will support this initiative and supply the required information. Should you have any query or comments, please do not hesitate to contact me.

In recognition of the time and expertise our reviewers provide to Nature Structural & Molecular Biology's editorial process, we would like to formally acknowledge their contribution to the external peer review of your manuscript entitled "Short- and long-range interactions in the HIV-1 5'UTR regulate genome dimerization and packaging". For those reviewers who give their assent, we will be publishing their names alongside the published article.

"Nature Structural & Molecular Biology has now transitioned to a unified Rights Collection system which will allow our Author Services team to quickly and easily collect the rights and permissions required to publish your work. Once your paper is accepted, you will receive an email in approximately 10 business days providing you with a link to complete the grant of rights. If you choose to publish Open Access, our Author Services team will also be in touch at that time regarding any additional information that may be required to arrange payment for your article.

If you have any questions, please do not hesitate to contact me directly.

With kind regards,

Beth

Beth Moorefield, Ph.D.
Senior Editor
Nature Structural & Molecular Biology

Reviewer #1 (Remarks to the Author):

The authors satisfactorily addressed our comments.

Reviewer #2 (Remarks to the Author):

Although the authors did the in-cell DMS probing on four individual mutations, the DMS profiles for the in-cell ones are quite different from the ones for in vitro. There is no description and discussion on these differences. Also, it is better to perform the in-cell DMS probing under low salt and high salt to link the biological functions.

Further comments to the responses:

#1. It is possible to infer the U and G base pairing status based on the A and C modification.

However, DMS itself prefers A over C. Thus, the modification efficiency of A and C are quite different, subsequently leading to the bias in estimating the base pairing status of U and G. Several previous studies including the first MaP method were based on the SHAPE probing, due to the achievement of four nucleotide structural information. The dimerization requires clear base pairing information for the four nucleotides. It is better to have the SHAPE dataset to support the conclusion rather than estimating U and G information based on A and C.

#2. To validate the new analysis method, particularly for probability models, it is important to independently test at least one or two RNAs with the known structures (such as rRNAs, tRNAs, and snRNAs). These validations support not only the new analysis method but also the estimation of the base pairing status of U and G based on A and C.

#3. The 3.4 fold and 7.8 fold increase in mutation rates could not address the low signal to noise issue. In SupFigure 2, the mutation rate for C was lower than 0.02 and the mutation rate for A was roughly 0.05. Going through all the supplementary 2D-structure-interference figures, the rate is still very low. The main concern is that the low mutation efficiency is not sensitive enough to distinguish the structural disruption.

#6. The quantitative measurement is to quantitatively justify the true/false positive and the true/false negative DMS signals. In the Figure 4, the nucleotides on the stem are with high DMS reactivity which could not be used as "fit to the structure".

#7. The Figure 6 could be colored with the third color for 2-7. This will be much clearer. The cap of 2 requires the justification. The data presentation should not be selected without clear justifications.

#8. The arrows in the Figure 6B are not a good way to present data. It is also very hard to understand the SuppFigure 9 without the nucleotides where A235C and A239C seem not change the structures at all.

Minor comments:

1. In Figure 2a, cap1G did not form dimerization in low salt condition where Brown J (Science 2020) showed cap1G adopts the dimerization. Different conditions?
2. All the structure model figures are not clear with bad figure quality in Figure 4, 5, and 6. The coloring styles and structure model presentations are very different in three figures. In figure 6, the nucleotides were partially colored. Why?

Reviewer #3 (Remarks to the Author):

The authors have answered most of my concerns. I do not have any additional concerns at this moment.

Author Rebuttal, first revision:

Reviewer #2: Remarks to Author

Although the authors did the in-cell DMS probing on four individual mutations, the DMS profiles for the in-cell ones are quite different from the ones for in vitro. There is no description and discussion on these differences.

In the revised manuscript, the data presented in Sup. Fig. 10 and Sup. Fig 11 establish structural changes for mutants that have altered monomer / dimer equilibriums.

In vitro (Sup. Fig. 10c and f) ensemble reactivities for the dimer- and monomer promoting mutants match the reactivities obtained from the isolated monomer and dimer. This demonstrates that the mutants alter the monomer/dimer equilibrium and produce the structural changes predicted by our model.

In cells (Sup. Fig. 11), reactivity changes from chemical probing experiments must be interpreted carefully because low reactivities can be induced by RNA structure *and* by cellular RNA binding proteins. It is therefore not surprising, nor unusual, that the DMS reactivities *in cell* and *in vitro* are not identical.

Nevertheless, despite this caveat, we were able to see that the dimer promoting mutants folded into a structure containing SL1, whereas monomer promoting mutants folded into structures where SL1 was hidden through long- and short-range interactions with polyA and PBS. We conclude that the regulatory mechanism we identified *in vitro* also takes place in cells.

We would be happy to include a brief discussion to explain the differences between *in vitro* and *in cell* probing data, and the limits of the *in cell* data.

Also, it is better to perform the in-cell DMS probing under low salt and high salt to link the biological functions.

Our experimental set-up does not permit changes in salt concentrations within cells. Even if this were possible, data interpretation would be extremely challenging because salt affects the binding of proteins to the RNA.

Further comments to the responses:

#1. It is possible to infer the U and G base pairing status based on the A and C modification.

However, DMS itself prefers A over C. Thus, the modification efficiency of A and C are quite different, subsequently leading to the bias in estimating the base pairing status of U and G. Several previous studies including the first MaP method were based on the SHAPE probing, due to the achievement of four nucleotide structural information. The dimerization requires clear base pairing information for the four nucleotides. It is better to have the SHAPE dataset to support the conclusion rather than estimating U and G information based on A and C.

The reviewer claims that DMS prefers A over C, leading to a bias in estimating the base pairing status of U and G.

In our rebuttal to reviewer number 3, we cited literature comparing the accuracy of DMS vs SHAPE in predicting RNA secondary structure (see <https://pubs.acs.org/doi/10.1021/bi3008802>). We quote from the paper “*We were surprised that DMS mapping gave similar or better information content, compared to SHAPE data, as the latter provides reactivities at approximately twice the number of nucleotides per RNA*”. Thus, DMS can be successfully used as an RNA structural probe, and on benchmarking tests, it can even outperform SHAPE reagents.

The reviewer also claims that we ‘estimate U and G information based on reactivities at A and C’. This is not correct. The final models presented in Fig. 5 and 6 are also based on *two-dimensional* signals derived from mutations *at all four nucleotides*. This was already pointed out in the first rebuttal (see previous answer to point #1), and is a major theme of the manuscript (see Fig. 5).

#2. To validate the new analysis method, particularly for probability models, it is important to independently test at least one or two RNAs with the known structures (such as rRNAs, tRNAs, and snRNAs). These validations support not only the new analysis method but also the estimation of the base pairing status of U and G based on A and C.

As stated in our previous rebuttal, the dimer model provides an internal control for the 2d-analysis, and this internal control captured many important and known features of the HIV-1 5'UTR.

Furthermore, validating the method on another RNA would constitute an entirely new study. In discussions with the editor before submitting the revised manuscript, it was agreed that additional validation was not needed.

#3. The 3.4 fold and 7.8 fold increase in mutation rates could not address the low signal to noise issue. In SupFigure 2, the mutation rate for C was lower than 0.02 and the mutation rate for A was roughly 0.05. Going through all the supplementary 2D-structure-interference figures, the rate is still

very low. The main concern is that the low mutation efficiency is not sensitive enough to distinguish the structural disruption.

Note that in Sup. Fig. 2 (and throughout the manuscript) mutations rates are not depicted as percentages, but rather as floating-point values. The mutation rate for C and A, when converted to percentages are 2% and 5%, respectively. These mutation rates are in line, or better, than equivalent DMS-MaP-seq work from others (PMID 27819661; 32555469).

Below, we attach Sup. Fig. 2 from the original DMS-MaP study from the Rouskin lab

(<https://www.ncbi.nlm.nih.gov/pmc/articles/PMC5508988/>), which shows mismatch frequencies of ~2.5-3%.

Sup. Fig. 2 from Zubradt *et al.* PMID 27819661

These DMS-MaP mutation frequencies are also *much higher* than those obtained in equivalent SHAPE-MaP experiments (below)

Below, we attached Sup. Fig. 4 from the original SHAPE-MaP study from the Weeks lab

(<https://www.ncbi.nlm.nih.gov/pmc/articles/PMC4259394/>), which shows mismatch frequencies of ~0.5%.

Sup. Fig. 4 from Siegfried *et al.* PMID 25028896

From a statistical standpoint, the ability to detect signals depends on both the effect size (= signal-to-noise ratio), as well as the sample size (= sequencing depth for probing experiments). In our analysis, the signal-to-noise ratio is better than achieved in previous studies, as discussed above. In addition, we chose a sequencing depth >2 million per sample, such that signals can be detected confidently.

The mutational rate for the 2d analysis is obviously lower than for single nucleotides, but we are still able to clearly identify stem-signals above the background noise (because the background noise also decreases in the 2d analysis). We do not claim to identify all RNA helices, but we were clearly able to identify 7 known helices in the consensus dimer structure (Fig 5c); in the monomer structure, three of these helices disappeared, and were replaced by alternative signals (Fig 5f) corresponding to the novel interactions.

In conclusion, Fig. 5 demonstrates that the 2d signal is strong enough to distinguish major structural changes.

Performing SHAPE-MaP experiments, with its lower signal-to-noise, would not improve these signals.

#6. The quantitative measurement is to quantitatively justify the true/false positive and the true/false negative DMS signals. In the Figure 4, the nucleotides on the stem are with high DMS reactivity which could not be used as “fit to the structure”.

In Fig 4, we do not see any signals in the stems that would conflict with our models. Nucleotides adjacent to loop regions are commonly reactive due to ‘breathing of the stems’ (intermittent opening/closing at the ends of the stem region), and perhaps these are positions that the reviewer is referring to.

In the revised manuscript, we already included non-parametric bootstrapping as additional statistical support for our claims. We do not believe further analysis is required.

However, for the reviewer’s interest, we have calculated receiver operating characteristic (ROC) curves to quantify the true positive / false positive DMS signals for both data sets.

These ROC curves show that the dimer and monomer data fit very well to their respective structures (ROC of 0.89 and 0.85 for dimer and monomer data, respectively).

#7. The Figure 6 could be colored with the third color for 2-7. This will be much clearer. The cap of 2 requires the justification. The data presentation should not be selected without clear justifications.

The justification was already provided in the previous rebuttal. That is, the cap is to better visualise the blue (dimerization enhancing) mutations, which were generally much weaker than the (dimerization inhibiting) mutations. We believe this representation is justified because (i) there is no data transformation, (ii) the representation is necessary to qualitatively interpret the data, and (iii) all raw and processed data is provided as extensive supplementary data.

We feel that a third colour would complicate the red (negative) - blue (positive) colour scheme. Also, the capped colours are already represented by a unique dark red colour.

We will discuss this point with the editor, and follow her guidance.

#8. The arrows in the Figure 6B are not a good way to present data. It is also very hard to understand the SupFigure 9 without the nucleotides where A235C and A239C seem not change the structures at all.

Below, we include an updated Sup. Fig. 9 where we have highlighted with red arrows the positions showing the diagnostic reactivity changes. We believe the Figure is clearer, and we thank the reviewer for helping us to make this improvement.

For this reason, we disagree with the reviewer that arrows are not a good way to highlight subtle changes the data that might be difficult to spot otherwise.

Minor comments:

1. In Figure 2a, cap1G did not form dimerization in low salt condition where Brown J (Science 2020) showed cap1G adopts the dimerization. Different conditions?

There are several reasons for this discrepancy, the two most important are discussed below.

First, the sample of Fig. 1c of Brown (science 2020) is run in PI buffer, which is equivalent to the high salt condition used in our study.

Second, as already discussed in the manuscript, Brown (science 2020) used a different HIV-1 strain Mal, compared to NL43 used in our study. There are important differences in the primer binding site (PBS) domain between these two strains that potentially interrupt the PBS-SL1 interaction seen in our work.

Nevertheless, the dimerization behaviour of the 1G vs 3G samples for the NL43 strain largely agrees with the work from the Summers group (e.g. PMID 27834211; 3232759), even if we postulate that there are alternative molecular mechanism at play.

2. All the structure model figures are not clear with bad figure quality in Figure 4, 5, and 6. The coloring styles and structure model presentations are very different in three figures. In figure 6, the nucleotides were partially colored. Why?

We apologise for the poor quality of the reviewer's copy. In our uploaded version the figures are in vector art format, so there should be no problems with zooming and/or quality.

In figure 6, the nucleotides are partly coloured to represent the effects of individual mutations (as noted in the figure and figure legend).

We will also update Fig 7d with a clearer schematic.

Final Decision Letter:

14th Feb 2022

Dear Redmond,

We are now happy to accept your revised paper "Short- and long-range interactions in the HIV-1 5'UTR regulate genome dimerization and packaging" for publication as a Article in Nature Structural & Molecular Biology.

As soon as your article is published, you can generate your shareable link by entering the DOI of your

article here: http://authors.springernature.com/share.

Corresponding authors will also receive an automated email with the shareable link

Note the policy of the journal on data deposition:

<http://www.nature.com/authors/policies/availability.html>.

Your paper will be published online soon after we receive proof corrections and will appear in print in the next available issue. You can find out your date of online publication by contacting the production team shortly after sending your proof corrections. Content is published online weekly on Mondays and Thursdays, and the embargo is set at 16:00 London time (GMT)/11:00 am US Eastern time (EST) on the day of publication. Now is the time to inform your Public Relations or Press Office about your paper, as they might be interested in promoting its publication. This will allow them time to prepare an accurate and satisfactory press release. Include your manuscript tracking number (NSMB-A45048C) and our journal name, which they will need when they contact our press office.

About one week before your paper is published online, we shall be distributing a press release to news organizations worldwide, which may very well include details of your work. We are happy for your institution or funding agency to prepare its own press release, but it must mention the embargo date and Nature Structural & Molecular Biology. If you or your Press Office have any enquiries in the meantime, please contact press@nature.com.

Please note that *Nature Structural & Molecular Biology* is a Transformative Journal (TJ). Authors may publish their research with us through the traditional subscription access route or make their paper immediately open access through payment of an article-processing charge (APC). Authors will not be required to make a final decision about access to their article until it has been accepted. Find out more about Transformative Journals

Authors may need to take specific actions to achieve compliance with funder and institutional open access mandates. For submissions from January 2021, if your research is supported by a funder that requires immediate open access (e.g. according to Plan S principles) then you should select the gold OA route, and we will direct you to the compliant route where possible. For authors selecting the subscription publication route our standard licensing terms will need to be accepted, including our self-archiving policies. Those standard licensing terms will supersede any other terms that the author or any third party may assert apply to any version of the manuscript.

With kind regards,

Beth

Beth Moorefield, Ph.D.
Senior Editor
Nature Structural & Molecular Biology